# High-precision U-Pb ages in the Early Tithonian to Early Berriasian and implications for the numerical age of the Jurassic/Cretaceous boundary

Luis Lena[1], Rafael López-Martínez[2], Marina Lescano[3], Beatriz Aguirre-Urrreta[3], Andrea Concheyro[3], Verónica Vennari[3], Maximiliano Naipauer[3], Elias Samankassou[1], Márcio Pimentel[4], Victor A. Ramos[3], Urs Schaltegger[1]

[1]Department of Earth Sciences, University of Geneva, Geneva, 1205, Switzerland
[2]Instituto de Geología, Universidad Nacional Autónoma de Mexico, Ciudad de Mexico, 04510, Mexico
[3]Instituto de Estudios Andinos Don Pablo Groeber (UBA-CONICET), Universidad de Buenos Aires, Buenos Aires, 1428, Argentina
[4]Instituto de Geociências, Universidade de Brasilia, Brasilia, DF, 70910-900, Brasil

*Correspondence to*: Luis Lena (lena.luis@gmail.com; Luis.FortesDeLena@unige.ch )

**Abstract.** The numerical age of the Jurassic/Cretaceous boundary has been controversial and difficult to determine. In this study, we present high-precision U-Pb geochronological data around the Jurassic/Cretaceous boundary in two distinct sections from different sedimentary basins: the Las Loicas, Neuquén Basin, Argentina, and the Mazatepec, Oriental Sierra Madre, Mexico. These two sections contain primary and secondary fossiliferous markers for the boundary as well as interbedded volcanic ash horizons allowing to obtain new radio-isotopic dates in the Late Tithonian and Early Berriasian. We also present the first age determinations in the Early Tithonian and tentatively propose a minimum duration for the stage as a cross check for our ages in the early Berriasian. Given our radio-isotopic ages in the Early Tithonian to Early Berriasian, we discuss the implications for the numerical age of the boundary.

## 1. Introduction

The age of the Jurassic/Cretaceous (J/K) boundary remains one of the last standing Phanerozoic system boundaries with a numerical age not tied by adequate radio-isotopic data. The numerical division of the geological record is ultimately dependent on accurate and precise radio-isotopic ages of well-defined fossiliferous datums. Over the years the numerical age of the J/K boundary has been difficult to measure due to the lack of datable horizons close to boundary markers, which made it difficult to ascribe a radio-isotopic age directly on fossiliferous datums. Consequently, the ill-defined age of the boundary has led to widely variable timescales for the Late Jurassic to Early Cretaceous (Channell et al., 1995; Gradstein et al., 1995; Lowrie and Ogg, 1985; Malinverno et al., 2012; Ogg, 2012; Ogg et al., 1991; Ogg and Lowrie, 1986; Pálfy, 2008; Pálfy et al., 2000a). These various approaches attempted to ascribe an age to the J/K boundary; nevertheless, the different estimates

for the age of the boundary lacked reproducibility varying from 135 to 145 Ma with a high degree of uncertainty with very little overlap. The most recently used timescale of the Late Jurassic is the M-sequence model of Ogg (2012). The model relies on the integration of data from a variety of fields such as M-sequence magnetic anomalies from the northwest Pacific Ocean, magnetostratigraphy, biostratigraphy, cyclostratigraphy, and scarce radio-isotopic ages. The model is based on the marine magnetic anomalies timescale of the northwestern Pacific Ocean (Channell et al., 1995; Larson and Hilde, 1975; Tamaki and Larson, 1988). The interval encompasses ~1000 km of oceanic crust over a period of ~35 Ma in the northwestern Pacific. The age of the polarity changes in the northwestern Pacific was dated by key fossiliferous datums from Mediterranean Tethys sedimentary sequences via the correlation with magnetostratigraphy in these sequences (Grabowski, 2011; Kent and Gradstein, 1985; Ogg et al., 1991; Ogg and Lowrie, 1986). The duration of the magnetic reversal changes are provided by cyclostratigraphic studies (Huang et al., 2010a, 2010b) for some of the magnetozone intervals and thus used to calculate a decreasing spreading rate with the distance associated with the magneto anomalies in the Hawaiin spreading center. The numerical age of stage boundaries from the Berriasian to Oxfordian were then back-calculated from the age of the M0n at the base of the Aptian. The age of the M0 used was 126.3±0.4 Ma which is the combination of the cycle duration of the Albian stage (Huang et al., 2010a) tied to a U-Pb age from the Aptina/Albian boundary of 113.1±0.3 Ma (Selby et al., 2009). This linear fitting model is the basis for various Late Jurassic and Early Cretaceous stage boundary numerical ages.

In the specific case of the J/K boundary the projected age of the M-sequence age model was 145.0±0.8 Ma (Ogg, 2012) which is almost identical to the radio-isotopic age reported in Mahoney et al. (2005) of 145.5±0.8 Ma (recalculated by Gradstein, 2012) for the sill intruded in Berriasian sediments in the Shatsky Rise with magnetization M21-M20. Furthermore, the magnetization of borehole 1213B is reasonably close to what has become a reliable secondary marker for the J/K boundary, the M19.2n (Wimbledon, 2017 and references therein). However, studies that obtained radio-isotopic ages directly on sedimentary sequences that spanned the J/K boundary reveal much younger ages for the boundary (Bralower et al., 1990; López-Martínez et al., 2015; Vennari et al., 2014).

Recently, the base of the Calpionella Zone, Alpina Subzone, has been selected as a principal biostratigraphic marker for the base of the Berriasian (Wimbledon, 2017). Nevertheless, its presence alone is not sufficient to locate the boundary, and secondary markers such as calcareous nannofossils and magnetostratigraphy are essential additional constraints to aid with the definition of the boundary, with the latter allowing sections to be normalized against a global framework. The most complete studies of the J/K boundary from a biostratigraphical and magnetostratigraphic standpoint are located in Mediterranean Tethys. Nevertheless, the radiometric age of the boundary is poorly defined in the Mediterranean Tethys due to the absence of active volcanism close by during the time of deposition of these sedimentary sequences. In this way, the western Tethys (proto-Gulf of Mexico) and the Austral Basins (Neuquen Basin, Argentina) offer a good opportunity to advance the study on the radio-isotopic age of the J/K boundary. Contrary to the Mediterranean Tethys, the sedimentary sequences in the proto-Gulf and Austral realms were deposited close to active plate boundaries where significant volcanism took place, which enabled the deposition of datable horizons suitable for U-Pb geochronology. Recently, calpionellid biostratigraphy has been reported in both regions (López-Martínez et al., 2013b, 2017) opening the possibilities for better

correlations with the Mediterranean Tethys. It is worth noting that even if calpionellid biostratigraphy of the Neuquén basin is still not complete and global correlations are still tentative, for now, they are the only known basins with occurrences of calpionellid as markers around the J/K boundary in the Austral realm with abundant datable horizons. A general definition of the J/K boundary would, however, need to be of global validity and allow correlation with the Tethys realm.

In the present study, we date two independent sections: one in Mexico and one in Argentina using precise radio-isotopic geochronological methods. We present high-precision U-Pb age determinations using chemical abrasion, isotope dilution, thermal ionization mass spectrometry (CA-ID-TIMS) techniques to date zircon from interbedded volcanic ash layers in the Las Loicas section, Neuquén Basin, Argentina, and the Mazatepec section, Mexico. Such dates have proved to yield robust estimates for the timing of the stratigraphic record especially in combination with Bayesian age-depth modeling

(e.g., Ovtcharova et al., 2015; Baresel et al., 2017; Wotzlaw et al., 2018). The coupling of high-precision U-Pb geochronology and age-depth modeling allowed us to ascribe specific numerical ages to key taxa in the Early Berriasian, Late Tithonian in the studied sections. We also report new nannofossil data from the section in Mexico such as the FO of *Nannoconus steinmannii steinmannii* and FO of *Nannoconus kamptneri minor*, respectively (Fig. 2). Additionally, we also present the first radio-isotopic age in the Early Tithonian at the base of the *Virgatosphinctes andesensis* biozone in the La

Yesera section, Neuquén basin, close to the Kimmeridgian/Tithonian boundary (KmTB) (Riccardi, 2008, 2015; Vennari, 2016). Lastly, our geochronological data allows us to re-evaluate the numerical age of the J/K boundary and discuss some complications with the currently accepted age of ~145 Ma.

## 2. Geological context and studied sections

To investigate the numerical age of the J/K boundary, we have selected two sections where J/K boundary markers such

as ammonites, calpionellids, and calcareous nannoplankton have been recognized. The first section is Las Loicas, exposed along the national road 145 (Argentina), from Bardas Blancas to the international border at the Pehuenche Pass. It is located near the Argentine-Chilean border, approximately one kilometer to the southwest of the settlement Las Loicas (Fig. 1). Geologically, the Las Loicas section (Vennari et al., 2014) is located in the Vaca Muerta Formation, Neuquén Basin, Argentina (Fig. 1). The Neuquén Basin in western Argentina accumulated an almost continuous record of 7000 m of

sediments from the late Triassic to Early Cenozoic. The basin is located on the eastern side of the Andes in Argentina between $32^{o}$ and $40^{o}$ S latitude (Fig. 1). The basin has a triangular shape, covers an area of over $1200^{2}$ km, and is bounded to the west by the Andean magmatic arc on the active margin of the South American Plate, to the northeast by the San Rafael Block, and to the southeastern part by the North Patagonia Massif (Fig. 1). Two main regions are commonly recognized in the basin: The Neuquén Andes to the west and the Neuquén Embayment to the east (Fig. 1). The Neuquén Embayment is

relatively undeformed, in contrast to the Neuquén Andes where the late Cretaceous-Cenozoic deformation has resulted in the development of a series of N-S oriented fold and thrust belts: Aconcagua, Malargüe, and Agrio, where a substantial part of the Mesozoic sequence outcrops (Legarreta and Uliana, 1991, 1996).

The Vaca Muerta Fm. is a 217 m-thick sedimentary sequence of marine shales and limestones, which spans an interval from the Lower Tithonian (*Virgatosphinctes andesensis* biozone) to the upper Berriasian (*Spiticeras damesi* biozone) (Aguirre-Urreta et al., 2005; Kietzmann et al., 2016; Riccardi, 2008, 2015). In Las Loicas, the *Substeueroceras koeneni* and *Argenticeras noduliferum* ammonite biozone and calcareous nannofossils have been described by Vennari et al. (2014).

Recently, López-Martínez et al. (2017) reported the occurrence of upper Tithonian to lower Berriasian calpionellids, which is the only known section where the primary markers for the J/K boundary occur together in Argentinian Andes. The section contains several ash beds, which allowed precise age bracketing of the boundary using high-precision U-Pb geochronology.

The La Yesera Section is located 50 km north of the town of Chos Malal in the northern sector the Neuquén Basin (Fig. 1) and is exposed along the national road 40. Geologically, the La Yesera section (Fig. 2C) represents a distal portion of the

basin farther from the magmatic arc than the Las Loicas section. Tuff beds are less frequent than in the Las Loicas section and generally thinner. The section has a total thickness over 400 m and is one of the best continuous exposures of Tithonian ammonite zones, from the Early Tithonian *Virgatosphinctes mendozanus* to the *Neocomites wichmanni /Early Valanginian* (Aguirre-Urreta et al., 2014). The section also has one of the best-exposed contacts between the Vaca Muerta Fm and the Tordillo Fm.

The Mazatepec section is located in the Puebla State, Mexico, southeast of Mexico City. Geologically, the Mazapetec section exposes the Pimienta and the lower Tamaulipas formations of the Oriental Sierra Madre geological province, Mexico (Fig. 1). The Oriental Sierra Madre is one of the many tectonic terranes composed of Mesozoic volcano-sedimentary sequences deformed during the Late Cretaceous and Early Cenozoic during the Laramide Orogeny in Mexico (Campa and Coney, 1983; Suter, 1980). A rift sequence characterizes the tectonic evolution of the proto-Gulf in the Late Triassic-

Oxfordian due to the rifting of Pangea characterized by continental sedimentation controlled by narrow grabens with no marine sedimentation taking place (Salvador, 1987). The post-rift phase is characterized by ample marine carbonate platforms of shallow waters. During the Tithonian to Early Cretaceous, a stable tectonic and climatic conditions prevailed with the sedimentation being significantly slower with the development of shallow marine water sedimentation, namely the deposition of Pimienta Fm. (carbonates) and Tamaulipas Fm. (argillaceous limestones, shales) (Padilla & Sánchez, 2007).

The Pimienta Fm. is composed of darkish clayey limestones and the Tamaulipas Fm. is a gray limestone (López-Martínez et al., 2013; Suter, 1980) The section has a dense occurrence of Late Tithonian Crassicollaria Zone (Colomi Subzone) and Early Berriasian calpionellids from Calpionella Zone (Alpina, Ferasini, and Elliptica Subzones) to Calpionellopsis Zone (Oblonga Subzone). In the upper part of the section, ash beds are scarce and occur at distinct levels. Ash bed MZT-81 is situated within the Elliptica Subzone in the lower Tamaulipas Formation (Fig. 2B).

**3. Methods**

The nannofossil biostratigraphy for the Mazatepec section was based on 17 samples from the Pimienta and Tamaulipas formations. For detailed calcareous nannofossil examination, simple smear slides were prepared using standard procedures

(Edwards, 1963). Observations were made and photographs were taken using a polarizing microscope Leica DMLP with increased 1000X and accessories such as λ one sheet of plaster and blue filter. The slides are deposited in the Repository of Paleontology, Department of Geological Sciences, University of Buenos Aires, under the catalog numbers BAFC-NP: Nº 4190-4206 photomicrographs of selected species are shown in Fig. 3; the distribution chart for the calcareous nannofossil species is presented in Supplementary Fig. S1.

We have used U-Pb zircon CA-ID-TIMS dating techniques to single zircon grains, which yields $^{206}Pb/^{238}U$ dates at 0.1-0.05% precision. The depositional age of ash beds has been calculated from the weighted means of the four youngest overlapping $^{206}Pb/^{238}U$ dates (Fig. 4), assuming that older grains record prolonged residence of zircon in the magmatic systems as well as intramagmatic recycling. In the text, all quoted ages of ash beds are weighted mean $^{206}Pb/^{238}U$ ages corrected for initial $^{230}Th$ disequilibrium.

The age of the various paleontological markers in Las Loicas have been calculated using the Bayesian age-depth model Bchron of Haslett and Parnell (2008) and Parnell et al. (2008). The model outputs an uncertainty envelope which is presented in Fig. 2B. The age-depth results are reported in TS.2, with age assigned to every meter of stratigraphic height. The Bchron code used in the R statistical package environment (R Core Team 2013) is included in the Supplementary Materials section 6.

## 4. Results

### 4.1 Calcareous nannofossils biostratigraphy in Mazatepec

Eighteen nannofossil species have been recognized in Mazatepec (Fig. S1). The heterococcoliths are mostly represented by Watznaueriaceae including *Watznaueria barnesae*, *W. britannica, W. manivitae, Cyclagelosphaera margerelii, and C. deflandrei; Zeugrhabdotus embergeri* is another frequent constituent. The nannoliths are represented by *Conusphaera mexicana, Polycostella senaria, Hexalithus noeliae, Nannoconus globulus,* and *N. kamptneri minor.* These nannofossils indicate Late Tithonian to Early Berriasian age for the Pimienta Formation and the lower part of the Tampaulipas Formation. The assemblage composed by *Conusphaera mexicana, Polycostella scenario,* and *Hexalithus noeliae*, indicates a Late Tithonian age. The only useful biological event recognized is the FO of *N. kamptneri minor.* An increase in the diversity of nannofossils is identified with 11 species among which, the presence of *N. steinmannii steinmannii* stands out (Fig. 2B).

### 4.2 U-Pb geochronology, age interpretations, age-depth modeling

A total of six ash beds were dated: four in the Las Loicas section, one in the Mazatepec section, and one in the La Yesera section. In the Las Loicas section, LL3 yielded an age of 139.238 ± 0.049/0.061/0.16 Ma, LL9 139.956 ± 0.063/0.072/0.17 Ma, LL10 140.338 ± 0.083/0/091/0.18 Ma and LL13 and age of 142.039 ± 0.058/0.069/0.17 Ma. In La Yesera, ash bed LY5 yielded an age of 147.112 ± 0.078/0/088/0.18 Ma, and in Mazatepec MZT-81 yielded an age of

140.512 ± 0.031/0/048/0.16 Ma (Fig. 4). All zircons considered in the age distribution of the ash are interpreted from ash-fall deposits from near-by volcanic eruptions. The final weighted mean ages are interpreted as a depositional age for each ash bed. Uncertainties are reported as X/Y/Z where X includes analytical uncertainty, Y includes additional tracer (ET2535) calibration uncertainty, and Z includes additional $^{238}$U decay constant uncertainty. A full and detailed description of the techniques, sample preparation, laboratory procedures, data acquisition, as well as data treatment are provided in the Supplementary Materials. The full U-Pb data set is reported in Table S1. Age-depth statistical modeling was performed outputting a numerical age for every meter of the Las Loicas sections, with a 95% confidence precision interval. The results on a meter-by-meter resolution are reported in Table TS.2.

### 4.3 Numerical age of faunal assemblages in studied sections

In Fig. 2A, the various markers, and assemblages are indicated as well as the age of the ash beds. In Las Loicas, López-Martínez et al. (2017) reported Late Tithonian Crassicollaria Zone, Colomi Subzone (Upper Tithonian) based on the occurrence of *Calpionella alpina* Lorenz, *Crassicollaria colomi* Doben, *Crassicollaria parvula* Remane, *Crassicollaria massutiniana* (Colom), *Crassicollaria brevis* Remane, *Tintinnopsella remanei* (Borza) and *Tintinnopsella carpathica* (Murgeanu and Filipescu), the First Occurrence (FO) of *Umbria granulosa granulosa* and *Substeueroceras koeneni* ammonite Zone (Vennari et al., 2014). Our Bchron model age predicts an age of 141.31 ± 0.56 Ma for the faunal assemblage of *Crassicollaria parvula* and *Crassicollaria colomi* and the FO of *Umbria granulosa granulosa* Fig. 2B). Another Late Tithonian marker in Las Loicas is the FO *Rhagodiscus asper*, also within the Crassicollaria Zone a Bchron age of 140.60 ± Ma (Fig. 2A).

In Las Loicas some Early Berriasian markers are present. For instance, the FO of *Nannoconus kamptneri minor* (Fig. 2A, Fig. S1) and *Nannoconus steinmannii minor* are considered indicators of the Early Berriasian (Vennari et al., 2014). Here they overlap with the base of the *Arginiceras noduliferum* ammonite Zone (López-Martínez et al., 2017; Vennari et al., 2014). The occurrence of the acme of *Calpionella alpina* (small and spheric) and scarce specimens of *Crassicollaria massutiniana, Tintinnopsella remanei,* and *T. carpathica* suggests an Early Berriasian age (López-Martínez et al., 2017) (Fig. 2A). These assemblages are bracketed by ash beds LL9 (139.956 ± 0.063 Ma) and LL10 (140.338 ± 0.083 Ma) (Fig. 2A) and overlaps with the FO of *Nannoconus kampteri minor* and *Nannoconus steinmannii minor*, the base of *Arginiceras noduliferum* Zone, and the base of the Alpina Subzone (ca., 34 m stratigraphic height) (Fig. 2A) The Bcrhon model age for this assemblage is 140.22 ± 0.13Ma (Fig. 2A).

In Mazatepec, ash bed MZT-81 is located within the Elliptica Subzone and has an age of 140.512 ± 0.031Ma. (Fig. 4). Due to the lack of datable horizons close to the Alpina Subzone in Mazatepec we have resorted to assumed sedimentation rates to back-calculate the age of base of the Alpina Subzone. Here we assume a sedimentation rate to be 2.5 cm/ka. Although there is no data on actual sedimentation rates in the Pimienta Fm., this rate is realistic for similar coeval deposits (e.g., Grabowski et al., 2010) as well as with the tectonic and environmental stability of the Oriental Sierra Madre in the

Tithonian-Berriasian stages (Padilla & Sánchez, 2007). It is worth noting that our new data allows only a confident numerical age for the Elliptica Subzone (Fig. 5).

Ash bed (LY-5) located below the contact, and it yielded an age of 147.112 ± 0.078 Ma (Fig. 2C). The ash bed is located in the Tordillo Fm, 1.5m below the contact with the Vaca Muerta Formation, thus very close to the base of the *Virgatosphinctes andesensis* Zone.

## 5. Discussion

### 5.1 The chronostratigraphic and biostratigraphic framework of the studied sections

In the past decade significant strides have been made in fixating the J/K boundary by coupling calpionelids, calcareous nannofossils, ammonites, and magnetostratigraphy (Wimbledon, 2017; Wimbledon et al., 2011). Correlations between sections within the Mediterranean Tethys have become consistent to the point of a trustworthy correlation framework being developed for the various markers (calpionellids, nannofossils, ammonites, and magnetostratigraphy) for the J/K boundary (Wimbledon, 2017 references therein). Even though important biostratigraphic studies have been carried out in other regions outside of the Mediterranean Tethys, such as that of the proto-Gulf and the Argentinian Andes, the correlation between these regions remains uncertain. Notably, the lack of magnetostratigraphic data in studies from the proto-Gulf (López-Martínez et al., 2013b, 2013a) and the Argentinian Andes (López-Martínez et al., 2017; Vennari et al., 2014) is a challenge and leaves room for ambiguity for biochronostratigraphical correlations. Here we attempt to describe the limitations of the biostratigraphical markers in the studied sections.

In Mazatepec, only two important calcareous nannofossil bioevent is recognized, i.e., the FO of *N. kamptneri minor* and *N. steinmannii steinmanii*. In the Tethys realm, former bioevent occurs within the M19.2n, slightly above the base of the Alpina Subzone (Bakhmutov et al., 2018), and it is used as an upper limit to the base of the Alpina Subzone (Wimbledon et al., 2013). In Mazatepec, the FO of *N. kamptneri minor* occurs 5 m above the base of the Alpina subzone, however, within the lower Ferasini Subzone, thus slightly younger than in the Mediterranean Tethys. Another bioevent in Mazatepec is the FO of the *N. steinmannii steinmannii*, which occurs within the Elliptica Subzone. This marker has been shown, in the past, to occur within the Elliptica Subzone and coincident within the M17r (Casellato, 2010), but has been found as low as the Alpina Subzone, the base of M18r (Bakhmutov et al., 2018; Hoedemaeker et al., 2016; Lukeneder et al., 2010), or even lower (Svobodová and Košťák, 2016). Even though our new calcareous nannofossils from Mazatepec is an addition to the biostratigraphic framework of the sections, it is very preliminary and does not provide any definite constraints for the J/K boundary or the base to the Alpina Subzone in the section. Valuable markers such as *N. steinmannii minor*, *N. wintereri*, *H. strictus* have not yet been reported. Furthermore, no calcareous nannofossils have been reported below the base of the Alpina subzone in Mazatepec. Nevertheless, we feel that the FO of *N. kamptneri minor* so close to the base of the Alpina subzone in Mazatepec boosts confidence for futures studies in the section.

In the Mediterranean Tethys, important markers for the J/K boundary are the FAD of *N. kamptneri minor* and *N. wintereri*. In the Tethys, these two markers usually occur in the middle of the M19.2n, however in distinct stratigraphic horizons and commonly bracketing the base of the Alpina Subzone (Wimbledon, 2017; Wimbledon et al., 2013). *N. wintereri*, for instance, occurs below the base of the Alpina Subzone (Elbra et al., 2018; Svobodová and Košťák, 2016; Wimbledon et al., 2013) and in one occurrence as low as the M19r (Lukeneder et al., 2010). In Las Loicas, on the other hand, both occur virtually within the same stratigraphic range (Vennari et al., 2014). The close FO of *N. kamptneri minor*, *N. wintereri, C. deflandrei*, and *M. pemmatoide* in Las Loicas (Vennari et al 2014) is also troublesome.

The most important secondary marker for the J/K boundary is the FAD of *N. steinmannii minor*, which usually occurs in the vicinity of the Alpina Subzone (Wimbledon, 2017), below (Bakhmutov et al., 2018), and slightly above (Hoedemaeker et al., 2016; Svobodová and Košťák, 2016). In Las Loicas, the FO of *N. steinmannii minor* is present and does occur in the vicinity of the Alpina Subzone, however, only limited to a single sample (Vennari et al., 2014) and not continuous. Furthermore, in Las Loicas the FO of the *N. kamptneri minor* and *N. wintereri* are recorded below the FO of *N. steinmannii minor*. This order of occurrence in Las Loicas is contradicting because the FO of *N. steinmannii minor* is considered older than FO of *N. kamptneri minor* and younger than FO of *N. wintereri*. These circumstances suggest that condensation and/or preservation issues might be affecting the completeness and continuity of the calcareous nannofossil biostratigraphy in Las Loicas and thus impeding a reliable correlation between the Argentinian Andes and the Tethys.

Another possible issue with the biostratigraphy in Las Loicas pertains to a couple of calpionellid assemblages that might seem unusual when compared to the Mediterranean Tethys. First is the presence of *Tintinnopsella remanei* in the upper part of the Crassicollaria Zone. This is a non-typical appearance in the Mediterranean Tethys, but usual in western Tethys as discussed in López-Martínez et al. (2017). Secondly, the record of *Crassicollaria massutiniana* in the lowermost part of the Alpina Subzone. Even when it can be unusual, the presence of this species in the lowermost Berriasian does not affect the biozonation scheme as the Alpina Subzone is defined by the acme of *Calpionella alpina* small and globular form and not the LO of any species. Therefore, the Alpina Subzone is defined in Las Loicas in the same way as in the Mediterranean Tethys and can be used as a reasonable marker for the base of the Berriasian in Las Loicas.

In conclusion, there is still ambiguity in the biostratigraphic framework of the studied sections with regards to the J/K boundary markers. The incompleteness and frequency of key taxa call for further investigation and improvements to the biostratigraphy, and important elements are still lacking for a definite and precise definition of the J/K boundary in both sections and correlations are still troublesome.

**5.2 Constraining the numeric age of the J/K boundary between the studied sections**

In Mazatepec, the middle of the Elliptica Subzone has an age of 140.512±0.031 Ma and consequently a numerical age in the lower Berriasian (Fig. 2 & 4). Conversely, in Las Loicas, the Bchron age-model predicts that approximately the same age, i.e., 140.54±0.37 Ma (ca. 28.5 m, see TS.2) is found in the Crassicollaria Zone, one meter above the FO *R. asper*, and thus Late Tithonian (Fig. 2A). In other words, the age of ~140.5 Ma in one section is coincident with Late Tithonian

fauna, and in the other, it yields an age coincident with Early Berriasian fauna. We see no reason to question the accuracy of the radio-isotopic dates. It becomes thus apparent that both sections are offset in age, and Mazatepec is older than Las Loicas. Therefore, our geochronology points to limitations in biochronostratigraphical correlation of these two sections.

Given the limitations of the biostratigraphy around the J/K boundary in both sections, our ability to quote a single numeric age for the J/K boundary is strongly hindered. Nevertheless, we feel that constraining, bracketing, and/or creating an age confidence interval for J/K boundary using the biostratigraphical and geochronological constraints from both sections is a reasonable alternative to circumventing these limitations. To constrain the interval, we have tentatively chosen upper and lower limits to the interval based on the available biostratigraphic markers and their estimated ages that best bracket the J/K boundary. In Mazatepec, we suggest the FO of *N. kamptneri* as the upper biostratigraphical marker for the J/K age interval. In this section, the FO of *N. kamptneri* is close to the base of the Ferasini Subzone, and thus a subzone normally associated with upper Alpina Subzone (Wimbledon, 2017, and references therein), the base of the 18r (Casellato, 2010), and M19n (Wimbledon et al., 2013), albeit it recently has been shown to be found at the base of the M19.2n (Bakhmutov et al., 2018). We feel that this could be used as a very conservative upper limit of the age of the J/K boundary. Using the sedimentation rate of 2.5 cm/ka in Mazatepec, we estimate the age of the FO of *N. kamptneri* and conceivably the base of the Ferasini Subzone to be ~140.7 Ma (Fig. 2B). This is a conservative estimate for the upper age of the J/K boundary in Mazapetec and could very likely be older since the FO of *N. kamptneri* is commonly older than the base of the Ferasini Subzone (Wimbledon, 2017, and references therein). The base of the Alpina Subzone in Mazatepec is estimated to be ~140.9 Ma, although a bracketing of the Alpina Subzone was not possible due to the absence of calcareous nannofossils commonly occurring at the base of the Alpina Subzone such as *N. steinmannii minor*, or older diagnostic markers such as *R. asper*, *N. erbae,* and *N. globulus*. Therefore, a lower limit to the boundary in Mazatepec cannot be delineated.

Conversely, in Las Loicas, a few Late Tithonian calcareous nannofossils occur in assemblage with Late Tithonian calpionellids such as FO *R. asper,* which is within the upper Crassicolaria Zone, and close to the FO of *U. granulosa* (Bralower et al., 1989; Casellato, 2010). These markers in Las Loicas allow for a lower age limit for the J/K boundary. Given these circumstances we suggest one meter above the FO *R. asper* as the lower limit of the J/K interval in Las Loicas. The Bchron model provides an age of the FO *R. asper* at 140.60±0.4 Ma (ca. ~27 m; see TS.2), which allow a small overlap between the estimated age of the base of the Alpina Subzone in Mazatepec, Late Tithonian and Early Berriasian assemblages in Las Loicas.

In summary, we have attempted to constrain the age of the J/K boundary using the biostratigraphical markers and our geochronology from Las Loicas and Mazatepec. Ash bed MZT-81 (middle of Eliptica Subzone) suggests a minimum age. As a result, the age of the J/K boundary has to be older than 140.512±0.031 Ma, most likely older than ~140.7 Ma (FO of *N. kamptneri* / base of the Ferasini Subzone Fig. 5 (base of the M18r/within M19.2n?), but the latter age estimate derives from an approximate sedimentation rate (2.5 cm/ka) which carries some uncertainty. In Las Loicas, the Bchron model age of the FO *R. asper* (middle of the M19r?) suggests a maximum age for the age of the J/K boundary at 140.60±0.4 Ma. Given that the age of the Alpina subzone in Mazatepec is estimated at ~140.9 Ma, we suggest that the age of the J/K boundary be

bracketed between 140.7 and 141.0 Ma. This interval accounts for the age of the boundary to be slightly older than the base of the Alpina Subzone in Mazatepec due to of the lack of secondary markers below the subzone. Our attempt to constrain the age of the J/K boundary is based only on the diagnostic markers for the boundary reported in the studied sections, and additionally that we can calculate/estimate their ages, even if the chosen upper and lower limit of the interval has been proven to lie distant to the J/K boundary. Given the inherited uncertainties of the biostratigraphy and geochronology, we consider this age bracket as our best estimate for the J/K boundary interval.

### 5.3 The Early Tithonian and the base of the Vaca Muerta Formation

The base of the Vaca Muerta Formation contains Early Tithonian ammonite assemblage of the *Virgatosphinctes andesensis* Zone (Riccardi, 2008, 2015; Vennari, 2016). The gradational contact between the Vaca Muerta and the Tordillo formations is very well exposed in the La Yasera section and contains ash beds very close to the contact (Fig. S2B). We dated ash bed LY-5, and it yielded an age of $147.112 \pm 0.078$ Ma (Fig. 2C). The ash bed is located in the Tordillo Fm, 1.5m below the contact with the Vaca Muerta Formation, thus very close to the base of the *Virgatosphinctes andesensis* Zone. This biozone is mostly equivalent to the Darwini Zone of the Tethys ocean, which is broadly regarded as Early Tithonian and widely distributed in various other regions including Mexico and Tibet (Riccardi, 2008, 2015; see Vennari, 2016 for a thorough review of the subject). Consequently, we suggest the age of ash bed LY-5 ($147.112 \pm 0.078$ Ma) can be regarded as an age in the Early Tithonian. This result is good agreement with other studies that have dated the Early Tithonian. For instance, Malinverno et al. (2012) quote an age $147.95 \pm 1.95$ Ma for the M22An magnetozone, and Muttoni et al. (2018) suggest that the base of the Tethyan Tithonian (top Kimmeridgian) falls in the lower part of M22n with an of ~146.5 Ma.

Assuming the age of the ash bed LY-5 ($147.112 \pm 0.078$ Ma) in La Yesera being Early Tithonian and coupling it with the age for the estimated bracketed age of the J/K boundary (140.7-141. Ma), we can calculate a minimum duration for the Tithonian of ~6-7 Ma (Fig. 2C). This is in good agreement with the current full duration of the Tithonian estimated at ~7 Ma (145.5 to 152.1 Ma, see Ogg et al., 2016). Furthermore, the M-sequence geomagnetic polarity time scale (MHTC12) of Malinverno et al. (2012) suggests a duration for the Tithonian of $5.75 \pm 2.47$ Ma (i.e., between magnetozones M22An and M19n.2n). Therefore, our new ages around the base of Berriasian and close to the Earliest Tithonian are in good agreement of other independent timescale estimates for the duration of the Tithonian. Incidentally, this result also has direct implications for the age of the KmTB. Currently, the age of the KmTB is $152.1 \pm 0.9$ Ma according to the International Commission on Stratigraphy (ICS) (see also Ogg et al., 2016b). Admittedly, the ash bed LY-5 is not at the KmTB, albeit close; therefore, we acknowledge that the age of KmTB would have to be older than bed LY-5. Nevertheless, if the age of the KmTB is 152.1 Ma, it would imply that the *Virgatosphinctes* ammonite Zone itself lasts more than ~5 Ma, resulting in a total duration of ~12 Ma for the Tithonian. It appears reasonable that our results for the Early Tithonian are in agreement with other studies that dated the KmTB, and also suggests that the current ICS KmTB age may need revision.

### 5.4 Implications for the numerical age of the J/K boundary

As of now, the age of the J/K according to the ICS is ~145 Ma, which is ~4 Ma older than our ages around the J/K boundary (Fig. 4 & 5). As we have explored in previous sections, the level of detail of the biostratigraphy in the studied sections needs improvement and fails to provide a precise constraint for the J/K boundary. A significant offset of potentially ~600 ka outlines the limitations of correlating biostratigraphy and geochronology between both sections. Nevertheless, the disparity between our ages presented here and the current age of the J/K boundary is such that even with the biostratigraphical limitations and the absence of magnetostratigraphy calsl for further attention to the numerical age of the J/K boundary. For instance, in Las Loicas the assemblage of *Crassicollaria parvula*, *Crassicollaria colomi* and the FAD of *Umbria granulosa granulosa* in Las Loicas has an age of 141.31 ± 0.56 Ma (Fig. 2A), the FO *R. asper* at 140.60±0.4 Ma, which can be considered to lie within Late Tithonian, and thus constrain the approximate age of the boundary. Furthermore, our age in the Elliptica subzone in Mazatepec is at 140.512 ± 0.031 Ma (Early Berriasian). Worthy of attention is the age of the ash bed LY5 in the *Virgatosphinctes andesensis* biozone (Early Tithonian) at 147.112 ± 0.078 Ma. These geochronological constraints make it fairly difficult to reconcile the base of the Berriasian to be ~145 Ma and also has important implications for the duration of the Tithonian (see discussion above on Early Tithonian). From our new geochronological data, ~145 Ma would be most likely an age in the middle of the Tithonian rather than the base of the Berriasian (Fig. 4). Other recent geochronological studies on the age of the J/K boundary using different geochronological approaches (e.g., Re-Os isochron ages from shales, or LA-ICP-MS U-Pb ages on zircons) and in the Early Cretaceous are also at odds with the current age of the boundary. López-Martínez et al. (2015, 2017), Pálfy et al. (2000a), and Tripathy et al., (2018) have published geochronological results that overlap within uncertainty with our age estimate of the J/K boundary (around 140-141 Ma). In summary, there is growing evidence that the age of the J/K boundary is most likely younger, albeit unequivocal evidence is still lacking.

The endurance of the numerical age of the boundary is mainly due to the perfect overlap between the M-sequence age model of Ogg, (2012) and Mahoney et al. (2005). The latter authors dated a basaltic intrusion in lower Cretaceous (NK1) sedimentary rocks and argued that the age of the basalt would be close to the age of the J/K. Their age for the intruded basalt is 144.2± 2.6 Ma ($^{40}$Ar/$^{39}$Ar). This age was later corrected by Gradstein et al. (2012) and Ogg et al., (2012) to 145.5±0.8 Ma with the recalibrated $^{40}$K decay constant of Renne et al. (2010). The magnetization of drill core 1213B proved to be between anomalies M19 and M20 (Sager, 2005), which was consistent at that time with the working model for the base of the Berriasian placed between M19 and M18 (now more precisely calibrated in the middle of the M19.2n; Wimbledon, 2017). This overlap was also in agreement with the numerical timescale of Gradstein et al. (1995). These facts have mainly been the anchors to the numerical age of the J/K boundary in the past years. However, analytical and biostratigraphical issues potentially reveal some inconsistencies of the numerical age for the boundary in Mahoney et al., (2005). For instance, the biostratigraphy of drill core 1213B poses problems. Bown (2005) pointed out that the sediments of this core are devoid of age-diagnostic NK1 nannofossils such as *Conusphaera* and *Nannoconus*. Important markers such as the family

Cretarhabdaceae are present but in rare occurrences. The drill core 1213B is limited to the occurrences of nannofossils considered secondary markers and lacked any primary markers for the boundary. Even with the existing problems in the biostratigraphy of the drill core 1213B, the magnetization of the dated basalt is in reasonable agreement with the magnetic time scale for the base of the Berriasian (Wimbledon 2017). More importantly, it is worth pointing out that Mahoney et al. (2005) report the dated basalts to be slightly altered, which could have consequences to the accuracy and precision of their age.

The accuracy of the M-sequence age model of Ogg, (2012) is ultimately dependent on the quality of available radio-isotopic ages and cyclostratigraphic data close or around stage boundaries from the Aptian to Oxfordian stages. New geochronological data from stage boundaries from the Late Jurassic to Early Cretaceous suggest that the age of these stage boundaries in this interval could be younger than used in the M-sequence model of Ogg, (2012). For instance, Zhang et al. (2018) provided magnetostratigraphic data to the U-Pb ages of Midtkandal et al. (2016) in the Svalbard cores, which suggests that the age of the M0 (base of the Aptian) is 121-122 Ma, rather than ~126 Ma. Aguirre-Urreta et al., (2015) presented high-precision U-Pb of 127.24±0.03Ma in the Late Hauterivian (close to the base of the Barremian) in the Agrio Fm., Neuquén Basin, which Martinez et al. (2015) used to anchor cyclostratigraphic studies in the in Rio Argo, Spain and calculated an age for the base of the Hauterivian at 131.96±1 Ma and the base of the Berremian at 126.02±1Ma. Aguirre-Urreta et al. (2017) later reported U-Pb high-precision age at the Early Hauterivian at 130.394±0.037 Ma, which is fairly close to the of Martinez et al. (2015) for the base of the Hauterivian. Therefore, new geochronological constraints in the Early Cretaceous render an apparent systematic offset of ~3-4 Ma younger than those used and predicted by the M-sequence age model of Ogg et al. (2012, 2016a). Incidentally, the data we present here for the J/K boundary and close to the KmTB displays the same systematic offset (~3-4Ma) compared to the M-sequence model age of Ogg, (2012); Ogg et al. (2012).

In summary, the M-sequence age model for the Late Jurassic to Early Cretaceous stage boundaries is a creative solution to present numerical ages to stage boundaries with a clear lack of reliable radio-isotopic ages. Nevertheless, recent geochronological developments in the early Cretaceous show that some of the ages used to anchor the model are likely younger than previously accepted. Consequently, future updated versions of the M-sequence model are bound to incorporate these newer age constraints and the critical overlap between the M-sequence model of Ogg. (2012) and Mahoney et al. (2005) for the age of the J/K boundary is likely to change. Being that as it may, reliable radio-isotopic ages for the J/K boundary with high-resolution biostratigraphical markers and magnetostratigraphy in a single section is still lacking, but growing evidence points to a younger age of the J/K boundary as well as other stage boundaries in the Late Jurassic and Early Cretaceous.

## 6. Conclusions

The age of the J/K boundary has been controversial and difficult to determine for the past decades. Our data presented here seem to restrict the J/K boundary from 140.7-140.9 Ma. This interval, nevertheless, carries uncertainty due to

statistical interpolation of the age-depth modeling and estimated sedimentation rate. Our geochronology highlights the problem of using FO and LO of key taxa between the studied sections. We suggest that this might be because (1) different degrees of sample density, (2) insufficient frequency of taxa, (3) preservation of the geological record, and (4) environmental-depositional differences. Nevertheless, our data impose certain constraints for a J/K boundary age at ~145

5    Ma. For instance, the Late Tithonian assemblage of *Crassicollaria parvula*, *Crassicollaria colomi* and the FAD of *Umbria granulosa granulosa* have an age of 141.31 ± 0.56 Ma, The FO *R. asper* s at 140.60±0.4 Ma, and the *Virgatosphinctes andesensis* Zone one at 147.112 ± 0.078 Ma, which calls for a revision of the age of the J/K boundary.

We are unable to precisely define the age of the J/K boundary, mainly because the biostratigraphy does not allow the same temporal resolution as do the used geochronological methods. Nevertheless, it is essential to recognize that the Las

Loicas and Mazatepec sections are unique since they contain datable horizons close or around the J/K boundary. Therefore, our U-Pb dates from these two sections, despite the discussed limitations, provide evidence for a younger numerical age of the J/K boundary.

## 7.    Acknowledgments

This paper is dedicated to the memory of Márcio Pimentel, which unfortunately passed away during the reviewing process of

this manuscript. Márcio was a champion of isotope geochemistry and geochronology in Brazil and played a vital role in the development of personnel and analytical capabilities of the field in Brazil during the '90s and early 2000s. His passing is a great loss to the community, and his presence will be sorely missed. The authors would like to thank reviewers W. Wimbledon, J. Pálfy, B. Galbrun for their valuable input, and especially the Editor S. Gardin for the thorough and careful comments which enormously improved this study. L. Lena would like to thank CAPES (under project 1130-13-7) and the

University of Geneva for financial support. Sam Bowring (MIT) is kindly acknowledged for his support during the initial stages of the project. Adam Curry and Gregor Weber are warmly thanked for their help with the sample from Mexico. This is contribution R-262 of the Instituto de Estudios Andinos Don Pablo Groeber. RLM was funded under grant PAPIIT IA103518.

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

**Figure 1: Location of the studied section, the general geological context of each section.**

**Figure 2: Age correlation between the Las Loicas, Mazatepec, La Yesera section. (A) Las Loicas section: Ash beds in light blue with respective name and U-Pb dates in black font; age-depth modeling ages are in red font next to green stars (this study); ammonites and nannofossils zonation Vennari, et al. (2014); calpionellid zonation Lopez-Martinez et al. (2017);. (B) Mazatepec section: ash bed in light blue with respective name and U-Pb age in black font, age calculated from sedimentation rate red font (this study); calcareous nannofossils (this study); calpionellid zonation Lopez-Martinez et al. (2013). (C) La Yesera section: ash bed in light blue with U-Pb age (Aguirre-Urreta et al., 2014) .**

**Figure 3: A-H. Representative calcareous nannofossils from Mazatepec section, Mexico. A) *Conusphaera mexicana* Trejo (BAFC-NP 4190) [2 m], B) *Conusphaera mexicana* Trejo (BAFC-NP 4196) [11 m], C) *Hexalithus noeliae* Loeblich and Tappan (BAFC-NP 4195) [7.5 m], D) *Hexalithus geometricus* Casellato (BAFC-NP 4205) [25 m], E) *Nannoconus kamptneri minor* Bralower (BAFC-NP 4201) [16 m], F) *Nannoconus globulus* Brönnimann (BAFC-NP 4205) [25 m], G-H) *Nannoconus steinmannii* subsp. *steinmannii* Kamptner (BAFC-NP 4205) [25 m] Our suggestion is to eliminate calcareous nannofossils images published previously from las Loicas section in order to avoid more confusion with taxonomy.**

**Figure 4: U-Pb weighted mean ages of the dated ash beds and the ages and the projected ages of the JKB interval, base of the Calpionella alpina Zone, top of the Crassicolaria Zone, *Virgatosphinctes andesesis* Zone, and the KmTB at ~148 Ma. Color bars represent grains considered in the weighted mean age.**

**Figure 5: Tentative correlation of the studied section with the Mediterranean Tethys correlation scheme of Wimbledon et al. (2017).**

# Figure 1

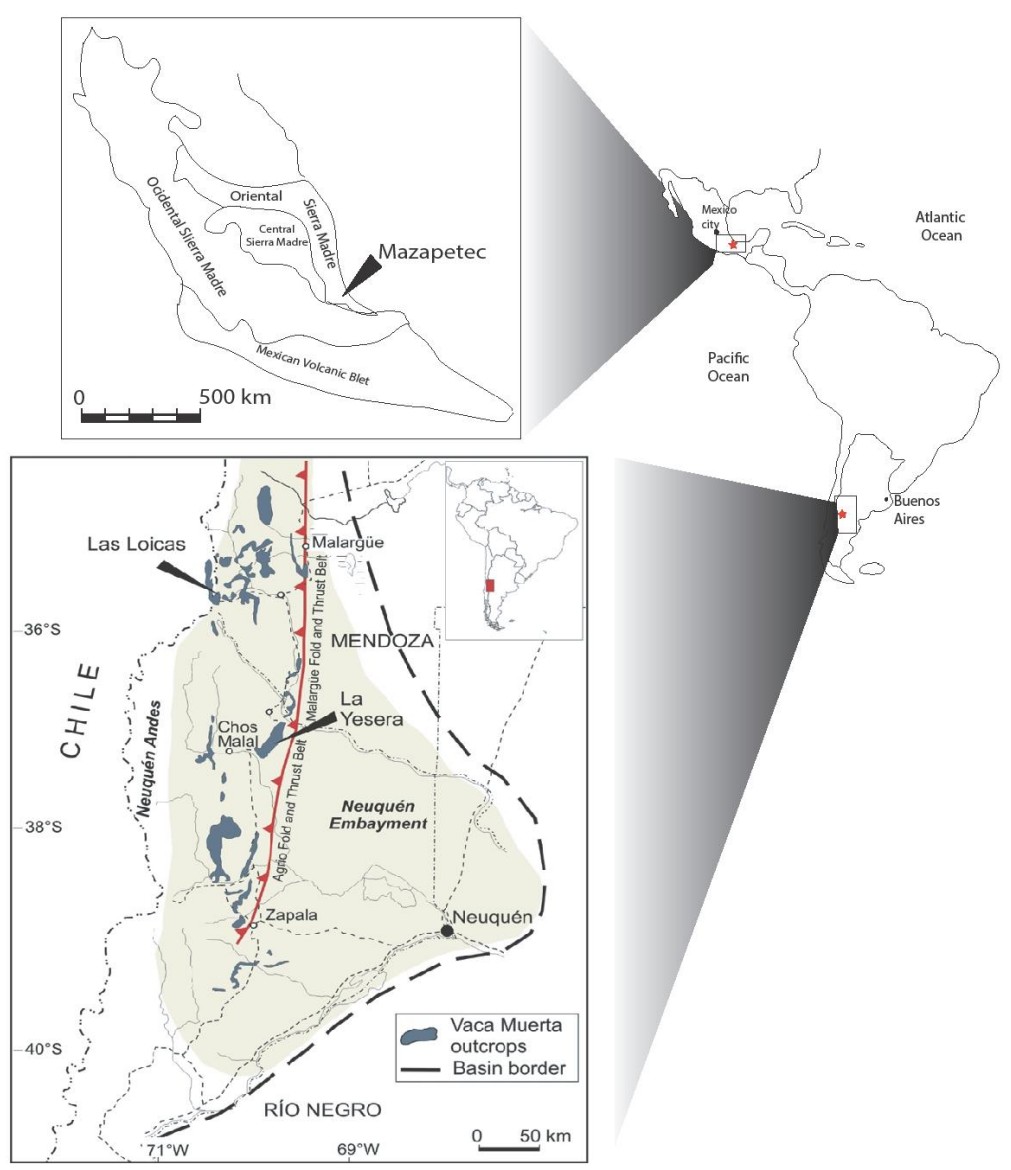

# Figure 2

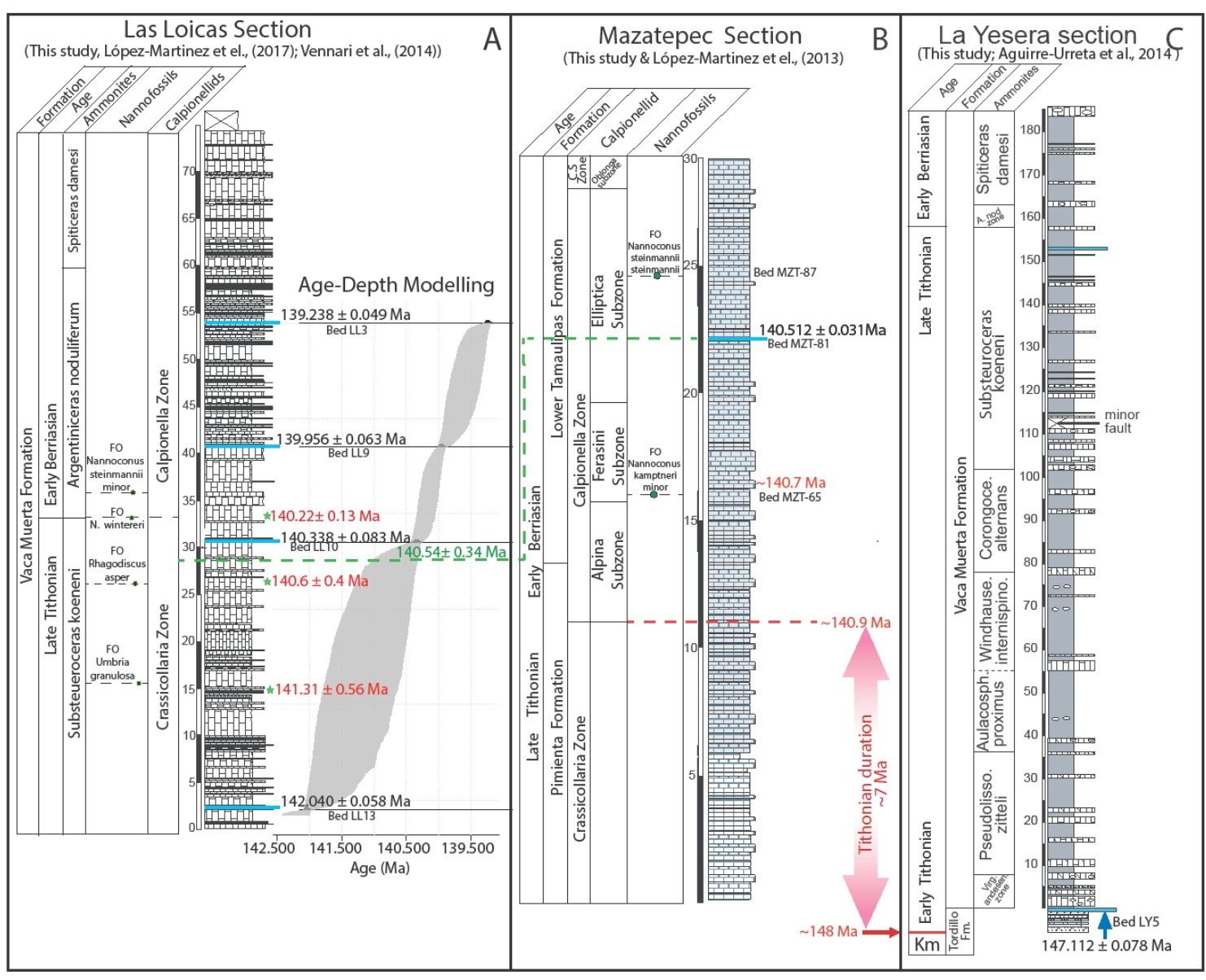

# Figure 3

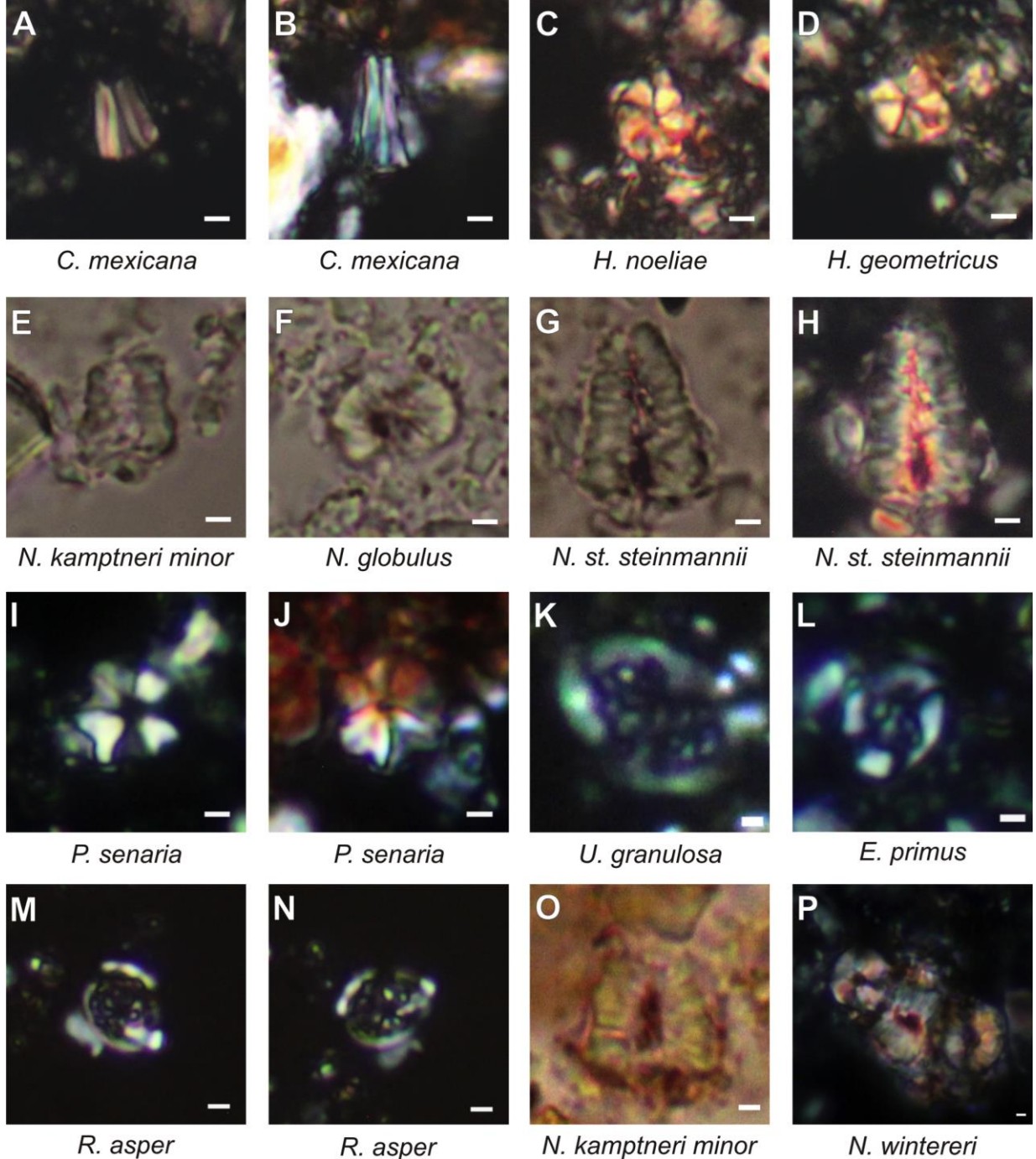

A    *C. mexicana*

B    *C. mexicana*

C    *H. noeliae*

D    *H. geometricus*

E    *N. kamptneri minor*

F    *N. globulus*

G    *N. st. steinmannii*

H    *N. st. steinmannii*

I    *P. senaria*

J    *P. senaria*

K    *U. granulosa*

L    *E. primus*

M    *R. asper*

N    *R. asper*

O    *N. kamptneri minor*

P    *N. wintereri*

# Figure 4

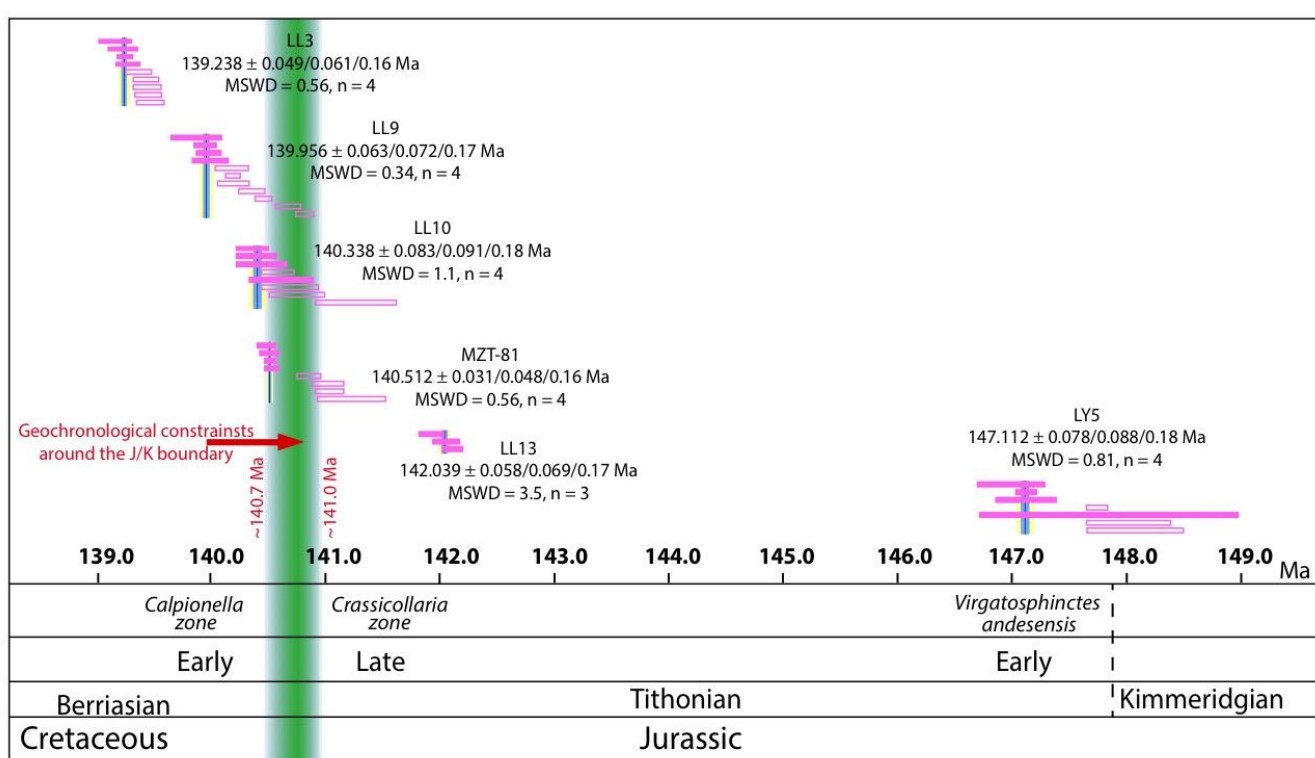

# Figure 5

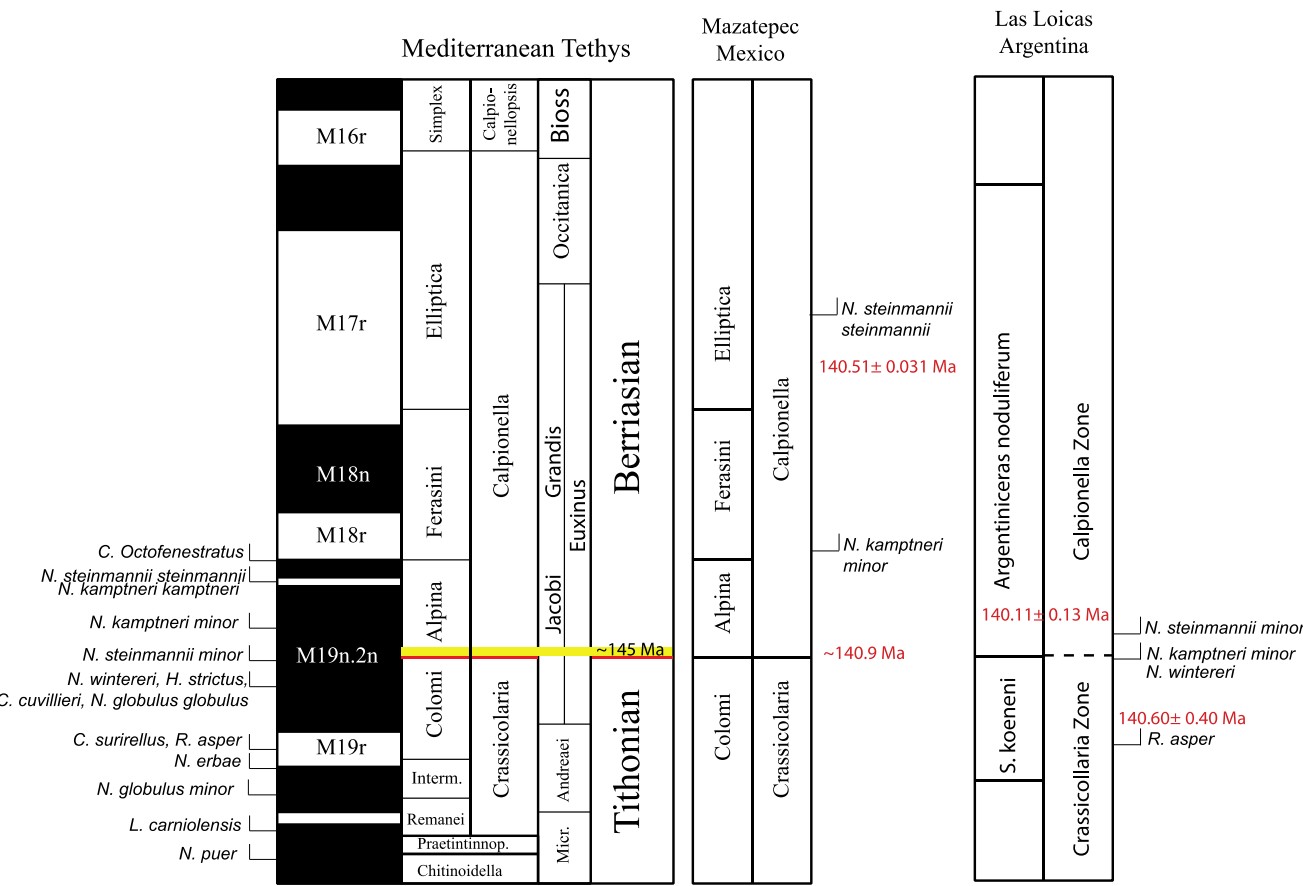