# Peer review of "High-precision U-Pb ages in the Early Tithonian to Early Berriasian and implications for the numerical age of the Jurassic/Cretaceous boundary"

_Solid Earth, 2018_

## Referee Comment (RC1) · W. Wimbledon (Referee) · 27 Jul 2018

See relevant comments on text General stratigraphic remarks Magnetostratigraphy, a key element in J/K definition, is lacking at the two sites described in this typescript. In the Andes, Las Loicas has no magnetostratigraphy, but Arroyo Lonconche does. The text should perhaps say that there is no possibility of magnetic calibration of Las Loicas with the many Tethaan sites where it has been documented; and, further, that the ammonite zonations applied at the LL and AL do not agree – a big problem. The calpionellid assemblage noted at Las Loicas is anomalous: such a mixed assemblage (with apparently derived Tithonain calpionellids) does not define or mark the base of the Berriasian. It should be made clear what is definitively lower Berriasian and what is not. The nannofossil literature cited as the justification for some of the text's discus-

sion and conclusions is rather old - Bralower and Casellato references are now 10 -30 years old. Many Tethyan sites have since been documented, and that make some of the species FADS and the zones discussed obsolete. Some Italian localities cited in the text are seen as anomalous in the positions of their nannofossil FADs. Thus it is not clear why these localities are selected by the authors for comparison with the LL and M sites, especially when they are not the best/most representative. The dating of the magnetozones needs to highlighted and discussed at more length in the Discussion. Also, the assumptions (as seen in most publications) about using the magnetostrati-graphic scale as a time scale could be laid out fully in the Introduction. It would help the reader if the authors could distinguish between ICS chronostratigraphic charts and others. Notably, Ogg et al 2016 is not at all 'official' and is not attributable to ICS, but this is not clear from the text. Structure Some of the material now in the Discussion should appear in the Introduction. The chronostratigraphic and biostratigraphic background should be made clear before consideration of any new data on radiometric dates. [As an aside, chronostratigraphy at the J/K boundary (or any other boundary) is not reliant on geochronology.] Correlations in the J/K interval has been advanced in recent years by the combination of magnetostratigraphy and calpionellid biostratig-raphy, plus ammonites and nannofossils. The extrapolation of the GMPTS to onshore localities is central. The paper is concerned with attaching radiometric dates to a bios-tratigraphic framework. But it says very little about how radiometric dates match the timescale used by, for instance, Gradstein et al 2012: a time framework linked to the oceanic magnetostratigraphic record, the GMPTS. The last is hardly mentioned. This imbalance should be corrected. The core of the paper could usefully be a careful examination of the calibration of the dated ash horizons and the levels with the key biostratigraphic markers – listing them in sequence, level by level. At present, the text does not give such a clear description. Precision, accuracy, meaning, English language There are numerous examples of rather problematic phrases and sentences which are not written in good English. But more critical is the lack of precision or looseness in language and terminology. This lets down the submission very badly. It is the thing that

needs most attention in a revision by the authors. The factual content of the paper is currently often undermined casual wording, and lack of focus on the essential elements in the data, or rather feeble or vague discussion of the facts (numerous markings on the text). Attention to biostratigraphic/chronostatigraphic accuracy are recommended for a revision: the text would then do justice to the new data being presented The loose wording of the Abstract's and Introduction's first sentences. No, the age of the J/K boundary is very clear. Lena et al. talk only about radiometric dating. They should say that the start of Berriasian age/base of Berriasian stage has been more or less fixed for some years [the authors actually quote several relevant papers that show this]. The typescript describes 'absolute' dates that are useful in constraining the boundary, or at least the boundary interval: it is such radiometric dates that have been lacking. Thus, the chronostratigraphy is clear, but sound geochronology is new. Repetitions could be removed. Every time a site is mentioned it is always "in the xx section". At A and at B would be welcome change for the reader. There are lots of alternative words to section: outcrop, exposure, profile,. . . . . . "JKB" is not standard terminology. It appears hundreds of times in the text. "J/K boundary" is the norm. Alternatives for use are: the base of the Alpina Subzone, base of Berriasian Stage, Tithonian/Berriasian boundary, or, less precisely, the J/K interval, the boundary interval. . . . . . Care is required is using the phrase J/K boundary. Anything that is not exactly correlated with the base of the Alpina Subzone can be said to be in the J/K interval, but not at the boundary. The reader is sometimes not sure what interval is referred to, or what horizon. Many times a fossil or date is somewhere in the J/K interval, but, to be accurate, nowhere near the actual boundary. Frequently, fossil names are incorrectly spelt. Also - palaeontology, metre. . . . . . Verb is to "crop out" not to "outcrop".

Conclusion I recommend this text for publication after substantial improvement. The underlying data is sound, but the structure of the account and, even more, the prose are in need of work. The description and the discussion do not do justice to the factual content.

Please also note the supplement to this comment:
https://www.solid-earth-discuss.net/se-2018-57/se-2018-57-RC1-supplement.pdf

[Figure]

**Supplement:**

[revised manuscript text omitted]

**4. Results and discussion**

**4.1 The age of the Jurassic/Cretaceous Boundary in the Vaca Muerta Formation**

The Las Loicas section ...nites and calcareous nannofossils (Vennari et al., 2014) as well as calpionellids (López-Martínez et al., 2017). In Fig. 4A the various primary marker assemblages and the age of the dated ash beds found in the Las Loicas section ...icated. The late Tithonian Crassicollaria Zone, Colomi Subzone (Upper Tithonian) is composed of *Calpionella alpina* Lorenz, *Crassicollaria colomi* Doben, *Crassicollaria parvula* Remane, Crassicollaria massutiniana (Colom), *Crassicollaria brevis* Remane, *Tintinnopsella remanei* (Borza) and *Tintinnopsella carpathica* (Murgeanu and Filipescu) (López-Martínez et al., 2013b, 2013a, 2015). This calpionellid assemblage occurs below the base of the NJK-B calcareous nannofossil Zone, characterized by the FAD of *Umbria granulosasa granulosa* (Bralower et al., 1989) and well within the *Substeueroceras koeneni* ammonite Zone (Vennari et al., 2014). All these markers have been considered late Tithonian in age (Bralower et al., 1989; Casellato, 2010; Riccardi, 2015). More importantly, the occurrence of *Crassicollaria parvula* and *Crassicollaria colomi* and the FAD of *Umbria granulosasa granulosa* are located 13 meters above ash bed LL13, which has an age of 142.040 ± 0.058 Ma. Since the assemblage is situated 13 meters above from the dated ash bed (ca. 15 m stratigraphic height), the ...on model age is 141.31 ± 0.56 Ma (Fig. 4A). Therefore, this age can be considered a minimum age for the late Tithonian based on the association of *Crassicollaria parvula* and *Crassicollaria colomi* in close occurrence with the FAD of *Umbria granulosasa granulosa*.

In the Las Loicas section, there are several well-known early Berriasian markers. For instance, the FAD of *Nannoconus kamptneri minor* (Fig.SA) and *Nannoconus steinmannii minor* are considered trustworthy indicators of the early Berriasian (Bralower et al., 19... where they overlap with the base of the *Argenticeras noduliferum* ammonite Zone (López-Martí... ...ri et al., 2014). The occurrence of the calpionellid assemblage dominated by *Calpionella alp*... ...ns of *Crassicollaria massutiniana, Tintinnopsella remanei,* and *T. carpathica* confirms the early B... ...rtínez et al., 2017a) (Fig. 4A)... ...by ash beds LL9 (139.956 ± 0.063... ...± 0.083 Ma) (Fig.SA). From... ...e of the Berriasian cannot be young... ...Ma, because ash bed LL9 is located 8 meters above the base of the *Argenticeras noduliferum* Zone. The early Berriasian calpionellid assemblage described in López-Martínez et al. (2017) overlaps with the FAD of *Nannoconus kampteri minor* (Fig. SA) and *Nannoconus steinmannii minor* and the base of *Argenticeras noduliferum* ammonite Zone (c.a 34 m stratigraphic height) (Fig. 3A). Using age-depth modeling, we ca... ...13Ma (Fig. 4A).

When calibrating the age of stage boundaries, magnetochrons are extremely important because they impose a single work frame ... normalized against. The use of magnetostratigraphy coupled with biostratigraphy has become a crucial tool for successfully correlating different JKB sections. Currently, in various sections that span the JKB, ... base of the Calpionella Zone is, in many cases, appears to be coincident with the M19n.2n

(Schnabl et al., 2015; Wimbledon, 2017). Therefore, the magnetochron M19n.2n has lately emerged as a reliable tool in [annotation: magnetozone] [annotation: If you call 1980s onwards lately?]

[revised manuscript text omitted]

biozones in the ... 2010). Unfortunately, the presence of *N. steinmanni minor* or *N.*

*wintereri* (Wim ... azatepec section. However, it is reasonable to assume that both

[annotation box: This does not match evidence from lots of sites. N. steinmannii steinmannii is not a marker for the Elliptica Subzone, especially when it occurs as low as the Alpina Subzone. You quote Wimbledon 2017?]

of these markers would be close to the base of the Alpina Zone since the FAD *N. steinmanni* is only 5 m above the base of the Alpina Zone. Therefore, the relative age of the palaeontological markers in the Mazatepec section is in full agreement with the working model of Wimbledon (2017) for the JKB.

To constrain the age of the JKB in the Mazatepec section, we have dated the ash bed in bed 81 which is located within the Elliptica Subzone and stratigraphically 10.1m above the base of the Alpina Subzone (Bed MTZ-45 Fig. SC), i.e., JKB (López-Martínez et al., 2013b) (Fig. 4B). The age of ash bed MZT-81 is 140.512 ± 0.036Ma (Fig.2). Unfortunately, in the Mazatepec section ash beds are scarce. Therefore, it was not possible to bracket the age of the JKB, as was the case in the Las Loicas section. Consequently, to estimate the age of the boundary, we have to resort to assumed sedimentation rates to back-calculate the age of the JKB. Since the sedimentation rate in the Pimienta and Tampaulipas formations is unknown, we use both high and low sedimentation rate because this takes into account our conjectural knowledge of the sedimentation rate in the Pimienta and Tampaulipas formations. Here we assume a low sedimentation rate to be 2.5 cm/ka and a high sedimentation rate to be 4.5 cm/ka. Therefore, the age of the JKB is estimated to be 140.7 Ma and 140.9 Ma, respectively.

**4.3 The early Tithonian and the base of the Vaca Muerta Formation**

The base of the Vaca Muerta Formation contains a well-established early Tithonian ammonite assemblage of the *Virgatosphinctes andesensis* Zone (Riccardi, 2008, 2015; Vennari, 2016). Fortunately, the gradational contact between the Vaca Muerta and the Tordillo formations is very well exposed in the La Yasera section and contains ash beds very close to the contact (Fig. SB). We have dated an ash bed (LY-5) located ??????????contact and it yielded an age of 147.112 ± 0.078 Ma (Fig. 4C). The ash bed is located in the Tordillo Fm, 1.5m below the contact with the Vaca Muerta Formation, thus very close to the *Virgatosphinctes andesensis* Zone depending on the nature of the contact he Darwini Zone Tethys was an ocean not a 
[revised manuscript text omitted]
 nannofossils considered boundary markers (Wimbledon, 2017) and lack/lacks primary markers. These facts collectively render the section biostratigraphically [...]ds to the JKB markers. In closing, we feel that the results

presented in this study are in good agreement with several other studies of the age of the JKB and thus it allows our bracketed interval to be considered as the age of the JKB globally.

*[comment: As a concluding sentence it is not effective. It says, more or less, our age agrees with other ages. Not a very weighty ending]*

**5.** *[comment: Cretaceous rock/time is base Berriasian stage and start Berriasian age. What you discuss is geochronology and radiometric dates]*

[revised manuscript text omitted]

JKB as in the text. J/K boundary or Tithonian/ Berriasian boundary
Species names should not have a calital letter

---

## Referee Comment (RC2) · J. Pálfy (Referee) · 31 Jul 2018

General comments

The paper by Lena et al. reports six new U-Pb ages from ash beds around the biostratigraphically constrained Jurassic-Cretaceous boundary (JKB) from three sections in Argentina and Mexico and uses them to argue that the age of the system boundary is 140 Ma, more than 5 m.y. younger than in the most widely used recent geological time scales. This contribution is significant and timely, as the calibration of the JKB has remained uncertain due to a dearth of reliable numeric ages around it. As a related but different issue (unfortunately not clearly separated in the paper), this is also a 'hot topic' as the JKB is the last Phanerozoic system boundary without an accepted GSSP

definition.

The main strength of the paper is providing most welcome new U-Pb dates from the JKB interval and placing them in a multiple biostratigraphic framework. The dates are of high analytical quality and precision, obtained using cutting edge CA-ID-TIMS methods. However, I take several issues with the interpretation, and may suggest guidance for a revised version which could better avoid the pitfalls of confusing regional and global biostratigraphic correlation issues. Instead, a re-focused discussion should emphasize the obvious significance of the radioisotopic dates in highlighting problems and contradictions in biostratigraphy. Additional recommendations will lead to improvements in the structure of the paper, its language, and the presentation of figures. I recommend publication after revision.

Specific comments

The following points and issues require attention during the revision (listed largely in the order of appearing in the original version of the Discussion Paper).

The paper needs a proper "Geological and stratigraphic setting" chapter to augment and replace the "Studied areas" in the current version. Formation names, i.e. the bare bone lithostratigraphy should be complemented with brief characterization of basin evolution and depositional environments, to provide context for assessment of stratigraphic completeness and sedimentation rates in the section, the latter being crucial in the authors' arguments in comparing the JKB age of different sections. Care should be taken to ensure consistency in terminology and usage of biozones.

Much biostratigraphic information is presented both in the text and in Fig. 4. However, it is not clear to the reader what, if any of these is new here, what is taken unchanged from the references cited, and what is revised from published sources. Cases where there is controversy in either the zonal subdivision of sections or their correlation, based on ammonoids, calpionellids and ammonoids (e.g. between Riccardi 2015 and Vennari et al. 2014), and the stance of the authors should be more clearly stated. The

reader might suspect that calcareous nannofossil occurrences are newly obtained as Supplementary Fig. 3 is promised to present them (p. 3, l. 26), but this figure is missing.

Details of reporting of the error and age interpretation would be better placed in the main text's Methods chapter rather than in the Supplementary Material. For the aimed global relevance in time scale studies, the most conservative error (i.e. that including the tracer calibration and decay constant errors) needs to be quoted and used for each U-Pb dates throughout the paper. This is typically still within $\pm 0.2$ Ma, a commendable high precision.

The chapter "Results and discussion" needs to be split into two, allowing results to be clearly separated from the interpretation.

Even though it is widely accepted that magnetostratigraphy is very useful for global correlation in the JKB interval, projecting the magnetozones identified in the Arroyo Loncoche section in the Neuquén Basin (Iglesia Llanos et al. 2017) introduces additional confusion (p. 5, l. 1-9, Fig. 4) to the already complex web of stratigraphic correlation of the three studied sections. The new results from Las Loicas do not appear to be closely correlatable with Arroyo Loncoche, Fig. 4 reveals that the placement of the JKB is offset by nearly one ammonoid zone, being near the base or at the top of the Substeueroceras koeneni zone, respectively. It would suffice to say that magnetostratigraphy of the Las Loicas section will be desirable to enhance the utility of the newly obtained U-Pb ages and clarify contentious biostratigraphic correlation issues.

Discussion on the age of the JKB in the Mazatepec section includes an assumption on the FAD of a nannofossil taxon, Nannoconus steinmanni minor, not actually found in the section (p. 5, l. 31 – p. 6, l. 3). Such speculation is best avoided.

Beware of the lack of formal definition of base Tithonian. There is no agreed-upon GSSP decision yet, contrary to what is implied here (p. 6, l. 25). The attendant uncertainties of stage boundary placement and its correlation with the Andean sections

make the time scale calibration use of La Yasera U-Pb date more problematic than admitted here.

The discussion on the duration of the Tithonian is interesting but contains a factual error and misses some further opportunities. The Geological Time Scale 2016 (Ogg et al. 2016) is misquoted, it assigns 150.8 Ma to the base of Tithonian Stage and 145.5 Ma to the JKB, hence the duration of the Tithonian is 5.3 m.y., less in agreement with the 6.9 m.y. arrived at here. It would be useful to compare two other, independent duration estimates. The Pacific M sequence of magnetic anomalies has long featured in time scale calibration. The recent work of Malinverno et al. (2012) (the MHTC12 scale) suggests 6 m.y. for the Tithonian, i.e. between magnetochrons M22An and M19n2n. The cyclostratigraphic analysis of Kietzmann et al. (2015; not cited by Lena et al.) identifies 10 long eccentricity cycles for almost the entire Tithonian, starting with the Virgatosphinctes mendozanus zone dated here at La Yesera, hence a duration of c. 4 m.y. The discussion should emphasize that the duration favored here is longer these previous estimates using other methods and offer possible reasons to explain the difference, perhaps considering biostratigraphic correlation issues.

Perhaps my most important criticism and suggestion pertains to the projection of a sedimentation rate-based JKB from the Mexican Mazatepec section into Las Loicas in Argentina. The authors can make a much stronger case and build a more logical argument by projecting the actual U-Pb date, expressing the stratigraphic height from the age-model calculation as ∼28.5 m and note the mismatch in biostratigraphies. Reading from Fig. 4, beds of the same numeric age thus appear assigned to nanno-fossil zone NJK-B vs. high in NJK-D, to calpionellid Crassicollaria zone vs. Calpionella zone (and its third subzone, the Elliptica subzone, and ultimately to lower Berriasian vs. upper Tithonian at Las Loicas and at Mazatepec, respectively. The discussion could thus be refocused to use the newly obtained high-precision and high-resolution U-Pb age framework to highlight biostratigraphic correlation issues, most likely due to diachronous FAD-LADs of certain key taxa.

To strengthen the argument for potential problems in biostratigraphic correlation, the authors might comment on the discrepancy of ammonoid-based correlation, and striking differences of thickness of zones in different sections even within the Vaca Muerta Fm. (e.g. Argenticeras noduliferum zone: ∼27 m in Las Loicas vs. 5 m in La Yesera section).

It the "Global correlation" chapter, the suggestion of understanding the JKB as an interval (p. 8, l. 1-10) is conceptually flawed and needs to be rephrased. By definition, the JKB boundary (as any other chronostratigraphic boundary) is a time line. It does indeed carry an uncertainty of our numeric calibration but it cannot be equated with an actual time interval in which different "boundary events" took place.

Also in this final chapter, consider the significance of your argument for a significantly younger JKB together with Martinez et al. (2015) suggested age for the base Valanginian at 137 Ma. This would make for a shorter than previously understood Berriasian Stage of a ∼3 m.y. duration. This in turn contradicts with the astrochronology of Kietzmann et al. (2015), who identify more than 10 long eccentricity cycles in the Berriasian part of the Vaca Muerta Fm.

The statement in chapter "6. Data availability" suggests that some of the raw data will be withheld until completion of the thesis of the first author. Instead, all data should be made available at the publication of this paper. Understandable practice is not to release data in a thesis prior to publication, but there should be no reason to justify an embargo the other way around.

Fig. 1 is of inferior quality, on a base map published nearly 25 years ago, unnecessarily carrying a title and showing non-American sections not mentioned in the text.

Table S1 contains the essential data for the U-Pb geochronology, it should be placed in the main part of the paper. Fig. S is also worth transferring from the Supplementary Material to the main part. (However, its labeling needs re-coloring so it be legible in black and white print, panel C might be more informative to show the dated ash bed, D

needs labels, and the figure needs a caption.)

Additional reference cited above (not used in the original version): Kietzmann, D.A., Palma, R.M. and Iglesia Llanos, M.P., 2015. Cyclostratigraphy of an orbitally-driven Tithonian–Valanginian carbonate ramp succession, Southern Mendoza, Argentina: Implications for the Jurassic–Cretaceous boundary in the Neuquén Basin. Sedimentary Geology, 315: 29-46.

Technical corrections

The comments below also include several suggestions for better English language, style and word choice.

p. 1, l. 12 (and elsewhere): age → numeric age

elusive → difficult to determine

l. 16: display → contain

l. 21: one of the last major Phanerozoic stage boundaries → last Phanerozoic system boundary

l. 23: absolute → [delete, avoid "absolute age" altogether]

p. 2, l. 3: Calpionella alpina subzone (cf. l. 16) [ensure consistency in zonal names and terminology]

l. 17: selected → suggested

l. 21: Kamptneri → kamptneri

p. 3, l. 6: spans → exposes

l. 12: out of sequence numbering of figures (not as they appear in text)

l. 26: optical images → photomicrographs

p. 4, l. 3: The section → The Las Loicas section

l. 29: impose → may provide

p. 5, l. 2, 6: fossil density → abundance of fossils

p. 6, l 24: Tithonian

p. 9, l. 22: thank

p. 10, l. 3: Neuquén

p. 11, l. 11: [delete] February

p. 12, l. 6: Potosí [+spell out journal name]

l. 13: & [delete]

p. 14, l. 4: Episodes [delete the rest of name]

l. 14: Aguirre-Urreta

l. 22: [provide doi instead of URL]

References cited in text but not listed in reference list: Edwards, 1963 R Core Team, 2013

p. 15, l. 3: Distribution of continents → Global paleogeography

l. 9-15 (Fig. 3): Give stratigraphic horizon of occurrence (e.g. m from base) fo each specimen photographed

p. 16, Fig. 1: delete title, consider using different base map, do not show migrazion routes and sections not discussed in text

p. 17, Fig. 2: Barriasian → Berriasian

[J/K boundary interval – see comments about conceptual flaw here]

[fonts too small in the upper part, too large in the lower part]

[Figure]

p. 18, Fig. 3: [it is redundant to show taxon names here, it is customary to give them in the caption only]

p. 19, Fig. 4: [this is the key figure of the paper, already need to refer to in the Geological setting, so make it Fig. 2; A: show meters; put Las Loicas section to a separate panel B, making the others C and D; La Yesera: indicate placement of JKB; some lettering uses illegibly small font]

Supplementary Material

p. 1, l. 1: Ash beds were crushed → Samples were crushed

p. 3, part 5

Give weight of each sample so zircon yield can be assessed in this context.

Grains discarded as too old are erroneously quoted as >∼150 Ma for each sample, provide true cut-off age of grains not included in age calculation.

5.3 (p. 4): Ash bed LL10 has n=6 grains in Fig. 2, four in text

5.4. Ash bed LL13: include date of discarded grains in Table S1 (really older than 450 Ma?)

5.5. "Due to its proximity to the Tordillo Fm." [it is from the Tordillo Fm.]

inherited grains or detrital grains?

5.6. MZT-81 (p. 5): check this descriptions, there are errors here. four discarded grains (not five), the grain numbers are in error (belong to sample LL10)

Fig. S needs a caption and should be transferred to the main part of the paper. The labels of the figures need to be recolored so they be legible in black and white print as well.

Table TS.1 is essential to assess the U-Pb dates reported so it should be transferred to the main part of the paper

Sample LY5 in Table TS.1: why discard grain z67 and keep z10, when the first one is not older and its error is not larger? This and similar issues of only marginally different-aged grains undermine the credibility of unbiased and rigorous selection of grains for the age interpretation.

8.2 (p. 11), Table TS.2: Why is the age value of 2 m any different from the age of LL13 taken from this level?

---

## Author Comment (AC1) · 16 Aug 2018

Reply to comments by reviewer J. Pálfy (reviewer #2) on the manuscript "Cross-continental age calibration of the Jurassic/Cretaceous boundary"

We would like to thank J. Pálfy for accepting to review our manuscript kindly. His comments were incredibly insightful and will undoubtedly improve the quality of this manuscript.

Comments by the reviewer have been copied and pasted in the italic blue font, and the answers are found immediately below in regular black font. The comments are in order of appearance in the reviewer's comments. We have taken the liberty of numbering the comments from 2.1 through 2.20. Subsequently, replies to technical comments are in order of appearance in the reviewer's comments, also in the same fashion as the generals comments. References are found here right after the "technical corrections" section.

Please also note the supplement to this comment:
https://www.solid-earth-discuss.net/se-2018-57/se-2018-57-AC1-supplement.pdf

─────────────────────────

**Supplement:**

**Reply to comments by reviewer J. Pálfy (reviewer #2) on the manuscript "Cross-**
**continental age calibration of the Jurassic/Cretaceous boundary"**

Comments by the reviewer have been copied and pasted in the *italic* blue font, and the
answers are found immediately below in regular black font. The comments are in order of
appearance in the reviewer's comments. We have taken the liberty of numbering the comments
from 2.1 through 2.20. Subsequently, replies to technical comments are in order of appearance in
the reviewer's comments, also in the same fashion as the generals comments. References are found
here right after the "technical corrections" section.

We would like to thank J. Pálfy for accepting to review our manuscript kindly. His
comments were incredibly insightful and will undoubtedly improve the quality of this manuscript.

Firstly, we appreciate the reviewer's enthusiasm towards our manuscript and also like to
say that we share the reviewer's opinion that the JKB is a "hot topic" and we acknowledge that we
should have made this point in the manuscript. As such, we feel that hot topics are better
communicated shortly and concisely, in a way that is more appealing to the general public. We
believe that other Phanerozoic system boundaries have overshadowed the JKB. In this sense, it is
essential to produce publications that are easily readable by the general public without too much
unnecessary information, by removing excessive description and revision of readily available
information that could otherwise be easily accessed in previous publications. Additionally, concise
writing allows for a faster turnaround time during the reviewing process, which is a big attraction
for submission. Therefore, we would like to answer a few of the reviewer's comments that directly
related to the above statement.

**General comments**
*2.1)    "… I take several issues with the interpretation, and may suggest guidance for a revised*
*version which could better avoid the pitfalls of confusing regional and global*
*biostratigraphic correlation issues. Instead, a refocused discussion should emphasize the*

*obvious significance of the radioisotopic dates in highlighting problems and contradictions*

*in biostratigraphy."*

This is a significant point, and it highlights the importance of dating the stratigraphic record using high-precision geochronology to unravel its subtle nuances. If we have interpreted the reviewer's advice correctly, our ages clearly show that assuming time-equivalency of biostratigraphic zones can lead to erroneous correlations regarding the numerical ages of FAD and

LOD. Possibly, this difference can arise from the migratory rates of these species resulting in the diachroneity of FDA and LOD. This is an interesting point to explore and discuss in the revised manuscript and will be incorporated into the revised version. Nevertheless, we feel that the essential aspect of our data is how younger the age of the JKB is with regards to the long-lasting age of 145 Ma. This is the most crucial contribution of the manuscript, and the discussion around how the age of the JKB in both sections favorably agree is still central to the manuscript.

**Specific comments**

*2.2)    The paper needs a proper "Geological and stratigraphic setting" chapter to augment and*

*replace the "Studied areas" in the current version. Formation names, i.e. the bare bone*

*lithostratigraphy should be complemented with brief characterization of basin evolution*

*and depositional environments, to provide context for assessment of stratigraphic*

*completeness and sedimentation rates in the section, the latter being crucial in the authors'*

*arguments in comparing the JKB age of different sections.*

In the "Studies areas" chapter, we chose simply to give a brief description of where the studies sections are located and cite important publications relevant to where the sections are exposed. There are numerous publications on the tectonic architecture and basinal evolution where the sections are that are cited in the manuscript. As it stands, the manuscript is 4626 words long, which we feel is an adequate length for a publication. If we were to expand the "Studies areas"

chapter with a detailed "Geological and stratigraphic setting" chapter, it would increase the manuscript to another 800-1000 words. Even then, it would not do justice to fully review the geological setting of both geological settings within 1000 words (e.g., 500 words each basin). The reviewer claims that such an expansion of the regional geology would be useful to understand better the sedimentation rate in Mazatepec, which is an integral part of our discussion. However, we make it pretty clear in the manuscript that the sedimentation rate in the Mazatepec section is

**unknown**, and we further use both a low and high sedimentation rate to back-calculate the age of the JKB in the section. Even with a thorough knowledge of the sedimentological and stratigraphical background, there is no hard evidence for the rate of sedimentation rate in the

Pimienta and Tamaulipas formations. Ultimately, this would inevitably leave us with a subjective choice of sedimentation rate based on the depositional environment and sedimentological structures present. Moreover, we also make the case that the choice of sedimentation rate is not that important. Nevertheless, we would not oppose slightly expanding the "Studies areas" chapter, or giving it a new title if the reviewer feels adamant about the subject. We leave this option to the discretion of the Handling Editor, because it influences the format with which publications in Solid

Earth are communicated.

*2.3)*     *Care should be taken to ensure consistency in terminology and usage of biozones. Much*

    *biostratigraphic information is presented both in the text and in Fig. 4. However, it is not*

    *clear to the reader what, if any of these is new here, what is taken unchanged from the*

    *references cited, and what is revised from published sources*

    In the caption for figure 4, there is ample information on the information that is new and what is cited from other publications. We will try to make the figure 4 clearer at the request of the reviewer well as its caption.

*2.4)*     *Cases where there is controversy in either the zonal subdivision of sections or their*

    *correlation, based on ammonoids, calpionellids and nannofossils (e.g., between Riccardi*

    *2015 and Vennari et al. 2014) and the stance of the authors should be more clearly stated.*

Ammonoids: There is no discrepancy among the biozonation of Riccardi, (2015), the Vennari et al., (2014), and the present manuscript regarding the sequence and names of index species of each biozone. It is worth to mention here that Riccardi explicitly states: "*There is no attempt to deal*

*here with the precise definition of the Jurassic-Cretaceous limit, and therefore the use of terms*

*such as "Tithonian," "Berriasian," "Upper/Late Jurassic" and "Lower/*

*Early Cretaceous have been kept to a minimum and is usually adopted when quoting other sources.*

*It is considered that once biostratigraphic correlations are well-established definition of Stage*

*and System boundaries will follow by convention*" (Riccardi 2015, p. 24).

Calpionellids: The data from this manuscript has been published by López-Martínez et al., (2013)

for the Mexican section and López-Martínez et al., (2017) for the Argentine section.

Nannofossils: The data from this manuscript has been published by Vennari et al. (2014) for the

Argentine section. The data presented here for the Mexican section is new, and a systematic paper is in preparation (Lescano et al. in prep.).

*2.5)    The reader might suspect that calcareous nannofossil occurrences are newly obtained as*

  *Supplementary Fig. 3 is promised to present them (p. 3, l. 26), but this figure is missing.*

  Yes. Unfortunately, we have not placed the Supplementary Figure 3 (distribution chart for the calcareous nannofossil species) in the Supplementary Materials as stated in p.3, l. 26. We apologize and promise to rectify.

*2.6)    Details of reporting of the error and age interpretation would be better placed in the main*

 *text's Methods chapter rather than in the Supplementary Material.*

 The detailed account of the geochronological data is intended for full disclosure of its meaning and interpretation; however, this would only be appealing to a specific subset of the geochronology community. The average reader, drawn by the interest of knowing the age of the

JKB, in our opinion, would be distracted by an excessively detailed description of the geochronological U-Pb data in the main text. Moreover, this information is not further referred nor directly used in the discussion and conclusion chapters, i.e., the meaning of a depositional age for the ash beds, number of grains selected for weighted means, etc. These are not information that is
central to the discussion of the data and conclusions. This is why we decided to keep it in the
Supplementary Materials.  Nevertheless, we leave it at the discretion of the Handling Editor do
choose what best fits the format of the journal because it would be an easy adjustment to make to
the revised manuscript.

*2.7)    For the aimed global relevance in time scale studies, the most conservative error (i.e., that*
*including the tracer calibration and decay constant errors) needs to be quoted and used*
*for each U-Pb dates throughout the paper. This is typically still within 0.2 Ma, a*
*commendable high-precision.*

The reason high-precision ages are reported with three errors (as explained in the
Supplementary Materials) is to allow for an appropriate propagation of errors when comparing
different geochronological datasets that been acquired through different geochronological methods
(e.g., $^{39}Ar/^{40}Ar$, U-Pb (SHRIMP, LA-ICP-MS)). In this manuscript, we do not directly compare
datasets from other studies. We do, indeed, aim to challenge the JKB recommend age in the ICS
is ~145 Ma, which mainly highlights the lack of precision and accuracy towards the JKB age. In
any case, the JKB age in the ICS is based on the $^{39}Ar/^{40}Ar$ age of Mahoney et al. (2005).
Nevertheless, our ages are so much younger than that of Mahoney et al. (2005), making precision,
not such a big deal for the sake of challenging the ICS age. Hopefully, other sections that span the
JKB will be dated in the future and most likely use U-Pb CA-ID-TIMS since it has become a gold-
standard in dating the stratigraphic record. Therefore, how we quote precisely in the manuscript is
not that big of a deal.

*2.8)    The chapter "Results and discussion" needs to be split into two, allowing results to be*
*clearly separated from the interpretation.*

In the same vein as the reply to comment 2.6, we wanted to make a concise manuscript.  In
this sense, we feel that the nitty-gritty dissection of the geochronological data should not be moved
to a separate "Results" chapter in the main text. Instead, we describe the data along with the
discussion, which in our opinion reads better and is not unusual in scientific communications. As far as the Solid Earth's author guideline goes, it does not mandate that results be separated from the discussion. Moreover, we think that the lack of a specific "Results" chapter does not compromise any of the discussion or conclusions in the manuscript. Therefore, we thought it might be better to leave the results and discussion together. We believe that this comment is more of a personal preference of the reviewer than a weakness of the manuscript. Nevertheless, we leave it at the discretion of the Handling Editor do choose what best fits the format of the journal because it would be an easy adjustment to make to the revised manuscript.

*2.9)    Even though it is widely accepted that magnetostratigraphy is very useful for global*

  *correlation in the JKB interval, projecting the magnetozones identified in the Arroyo*

  *Loncoche section in the Neuquén Basin (Iglesia Llanos et al. 2017) introduces additional*

  *confusion (p. 5, l. 1-9, Fig. 4) to the already complex web of stratigraphic correlation of*

  *the three studied sections. The new results from Las Loicas do not appear to be closely*

  *correlatable with Arroyo Loncoche, Fig. 4 reveals that the placement of the JKB is offset*

  *by nearly one ammonoid zone, being near the base or at the top of the Substeueroceras*

  *koeneni zone, respectively. It would suffice to say that magnetostratigraphy of the Las*

  *Loicas section will be desirable to enhance the utility of the newly obtained U-Pb ages and*

  *clarify contentious biostratigraphic correlation issues.*

  In the manuscript, we do not project the magnetozones of the Arroyo Loncoche section to the Las Loicas or any other section. We merely attempt to correlate the JKB in the Arroyo

Loncoche to the Las Loicas section using the Alpina Subzone and the M19.2n, which are the most compelling evidence for the JKB in either section. We admit that there is a mismatch between the ammonite zonations, which is clearly stated in the manuscript (p. 5, l. 8-9). Additionally, we also cited a discussion on the matter in López-Martínez et al., (2018). Nevertheless, the thickness of biozones changes as a function of facies, randomness of finding markers in the field, the latter hugely influenced by preservation, and paleogeographical position within a sedimentary basin.

Therefore, although we do see that better understanding the mismatch between both sections as an incentive for future research, we do not, however, see this as a significant issue to be explained.

One needs to keep in mind that another principal aim of this manuscript is to try to show
that the age of the JKB in ICS is too old. In an idealized case, one would find the age of the M19.2n
and the base of the Calpionella alpina Subzone to be the same age (assuming these markers are
exposed in different sections as is the case in this manuscript). This would require that both of
these markers have a datable horizon very close by. However, in the real world, this scenario is
quite hard to come by, and we need to try and reconcile the available data despite its shortcomings.
In the context of trying to show that the ICS age of the JBK (145 Ma) is too old, the data from Las
Loicas and Arroyo Loconche seems to be in reasonable agreement, in our opinion. That is, if we
consider that the most trustworthy markers for the JKB are the M19.2n and the base of the
Calpionella alpine Subzone, even with the mismatch of the ammonite zones between Las Loicas
and Arroyo Loconche (which would be a couple 100 ka), the age markers for the JKB of these two
sections would not be off by 5 Ma. **Furthermore, it is important to point out that in the absence**
**of a reliable biostratigraphic framework, such as the case of Arroyo Loconche,**
**magnetostratigraphy is just a floating scale (very important to bear in mind).** Therefore, from
this perspective, even with the ambiguity in the correlation between these two sections, the age of
the JKB at ~145 Ma is hard to reconcile. We do understand that the M19.2n in Arroyo Loconche
might seem older than the base of the Calpionela alpine Subzone in Las Loicas when compared
against the Substeueroceras koeneni biozone as a relative timescale. Nevertheless, this discrepancy
would not allow, for instance, the interpretation that the age of the M19.2n in Arroyo Loconche to
be as old as 145 Ma and the age of the Calpionella alpina Subzone in Las Loicas to be at ~140 Ma,
which would be the alternative to invalidating our conclusion.  Furthermore, our age in the
*Virgatosphinctes andesensis* biozone (Early Tithonian) would certainly not allow this
interpretation. In closing, the explanation above only exposes the how poorly constrained the
current age of the JKB is, that even with a crude correlation (which is what is available at our
disposable at this conjecture) the age of the JKB at 145 Ma seems implausible.

Additionally, in the manuscript, we cite many references that have also dated the JKB and
found ages similar to ours. Furthermore, our goes for the base of the Berriasian are much easier to
reconcile with the ages for the Early Cretaceous ages (see page 8, lines 10-10 in the manuscript).
In closing, there is substantial evidence from different fields that point to an age of the JKB that is much younger than in the ICS (We would also like to refer the reviewer to the reply on comment
1.2, i.e., in reply to reviewer #1).

Having said this, we realize that both reviewers took issue with our attempt to correlate the
M19.2n in Arroyo Loconche and the base of the Alpina Subzone in Las Loicas in an attempt to
build a more solid case for our age of the JKB. If our arguments remain unconvincing, we will not
oppose removing entirely this from the discussion and figure 4. Hopefully, our explanation was
satisfactory. We would, in this case, value comments and advice from the handling Editor on the
matter for the revised manuscript.

*2.10)    Discussion on the age of the JKB in the Mazatepec section includes an assumption on the*
*FAD of a nannofossil taxon, Nannoconus steinmannii minor, not actually found in the*
*section (p. 5, l. 31 – p. 6, l. 3). Such speculation is best avoided.*

Our consideration of the *N. Steinmannii* is speculative, very short. We certainly do not
substantiate any conclusion on this comment. Nevertheless, the N. steinmannii defines the base of
the biozone and is the main bioevent, and the others are defined as close and secondary with
regards to this bioevent. Therefore, this was just to give the reader food for thought, as so to speak.

*2.11)    Beware of the lack of formal definition of base Tithonian. There is no agreed-upon GSSP*
*decision yet, contrary to what is implied here (p. 6, l. 25). The attendant uncertainties of*
*stage boundary placement and its correlation with the Andean sections make the time scale*
*calibration use of La Yasera U-Pb date more problematic than admitted here.*

Indeed, there is no agreement on the GSSP for the Kimmeridgian-Tithonian boundary.  We will
remove the sentence in brackets that suggested otherwise (p.6, l 25), and will make it clear that the
KmTB is not formally defined.

*2.12)    The discussion on the duration of the Tithonian is interesting but contains a factual error*
*and misses some further opportunities. The Geological Time Scale 2016 (Ogg et al. 2016)*
*is misquoted, it assigns 150.8 Ma to the base of Tithonian Stage and 145.5 Ma to the JKB.*

In the ICS chart 2018, the age of the KmTB is 152.1 ± 0.9 Ma. Please see
http://www.stratigraphy.org/index.php/ics-chart-timescale. Additionally, in the compilation of
Ogg et al., (2016), Chapter 12 – Jurassic, Figure 12.1 page 152, and Figure 12.4 page 157, the age
quoted is 152.1 Ma for the base of the Tithonian.

*2.13)   It would be useful to compare two other, independent duration estimates. The Pacific M*
*sequence of magnetic anomalies has long featured in time scale calibration. The recent*
*work of Malinverno et al. (2012) (the MHTC12 scale) suggests 6 m.y. for the Tithonian,*
*i.e., between magnetochrons M22An and M19n2n.*

We thank the reviewer for pointing this out to us, and we will undoubtedly discuss and
compare Malinverno et al., (2012) timescale for the Tithonian in the discussion, especially since
it is very close to our estimate for the duration of the Tithonian.

*2.14)   The cyclostratigraphic analysis of Kietzmann et al. (2015; not cited by Lena et al.)*
*identifies 10 long eccentricity cycles for almost the entire Tithonian, starting with the*
*Virgatosphinctes mendozanus zone dated here at La Yesera, hence a duration of c. 4 m.y.*
*The discussion should emphasize that the duration favored here is longer these previous*
*estimates using other methods and offer possible reasons to explain the difference, perhaps*
*considering biostratigraphic correlation issues.*

There are two issues here: First, in Kietzmann et al., (2011) the Tithonian was more than
210 m thick in Arroyo Loncoche; then in Kietzmann et al., (2015) the Tithonian is reported as 195
m thick; and finally in Iglesia Llanos et al., (2017) the Tithonian was reported with less than 160
m. This makes more inadequate the ten long eccentricity cycles for almost the entire Tithonian.
The second issue is that following Vennari et al. (2014) and Riccardi et al. (2015), and in the
present manuscript, the andesensis (former mendozanus) zone is correlated with the Tethyan
ammonite zones, which are above the base of the Tithonian, i.e. the hybonotum zone is not
represented in Vaca Muerta Formation.

*2.15)   Perhaps my most important criticism and suggestion pertains to the projection of a*
*sedimentation rate-based JKB from the Mexican Mazatepec section into Las Loicas in*

*Argentina. The authors can make a much stronger case and build a more logical argument*

*by projecting the actual U-Pb date, expressing the stratigraphic height from the age-model*

*calculation as ~28.5 m and note the mismatch in biostratigraphies. Reading from Fig. 4,*

*beds of the same numeric age thus appear assigned to nannofossil zone NJK-B vs. high in*

*NJK-D, to calpionellid Crassicollaria zone vs. Calpionella zone (and its third subzone, the*

*Elliptica subzone, and ultimately to lower Berriasian vs. upper Tithonian at Las Loicas*

*and at Mazatepec, respectively. The discussion could thus be refocused to use the newly*

*obtained high-precision and high-resolution U-Pb age framework to highlight*

*biostratigraphic correlation issues, most likely due to diachronous FAD-LADs of certain*

*key taxa.*

We have partially addressed this inquiry in question 2.1. Nevertheless, we are happy with this comment because it further substantiates our arguments, especially for a JKB interval. We agree with J. Pàlfy that the mismatch in the age of the FAD-LAD in Las Loicas and Mazapetec is clear evidence that assuming age-equivalency of markers and stage boundaries is problematic when working at the sub-100 ka level and highlights the importance of high-precision geochronology to the stratigraphic record.  Furthermore, in the context of the JKB, it stresses the importance of leaving **the age** of the JKB confined to an interval (we further explore this in reply to question 2.16). We welcome this comment and will surely incorporate this into the revised manuscript because we see this as an essential implication from our data.

*2.15)*   *To strengthen the argument for potential problems in biostratigraphic correlation, the*

*authors might comment on the discrepancy of ammonoid-based correlation, and striking*

*differences of thickness of zones in different sections even within the Vaca Muerta Fm. (e.g.*

*Argenticeras noduliferum zone: ~27 m in Las Loicas vs. 5 m in La Yesera section).*

There are important facies and thickness changes between Las Loicas and La Yesera sections due to their different paleogeographic positions within the Neuquén Basin. La Yesera section is further east (see paleogeographic sections for example in Kietzmann et al. (2015).

*2.16)*   *It the "Global correlation" chapter, the suggestion of understanding the JKB as an interval*

*(p. 8, l. 1-10) is conceptually flawed and needs to be rephrased. By definition, the JKB*
*boundary (as any other chronostratigraphic boundary) is a time line. It does indeed carry*
*an uncertainty of our numeric calibration but it cannot be equated with an actual time*
*interval in which different "boundary events" took place.*

The reviewer may not have understood what we meant by the term Jurassic/Cretaceous
interval. We want to make it clear the JKB is not tantamount to JKB interval; in other words, they
are not the same thing. We did not suggest that the JKB be understood as an interval (at least that
was not our intention), but rather the age of the JKB be left within a bracketed interval, thus the
idea of the JKB interval. This mainly stems from the fact that the age of the JKB in both sections
do not overlap within our analytical uncertainty, and are offset by ~670 ka (± 335 ka). Furthermore,
as pointed out by the reviewer in comment 2.1, the markers are offset in an age which, in our
opinion, only builds a stronger case to leave the age of the JKB confined to an interval, the JKB
interval. In other words, what we propose here is that the interval constrained by our
geochronology is short enough that the JKB can be placed somewhere in that interval because a
single age is yet out of our reach. We feel confident that this interval can get tighter as newer
sections are dated in the future. Even though they do not overlap, the ages presented here highlight
a discrepancy between the age of the JKB in the ICS and the ages that we have measured.

*2.17)  Also in this final chapter, consider the significance of your argument for a significantly*
*younger JKB together with Martinez et al. (2015) suggested age for the base Valanginian*
*at 137 Ma. This would make for a shorter than previously understood Berriasian Stage of*
*a -3 m.y. duration. This in turn contradicts with the astrochronology of Kietzmann et al.*
*(2015), who identify more than 10 long eccentricity cycles in the Berriasian part of the*
*Vaca Muerta Fm.*

The issue with Valanginian boundary is presently in the discussion as well as the
Hauterivian and Barremian ages by new high precision U-Pb CA-ID-TIMS dating together with
cyclostratigraphy and the ammonoid and nannofossil biostratigraphy in the Neuquén Basin by

Beatriz Aguirre Urreta and Mathieu Martinez (in prep.). Some results already published also show several million-year discrepancies with the ICS Time Table.

*2.18)    The statement in chapter "6. Data availability" suggests that some of the raw data will be*

*withheld until completion of the thesis of the first author. Instead, all data should be made*

*available at the publication of this paper. Understandable practice is not to release data*

*in a thesis prior to publication, but there should be no reason to justify an embargo the*

*other way around.*

We will remove this section since all the data is reported in the data table in the supplementary materials. The reported U-Pb table data can easily be copied and pasted on the excel sheet, where it can easily be manipulated in Isoplot in Excel and or RStudio, for instance. Or instead, we can state the latter in chapter 6.

*2.19)    Table S1 contains the essential data for the U-Pb geochronology, it should be placed in*

*the main part of the paper.*

We disagree with the reviewer to place the U-Pb data Table, T.S1 to the main text. With the aim of keeping the manuscript more appealing, we feel that by putting raw data tables cuts the flow of the written text and distracts the reader. Therefore, we think that the data table T.S1 is better viewed separately from the main text, especially when reading in a digital format (which we encourage). It allows going back and forth from the text to the data table more readily if the reader deems necessary. Nevertheless, we leave it at the discretion of the Handling Editor do choose what best fits the format of the journal, and also because it would be an easy adjustment to make in the revised manuscript.

2.20)    *Fig. S is also worth transferring from the Supplementary Material to the main part.*

*(However, its labeling needs re-coloring so it be legible in black and white print, panel C*

*might be more informative to show the dated ash bed, D needs labels, and the figure needs*

*a caption.)*

In trying to keep the manuscript short, concise and to the point, we have opted to leave field figures (Fig. S) in the Supplementary Materials. We feel that figures that do not directly support any of the discussion or conclusion and are best kept in the supplementary material.

Nevertheless, we leave it up to the Handling Editor to advise us on what better suits the format of the journal. We thank the reviewer for pointing out that the figure was, unfortunately, left out the caption and we will incorporate his advice on how to better the figures such as recoloring for printing and better labeling.

**Closing remarks from the authors**

In closing, we would like once more to show our appreciation to J.Páfly for reviewing our manuscript and accepting it for publication after the revision. Many of the reviewer's suggestions we agree and will fully accept, with only a very few where we disagree or would not favor the change. For instance, there where two comments that, in our opinion, that standout and substantially add to the manuscript. First, the refocusing the discussion around the apparent mismatch between the ages of the biozones in Las Loicas and Mazatepec, which we address in question 2.1 and 2.14. This is the most critical comment from the reviewer, and we welcome it and assure we will incorporate this will be added to the discussion in the revised version. Second, the renaming of the "Studies areas" section for a "Geological and Stratigraphical Setting" and an expansion of both sections. On this comment, we argue as to why we felt it was essential to leave the Studies areas section short, but did not oppose to reviewer's suggestion. In any case, leave it to the decision of the Handling Editor for the revised version.

All other comments from J. Pálfy, albeit pertinent, we feel that they are minor and straightforward to adjust. For instance, the reviewer suggests a "Results" section separate from the

Discussion section, which would imply moving the description of the results found in the

Supplementary Material to a new chapter entitled Results in the main text. Another similar request is to place the raw data tables in the Supplementary Material in the main text. Even though we oppose such changes in the structure, we do not see it as a significant modification to the manuscript, and we leave it to the Handling Editor to decide what would best fit the journal's format. Other requests pertain to improving the readability and clarity of the figures, adding a caption to one of the supplementary figures. Modification in the grammar usage, word choice, style, and spelling will promptly modify since they will improve the manuscript. In short, we feel that we have dealt with all of the reviewer's comments adequately and hopefully, the answers fulfill the requirements for publications by both the reviewer and the Handling Editor.

**Technical corrections**

*The comments below also include several suggestions for better English language,*

*style and word choice.*

*p. 1, l. 12 (and elsewhere): age ! numeric age*

We discuss this in reply to comment 1.10 to reviewer #1 W. Wimbledon. Since this is a paper that discusses the age of a boundary from a geochronological perspective age is necessarily a numerical age. In our view, it would be some tedious to specify age every time. In the introduction, however, we use the "absolute age" nomenclature to distinguish it from the more older ages derived from statistical interpolation. Therefore, in that context, we felt it was necessary to make the distinction.

However, throughout the text when we mention "ages", it can only be numerical ages or numeric ages because what we present are U-Pb ages, which are numerical by definition. Therefore, we feel a distinction is not necessary.

*elusive ! difficult to determine*

OK. Agreed

*l. 16: display ! contain*

OK. Agreed.

*l. 21: one of the last major Phanerozoic stage boundaries ! last Phanerozoic system*

*boundary*

OK. Agreed.

*l. 23: absolute ! [delete, avoid "absolute age" altogether]*

OK. Agreed.

*p. 2, l. 3: Calpionella alpina subzone (cf. l. 16) [ensure consistency in zonal names*

*and terminology]*

OK. Agreed.

*l. 17: selected ! suggested*

OK. Agreed.

*l. 21: Kamptneri ! kamptneri*

OK. Agreed.

*p. 3, l. 6: spans ! exposes*

OK. Agreed.

*l. 12: out of sequence numbering of figures (not as they appear in text)*

OK. Agreed. We will modify.

*l. 26: optical images ! photomicrographs*

OK. Agreed.

*p. 4, l. 3: The section ! The Las Loicas section*

OK. Agreed.

*l. 29: impose ! may provide*

OK. Agreed.

*p. 5, l. 2, 6: fossil density ! abundance of fossils*

OK. Agreed.

*p. 6, l 24: Tithonian*

OK. Agreed.

*p. 9, l. 22: thank*

OK. Agreed.

*p. 10, l. 3: Neuquén*

OK. Agreed.

*p. 11, l. 11: [delete] February*

OK. Agreed.

*p. 12, l. 6: Potosí [+spell out journal name]*

OK. Agreed.

*l. 13: & [delete]*

OK. Agreed.

*p. 14, l. 4: Episodes [delete the rest of name]*

OK. Agreed.

*l. 14: Aguirre-Urreta*

OK. Agreed.

*l. 22: [provide doi instead of URL]*

*References cited in text but not listed in reference list: Edwards, 1963 R Core Team,*

*2013*

OK. Agreed.

*p. 15, l. 3: Distribution of continents ! Global paleogeography*

OK. Agreed.

*l. 9-15 (Fig. 3): Give stratigraphic horizon of occurrence (e.g. m from base) fo each*

*specimen photographed*

OK. Agreed.

*p. 16, Fig. 1: delete title, consider using different base map, do not show migrazion*

*routes and sections not discussed in text.*

OK. Agreed. We will consider just leaving only the two sections studied. However, it is quite common to add sections that are of the same age to a paleogeographical maps to give a to give the sense of the time equivalent between sections even though they are not discussed in the text.

*p. 17, Fig. 2: Barriasian ! Berriasian*

*[J/K boundary interval – see comments about conceptual flaw here]*

*[fonts too small in the upper part, too large in the lower part]*

OK. Agreed.

*p. 18, Fig. 3: [it is redundant to show taxon names here, it is customary to give them*

*in the caption only]*

OK. Agreed.

*p. 19, Fig. 4: [this is the key figure of the paper, already need to refer to in the*

*Geological setting, so make it Fig. 2; A: show meters; put Las Loicas section to a*

*separate panel B, making the others C and D; La Yesera: indicate placement of JKB;*

*some lettering uses illegibly small font]*

Here we disagree with the reviewer. There is no sedimentological or geological consideration is figure 4, but rather a comparison between the ages of markers from each section. Furthermore,

Figure 4 should be within the discussion chapter as the bulk of the discussion pertains to this figure.

Therefore, we do not see the purpose of it being placed at the beging on the manuscript.

*Supplementary Material*

*p. 1, l. 1: Ash beds were crushed ! Samples were crushed*

OK. Agreed.

*p. 3, part 5*

*Give weight of each sample so zircon yield can be assessed in this context.*

*Grains discarded as too old are erroneously quoted as >_150 Ma for each sample,*

*provide true cut-off age of grains not included in age calculation.*

Weight of the samples was not made because it is not customary to do so. Grains 150 Ma were discarded. The cut-off age for grains included in the weighted mean is sample dependent and are usually the youngest overlapping grains.

*5.3 (p. 4): Ash bed LL10 has n=6 grains in Fig. 2, four in text*

OK. Agreed, will change it to 4, not 6.

*5.4. Ash bed LL13: include date of discarded grains in Table S1 (really older than 450*

*Ma?)*

We do not see the point of reporting the age of grains that are significantly older than the weighted mean of the ash bed. It serves no purpose. Ages much older than the weighted mean are hard to evaluate if they are detrital of inherited from older basement rocks volcanic source.

*5.5. "Due to its proximity to the Tordillo Fm." [it is from the Tordillo Fm.]*

*inherited grains or detrital grains?*

OK. Agreed.

*5.6. MZT-81 (p. 5): check this descriptions, there are errors here. four discarded grains*

*(not five), the grain numbers are in error (belong to sample LL10)*

OK. Agreed. Thanks for pointing this out. Will be rectified.

*Fig. S needs a caption and should be transferred to the main part of the paper. The*

*labels of the figures need to be recolored so they are legible in black and white print as*

*well.*

OK. Agreed.

*Table TS.1 is essential to assess the U-Pb dates reported so it should be transferred*

*to the main part of the paper.*

Please see the discussion to comment 2.19.

*Sample LY5 in Table TS.1: why discard grain z67 and keep z10, when the first one is*

*not older and its error is not larger? This and similar issues of only marginally different aged*

*grains undermine the credibility of unbiased and rigorous selection of grains for the age*

*interpretation.*

Weighted mean ages are nothing other than the average mean value of set of dates (youngest grains). In this case, grain LY z67 has a mean value of 147.740 Ma and the precision with what we know the true age of the grain is 93 ka. In figure 2, it is quite clear that LY z67 does not overlap with the weighted mean age of the youngest grains, which means it has little to no chance of statistically belonging to the subset of youngest grains of the population. On the other hand, LY z10 has a mean value of 147.8 Ma and the precision with which we know the age of the

1.1 Ma (much lower precision), and from Fig. 2 it clearly overlaps with the weighted mean age of the sample, which implies that it does have some probability of being a part of the subset of younger grains. In short, LY z10 statistically has a better chance of belonging to the subset of the youngest grains than LY z67, even though the mean value of LY z67 is slightly younger than LY

z10. This is just a question of precision, or how well-known is the confidence interval for a particular physical measurement. We draw the attention of the reviewer to compare the Pb*

concentration of these two grains. Here, precision is mainly limited by the amount of sample. If the sample size was any bigger, the precision would be higher. Thus the confidence interval reduced. And in that case, grain LY10 would have possibly been excluded from the weighted mean age of the ash bed.

*8.2 (p. 11), Table TS.2: Why is the age value of 2 m any different from the age of LL13*

*taken from this level?*

This is because the stratigraphic height of LL13 is in fact at height three m and not two m.

This will be rectified in the main text. Notice that the age of LL13 is $142.039 \pm 0.058$ Ma and the age of stratigraphic height 3 m is 142.04 ± 0.06 Ma, which is because we have rounded the numbered to two decimal places rather than three. Thank you for pointing that out.

**References**

Iglesia Llanos, M. P., Kietzmann, D. A., Martinez, M. K. and Palma, R. M.: Magnetostratigraphy of the Upper Jurassic–Lower Cretaceous from Argentina: Implications for the J-K boundary in the

Neuquén Basin, Cretac. Res., 70(February), 189–208, doi:10.1016/j.cretres.2016.10.011, 2017.

Kietzmann, D. A., Martí-Chivelet, J., Palma, R. M., López-Gó, J., Lescano, M. and Concheyro,

A.: Evidence of precessional and eccentricity orbital cycles in a Tithonian source rock: The mid- outer carbonate ramp of the Vaca muerta formation, Northern Neuqué Basin, Argentina, Am.

Assoc. Pet. Geol. Bull., 95(9), 1459–1474, doi:10.1306/01271110084, 2011.

Kietzmann, D. A., Palma, R. M. and Iglesia Llanos, M. P.: Cyclostratigraphy of an orbitally-driven

Tithonian-Valanginian carbonate ramp succession, Southern Mendoza, Argentina: Implications for the Jurassic-Cretaceous boundary in the Neuqu??n Basin, Sediment. Geol., 315, 29–46, doi:10.1016/j.sedgeo.2014.10.002, 2015.

López-Martínez, R., Barragán, R. and Reháková, D.: The Jurassic/Cretaceous boundary in the

Apulco area by means of calpionellids and calcareous dinoflagellates: An alternative to the classical Mazatepec section in eastern Mexico, J. South Am. Earth Sci., 47, 142–151, doi:10.1016/j.jsames.2013.07.009, 2013.

López-Martínez, R., Aguirre-Urreta, B., Lescano, M., Concheyro, A., Vennari, V. and Ramos, V.

A.: Tethyan calpionellids in the Neuquén Basin (Argentine Andes), their significance in defining the Jurassic/Cretaceous boundary and pathways for Tethyan-Eastern Pacific connections, J. South

Am. Earth Sci., 78, 1–10, doi:10.1016/j.jsames.2017.06.007, 2017.

López-Martínez, R., Aguirre-Urreta, B., Lescano, M., Concheyro, A., Vennari, V. and Ramos, V.

A.: Reply to comments on: "Tethyan calpionellids in the Neuquén Basin (Argentine Andes), their significance in defining the Jurassic/Cretaceous boundary and pathways for Tethyan-Eastern

Pacific connections" by Kietzmann & Iglesia Llanos, J. South Am. Earth Sci., 84, 448–453, doi:10.1016/j.jsames.2017.12.003, 2018.

Mahoney, J. J., Duncan, R. A., Tejada, M. L. G., Sager, W. W. and Bralower, T. J.: Jurassic-

Cretaceous boundary age and mid-ocean-ridge-type mantle source for Shatsky Rise, Geology,

33(3), 185–188, doi:Doi 10.1130/G21378.1, 2005.

Malinverno, A., Hildebrandt, J., Tominaga, M. and Channell, J. E. T.: M-sequence geomagnetic polarity time scale (MHTC12) that steadies global spreading rates and incorporates astrochronology   constraints,   J.   Geophys.   Res.   Solid   Earth,   117(6),   1–17, doi:10.1029/2012JB009260, 2012.

Ogg, J. G., Ogg, G. M. and Gradstein, F. M.: Permian, in A Concise Geologic Time Scale, pp.

115–131, Elsevier., 2016.

Riccardi, A. C.: Remarks on the Tithonian-Berriasian ammonite biostratigraphy of west central

Argentina, Vol. Jurassica, XIII(2), 23–52, doi:10.5604/17313708, 2015.

Vennari, V. V., Lescano, M., Naipauer, M., Aguirre-urreta, B., Concheyro, A., Schaltegger, U.,

Armstrong, R., Pimentel, M. and Ramos, V. a.: New constraints on the Jurassic-Cretaceous boundary in the High Andes using high-precision U-Pb data, Gondwana Res., 26(1), 374–385, doi:10.1016/j.gr.2013.07.005, 2014.

---

## Author Comment (AC2) · 16 Aug 2018

Reply to comments by reviewer W. Wimbledon (reviewer #1) on the manuscript "Cross-Continental age calibration of the Jurassic/Cretaceous boundary"

Comments by the reviewer on his report have been copied and pasted in the italic blue font, and our replies are found immediately below in regular black font. The comments/replies are in order of appearance as in the reviewer's report. We have taken the liberty to number them 1.1 through 1.14 to facilitate proofing from the Handling Editor and the reviewer himself. Subsequently, selected (the majority) comments made by the reviewer in the "supplementary comments pdf" (i.e., comments made made directly on to the manuscript) are also replied to here in order of appearance referencing

the page and line where they were made. The remaining replies to some of the comments made by the reviewer in his "supplementary comments" (mainly just language usage corrections) have been replied directly on reviewer's "supplementary comments pdf" attached to the discussion. There, our replies are found in red background with black font. References are found here, right after the replies to the "supplementary comments" section.

First, we would like to kindly thank W. Wimbledon for accepting to review this manuscript and say that we welcome his comments and corrections. Before directly addressing the reviewer's comments, we would like to preface this reply by objectively restating the aims and goals of the manuscript. By doing so, we hope to clarify our objectives and use it to substantiate our answers, because we believe that some of the comments and suggestions from W. Wimbledon seem to reveal a misunderstanding of the aims of the manuscript. Briefly stated, we aim to measure the numerical age of the Jurassic/Cretaceous boundary (JKB). In the introduction of the manuscript, we briefly explain why the numerical age of the JKB has remained contentious throughout the years and thus highlight the relevance of the research. We refer to the opening comment J. Pálfy, which states: "This contribution is significant and timely, as the calibration of the JKB has remained uncertain due to a dearth of reliable numeric ages around it". Subsequently, we describe how we aim to tackle the issue. In our opinion, the best way to measure the numerical age of the Jurassic/Cretaceous boundary (JKB) is to date the age of the base of the Calpionella alpina Subzone, which is the primary marker for the JKB. To avoid local and regional biases, we cross-calibrate the numerical age of the JKB by dating it in two independent sections, in a distinct geological context where the JKB is well-defined by more than one marker. The geochronological tool of choice was high-precision U-Pb zircon geochronology on interbedded volcanic ash deposits found in each section., which recently has become the best geochronological tool to date the stratigraphic record. Furthermore, we also present calcareous nannofossil results to better anchor the JKB in the Mazatepec section, previously only calibrated by Calpionelldis. One of the conclusions of the manuscript is that the reasonable agreement

between the age of the JKB in both sections represents the age of the boundary, and can be used to date the JKB in other sections that contain the boundary. More importantly, the main contribution of this manuscript is to provide evidence that the quoted age for the JKB, as currently found in ICS 2017 at ∼145 Ma, is substantially older when compared to our geochronological dataset. Lastly, in the manuscript, we point to the fragility of the biostratigraphy on which the current age of the JKB (∼145Ma) is anchored. For instance, comparatively, the biostratigraphy presented in Mahony et al. (2005) pales in comparison to the biostratigraphy of both sections investigated in this study. Additionally, Mahony et al. (2005) use 39Ar/40Ar geochronology, which over the years has gradually lost its usefulness as a geochronological method for dating the stratigraphic record for the lack such accuracy and precision required to calibrate stage boundaries. In closing, W. Wimbledon's comment that we only talk about geochronology is to some extent quite right. The primary aim of the paper is to study the JKB from a geochronological perspective, and we leave that very clear from the very beginning. This makes this study especially important because geochronology has been a void surrounding the topic for many years. This leads us to the first comment by W. Wimbledon regarding magnetostratigraphy.

General Stratigraphic remarks – Magnetostratigraphy

COMMENT 1.1) The text should perhaps say that there is no possibility of magnetic calibration of Las Loicas with the many Tethyan sites where it has been documented.

REPLY: A subset of the authors are pursuing the magnetic calibration in Las Loicas. The preliminary sampling has already been done, and some results are available yet not published. The main obstacle is that the basinal facies in Las Loicas which makes it difficult to have a dense suitable sampling for magnetostratigraphy. Therefore, efforts are being made to overcome this issue. Please see reply to comment 1.2 and 2.9 (i.e., in reply to reviewer #2 J. Pálfy) for further clarification on our attempt to correlate the magnetostratigraphic data from Arroyo Loconche with Las Loicas.

COMMENT 1.2) the ammonite zonations applied at the LL and AL do not agree – a big problem.

REPLY: It needs to be made clear that we do not present any new magnetostratigraphic data, but instead use the magnetostratigraphy of Iglesias 2017 to aid marginally and back-up our age of the JKB in Las Loicas. Magnetostratigraphy is not the focus of the paper nor did we state in the manuscript that we aimed to do that, but rather an aside. Meaning, we use it as a reflection on how other substantial evidence for the JKB from the Neuquen Basin might agree with our data. We recognize that magnetostratigraphy is a significant component in calibrating and locating the JKB in sections that span the JKB. However, we are fully aware of the seemingly conflicting evidence from the ammonite zonation from Arroyo Loncoche and Las Loicas. We clearly stated that the ammonite zonation in Arroyo Loncoche is preliminary and also cited a discussion around the matter in López-Martínez et al. (2018). The main point discussed in López-Martínez et al. (2018) is that both discussions contain the different resolution of data. The ammonite biostratigraphy of Las Loicas is based on the bed by bed collection from 54 fossiliferous levels with 450 ammonite specimens. López-Martínez et al. (2017 Fig. 1) and Vennari et al. (2014) recorded 35 fossiliferous levels and studied 228 ammonite specimens. Therefore, we feel that the ammonite zonations in Las Loicas is well-defined and described. On the other hand, in the Arroyo Loncoche region there is not a single published section with the ammonite levels, or the number of specimens collected, which renders the definition of the biozones unreliable. It is also evident that the boundaries of the biozones in Arroyo Loncoche have been changing along the years, as well as unit thickness, the presence of sills, etc. We invite the reviewer to take a closer look at the discussion in Lopez-Martinez et al. 2018 but include here an extraction from the paper to illustrate the issue. Iglesia Llanos et al. (2017), p. 194 state that "The boundary between ammonite zones in Arroyo Loncoche was placed according to the first occurrence of the index species." However, the range chart with vertical distribution of the taxa (their Fig. 2) and the ammonite biozones do not follow this criterion. For instance, the base of the Corongoceras alternans zone is placed at

the first occurrence of Corongoceras sp. and the index species is not even recorded in this section. Furthermore, the base of the Substeueroceras koeneni zone is placed on the first occurrence of Substeueroceras sp. (at 150m of the base of the section) while the index species appears higher (above 180 m). This more than 30m discrepancy explains the different biozonation of the same section published by Kietzmann et al. (2011 Fig. 3) where they placed the base of the Substeueroceras koeneni zone at 190 m of the base of the Arroyo Loncoche section. Lastly, it is important to point out that in the absence of a reliable biostratigraphic framework, such as the case of Arroyo Loconche, magnetostratigraphy is just a floating scale. In conclusion, the paper aims to calibrate the numerical age of the JKB using high-precision geochronology in Las Loicas and Mazatepec using the base on the base of the Calpionella alpine zone as the primary marker for the JKB. In Figure 4, we correlate the JKB Arroyo Loconche and Las Loicas based on more compelling evidence for the JKB which is the M19.2n (Arroyo Loconche) and the base of the Calpionella alpine Subzone (Las Loicas). Therefore, we avoided normalizing the two sections based on ammonite. Incidentally, in his 2017 review of the JKB, W. Wimbledon (Wimbledon, 2017) has also normalized Las Loicas and Arroyo Loncoche disregarding the apparent mismatch of the ammonite zones in the working model for correlating the regions for the JKB. Furthermore, in our Figure 4, the correlation between the M19.2n in Arroyo Loncoche and the base of the Calpionella alpine subzone in Las Loicas is a dashed red line, which suggests that the correlation is merely conjectural. As W. Wimbledon pointed out, Phanerozoic stage boundaries are not dependent on geochemistry, magnetostratigraphy or geochronology. These are just tools used to aid the calibration of stage boundaries, and mismatches are commonplace. Therefore, we do not see this first point as a big problem. The reviewer #2, J. Pálfy, also took an issue with this matter. We kindly ask the reviewer to also read the reply on comment 2.9 (i.e., in reply reviewer #2). Hopefully, it will supplement this reply and vice-versa.

COMMNENT: 1.3) The calpionellid assemblage noted at Las Loicas is anomalous: such a mixed assemblage (with apparently derived Tithonian calpionellids) does not

define or mark the base of the Berriasian. It should be made clear what is definitively lower Berriasian and what is not.

REPLY: First, it is essential to recognize that sections containing datable horizons close to boundaries, such as the Las Loicas and Mazatepec, are extremely rare which is a significant hindrance in calibrating the age of stage boundaries in general. Tethyan and Mediterranean sections do not contain datable horizons, because these sections are deposited in passive margins far from plate tectonic boundaries where a considerable amount of acidic-aerial volcanism output is produced allowing for the deposition of ash fall deposits (ash beds). Therefore, even though the issues surrounding the JKB have been concentrated in the Tethys region, its age, on the other hand, will not. Although the reviewer claims that the Las Loicas contains "anomalous" calpionellid assemblages, if we are ever to advance in the knowledge of the numerical age of the JKB, we have to use everything at our disposal. Replying to the reviewer's comments, the only reported "anomalies" in Las Loicas are a) he presence of Tintinnopsella remanei in the upper part of the Crassicollaria Zone. This is a none typical appearance in the Mediterranean Tethys, but usual in western Tethys as discussed in López-Martínez et al., (2017), and b) the record of Crassicollaria massutiniana in the lowermost part of the Alpina Subzone. Even when it can be unusual the presence of this species in the Lowermost Berriasian, this does not affect the biozonation scheme as the Alpina Subzone is defined by the acme of Calpionella alpina small and globular form and not the Last Occurrence of any species. Then, the Alpina Subzone is defined in the same way as in the Mediterranean Tethys and can be used as a marker of the JKB in Las Loicas.

COMMENT: 1.4) The nannofossil literature cited as the justification for some of the text's discussion and conclusions is rather old - Bralower and Casellato references are now 10 -30 years old. Many Tethyan sites have since been documented, and that make some of the species FADS and the zones discussed obsolete. Some Italian localities cited in the text are seen as anomalous in the positions of their nannofossil FADs. Thus it is not clear why these localities are selected by the authors for comparison with the

LL and M sites, especially when they are not the best/most representative.

REPLY: We, unfortunately, have to disagree with this comment. There are two standard calcareous nannofossils zonations for the studied interval. (Bralower et al., 1989) proposed a calcareous nannofossil zonation for the Jurassic and Cretaceous based on southern European land sections and the western North Atlantic, DSDP Sites 391C and 534A. (Casellato, 2010) proposed a new calcareous nannofossil biostratigraphic scheme for the Tithonian–Early Berriasian established for the Southern Alps in Northern Italy. Even though many recent papers deal with nannofossils of this time interval, there are no new zonations for this interval. Therefore, these two papers form the basis for newer studies, with many of the recent publications still citing the zonation in the classic papers of Bralower et al. (1989) and Casellato (2010). We agree with the reviewer that these publications might be considered old, but in no way, shape, or form, can they be considered outdated or overtaken since the zonation presented in them form the basis of the more recent works on calcareous nannofossils zonation of this period. To illustrate, we take the liberty of copying below excerpts from the newer publications on calcareous nannofossil of this period that promptly cite the work of Bralower et al. (1989) and Casellato (2010), as we have. a) Grabowski et al., (2017). Sedimentary Geology 360, p. 57, state: "For biostratigraphic purposes, the available biostratigraphic schemes of Bralower et al. (1989), Bown and Cooper (1998) and Casellato (2010) were considered. The latter was selected to apply for the Lókút section, as the most appropriate for nannofossil record in this Tethys location". b) Hoedemaeker et al. 2016. Revue de Paleobiologie 35, p. 190, state: "CALCAREOUS NANNOFOSSILS (C. E. Casellato and S. Gardin). . . . Calcareous nannofossils are rare to common and poorly to well preserved, with overgrowth more pervasive than etching. Assemblages are of Tethyan affinity . . . . . .(and) the biostratigraphic schemes adopted in this study are those of Bralower et al. (1989) and Casellato (2010). . . ." c) Ogg et al., (2012). A Concise Time Scale, p. 170, state: Use for defining the JK boundary in the Mediterranean Tethys similar FOs of calcareous nannofossils as those used in our paper (fig. 13.2.). d) Schnabl et al., (2015). Geologica Carpathica 66, p. 491,

state: "For several generations, apart from occasional aberrations, definitions of a J/K boundary have focused on one interval, between the base and top of one ammonite subzone (that of Berriasella jacobi), and, in the last thirty years, more and more, on the widespread and more consistently recognized turnover from Crassicollaria assemblages to small Calpionella. . .. Latterly this has been widely reinforced by the use of calcareous nannofossil FADs (references in Casellato 2010)." e) Sbodova and Kotsak 2016. Geologica Carpathica 67, p. 225 state: "Biostratigraphic data were interpreted with reference to the nannofossil zonation of Casellato (2010), commonly used for the Upper Jurassic and the Lower Cretaceous in the Tethyan/Mediterranean area". Regarding some anomalous positions of the nannofossils. p. 231. "It should be noted, that the LO of N. kamptneri minor usually appears a little above the LO of N. steinmannii minor, but in this paper it occurs together with the LOs of N. steinmannii steinmannii and N. kamptneri kampteri in bed 35. This anomaly can be explained by the very poor preservation and extreme etching of calcareous nannofossils between beds 32 and 34. Moreover, the appearance of these four species together suggests the presence of a hiatus." f) Bakhmutov et al., (2018), Geological Quarterly, 62 p. 232, state: "The first appearances of species of significant calcareous nannofossils at Theodosia are shown in Figure 23. The appearances are not consistently equivalent to all records in western Tethys (Casellato, 2010; Schnabl et al., 2015), one reason being that in this preliminary study we did not sample beds below a level we believe to be assignable to the lower to middle part of M19n.2n. . .. . .. However, the FADs in M19n of H. strictus, C. cuvillieri, N. wintereri, N. steinmannii minor and N.kamptneri minor appear to be consistent with other regions."

COMMENT: 1.5) The dating of the magnetozones needs to highlight and discussed at more length in the Discussion.

REPLY: Very confusing comment. Also on magnetostratigraphy, the reviewer comments on the dating of magnetozones. Nowhere in the manuscript, it is stated that we have dated magnetozones, or implied doing so. This is clearly beyond the scope of

the manuscript, and additionally, not even possible since we do not present any magnetostratigraphic new data in any of the studied sections. To date magnetozones, one would have to present magnetostratigraphic and geochronological data on the same section, which we have not done nor said we had. It would not be scientifically sound to do otherwise. Therefore, we feel the request of the reviewer is rather odd and unjustified. 1.6) Also, the assumptions (as seen in most publications) about using the magnetostratigraphic scale as a time scale could be laid out fully in the Introduction The reviewer also asks us to discuss the assumptions of using magnetostratigraphy, which again, is not the aim of the paper since we do not present any magnetostratigraphic data whatsoever. Therefore, to discourse about the use of magnetostratigraphy is beyond the scope of the dataset we present not to mention beyond the point of the problem we are trying to solve and the methods we use. Again, an odd comment from the reviewer.

COMMENT: 1.7) Notably, Ogg et al 2016 is not at all 'official' and is not attributable to ICS, but this is not clear from the text. REPLY: The reviewer does have a point and will make the distinction clearer between the ICS and Ogg 2016.

Structure

COMMENT: 1.8) The chronostratigraphic and biostratigraphic background should be made clear before consideration of any new data on radiometric dates.

REPLY: The issues and intricacies with fixating the JKB are well-documented in several publications which W. Wimbledon himself has authored and co-authored. The biostratigraphy in both sections we use has also been well-documented in other publications by some of the authors in this manuscript. Therefore, we feel that this request is somewhat unnecessary since the issue has been dealt with quite thoroughly in other publications and have been cited throughout the manuscript when necessary. Furthermore, we have taken Solid Earth's recommendation on the manuscript type, where manuscripts should be short, concise, and to the point which is a trend among high-

impact journals. We feel that reviewing the biostratigraphic framework would make the manuscript unnecessarily long. In closing, the manuscript is not dedicated to reviewing any previous data, but rather presenting new data and building on pre-existing biostratigraphy.

COMMENT: 1.9) The extrapolation of the GMPTS to onshore localities is central. The paper is concerned with attaching radiometric dates to a biostratigraphic framework. But it says very little about how radiometric dates match the timescale used by, for instance, Gradstein et al 2012: a time framework linked to the oceanic magnetostratigraphic record, the GMPTS. The last is hardly mentioned.

REPLY: Extrapolating the GMPTS would be over-interpreting our data and certainly beyond the scope of the manuscript. Nevertheless, we feel this is the main problem that the International Chronostratigraphic Chart of International Commission on Stratigraphy (2005 to 2018 versions) is facing. The 145 Ma age for the JKB boundary is based on the Shatsky Rise magnetozones. The main drawback for this assertion is the accepted age for the base of the Berriasian, supported by a poorly dated age of the Shatsky Rise in the Pacific Ocean. The radiometric Ar-Ar dating, even if only the best two samples were considered, have reduced plateaux that could indicate some 39Ar recoil (Mahoney et al., 2005). The ages of these samples are 144.8 $\pm$ 1.2, 143.7 $\pm$ 3.0, and 142.2 $\pm$ 5.3 Ma, but 145 Ma was the preferred age as it coincides with the spreading rate assumed for this part of the Pacific Ocean floor (Ogg et al., 2012). Besides, when the biostratigraphic controls of the sediments of the Shatsky Rise intruded by the dated sills are taken in consideration, the results are not very well constrained (Mahoney et al. 2005 cited (Bown, 2005)): Quoting Bown (2005) : "Zone NK1; Berriasian (Site 1213): The Jurassic/Cretaceous boundary interval zonation of Bralower et al. (1989) is based on a distinctive succession of nannolith appearances, notablyÂăConusphaeraÂăandÂăNannoconus; however, these taxa were absent in this part of the section and the former was absent throughout. In addition, a number of important marker species of the family Cretarhabdaceae (C. cuvillieri,ÂăR. angustiforata, andÂăRetecapsa octofenestrata) and genusÂăEiffellithusÂă(E. primus,ÂăE. windii, andÂăE. striatus), although present, are rare and restricted to a small number of samples, and their first and last occurrences may not be biostratigraphically reliable. The lowest Cretaceous zones are thus identified using marker species, where present, together with alternative datum events and aspects of the entire assemblages. The lowermost productive samples (Core 198-1213B-27R) yieldedÂăH. chiastia,ÂăL. carniolensis,ÂăTubodiscus bellii, andÂăR. laffittei, indicating Subzone NJKc or younger. The nannofossils do not unambiguously indicate a Cretaceous age, but correlation with Zone NK1 is inferred based on the presence of the genusÂăTubodiscusÂăand absence ofÂăR. angustiforata,ÂăP. fenestrata, andÂăR. wiseiÂă(Bralower et al., 1989). Support for this interpretation also comes from radiolarian fauna that also indicate a Berriasian age for the lowermost cores (H. Kano, pers. comm., 2003)". In conclusion, Bown (2005) makes it clear that the nannofossils do not unambiguously indicate a Cretaceous age but also the inferred correlation with zone NK1 is based in the presence of ONE GENUS and the ABSENCE of three species (negative evidence!). There are not markers, no first appearances, etc. Regarding the radiolarians, the data are based only in personal communication. The facts stated above discredit the JKB extrapolation of the Global Magnetic Polarity Time Scale (GMTPS) based on the Shatsky Rise to onshore localities currently in use. In any case, it is not the focus of this paper to criticize the age of the magnetic polarity time scale used to define the boundary, which needs a deep revision in our opinion.

COMMENT: 1.10) The core of the paper could usefully be a careful examination of the calibration of the dated ash horizons and the levels with the key biostratigraphic markers – listing them in sequence, level by level. REPLY: The detailed biostratigraphy of Las Loicas with an indication of the fossiliferous levels and relevant markers is indicated in Vennari et al., (2014) and López-Martínez et al. (2017). The objective of this paper is to date this biostratigraphy with accurate and precise ages in the relevant interbedded tuffs coupled with age-depth modeling. In the Mazatepec section, biostratigraphy based in calpionellids in found López-Martínez et al. (2013) is complemented by new

calcareous nannofossil occurrences which are presented here and presented in figures and pictures.

Precision, accuracy, English Language

COMMENT: 1.11) There are numerous examples of rather problematic phrases and sentences which are not written in good English. But more critical is the lack of precision or looseness in language and terminology. This lets down the submission very badly. It is the thing that needs the most attention in a revision by the authors. We do admit that many of W. Wimbledon's suggestions on our English usage and grammar (or lack thereof) are correct and we welcome them. We incorporate all the suggested words, variations and rewrite all sentences pointed out that remain unclear and confusing. Names of species will be thoroughly revised. Specific replies to comments on the supplementary section are found below.

REPLY: Corrections on some spelling mistakes are just differences between American and British English. Spellings such as gray, meter, catalog, paleontological, memorize, analog, analyze, defense, color, aging, inquiry, license among many other words are a correct and legitimate form of spelling in American English. Therefore, since Solid Earth does not dictate which kind of English is to be used in their publications, we chose to use American spelling. Furthermore, we feel we were consistent with our choice of spelling thought out the manuscript. Therefore, we have decided to disregard the reviewer's comments on these spelling mistakes.

COMMENT: 1.12) The loose wording of the Abstract's and Introduction's first sentences. No, the age of the J/K boundary is very clear. Lena et al. talk only about radiometric dating. They should say that the start of Berriasian age/base of the Berriasian stage has been more or less fixed for some years [the authors actually quote several relevant papers that show this]

REPLY: This comment is quite confusing, and we are not sure what the reviewer meant by this. We hypothesize it might have to do with how different fields in the Earth Sciences use the word "age" with subtle nuances, which is understandable. For instance, in the field of paleontology, an age of a fossil can sometimes be ascribed as an age of a stage. For instance, saying "fossil XY has a Tithonian age" or "is Tithonian" is perfectly acceptable when used in this context. However, in the matter of calibrating the numerical age of stage boundaries such usage of the word age is too loose because a stage boundary can last for millions of years; therefore, it lacks accuracy and precision. In the context of calibrating the age of a stage boundary, the word "age" needs necessarily to be taken as a numerical age (or radiometric age), usually arising from a physical measurement which carries a mean value and an error. Since this manuscript deals with the age calibration of stage boundary from a geochronological perspective, no other meaning of the word "age" is possible other than a numerical age. Therefore, every time the word age appears in the manuscript, it should necessarily be interpreted and understood as a numerical age. Singling out what type of age we are talking about as radiometric age is redundant and unnecessary since no other meaning is possible. The main aim of the manuscript is to dispute the (numerical) age of the JKB, therefore "Berriasian age" is meaningless and confusing with the aim of calibrating the age of a boundary. Certainly, the sentence suggested by the reviewer "The base of the Berriasian stage has been fixed for some years" is 100% correct, which one would correctly interpret as the base of the Berriasian has been fixed at the base of the Calpionella alpine Subzone and has been for many years. However, the sentence does not bear any relation to the numerical age of the base of the Berriasian, aka the JKB, which is the foremost purpose of this manuscript. As described in the introduction, there have been many (numerical) ages for the base of the Berriasian over the years, 135 Ma, 140 Ma, 144 Ma, 145 Ma. This represents a span of 10 Ma, which begs the question: What is the age of the JKB after all? Having an age of a boundary that is floating around a span of 10 Ma is less the ideal. Therefore, by any standards, the age of the JKB has been contentious over the past years. Sure, one could suggest it to be Berriasian, but this is too loose of a definition for the sake of numerical calibrating the geological timescale. Since 2005, the ICS has the JKB at ~145 Ma, which means approximately

145 Ma. From a geochronological perspective this far from ideal for ascribing a numerical age to a boundary. Admittedly, for many outside the field of geochronology such nuance bears no meaning, but for an accurate division of the geological timescale is it is imperative to find a more realistic age for the JKB, where geochronological data from many sections seem to converge to a similar age. We are confident we have demonstrated this in the manuscript. In short, geochronology is of the utmost importance to understand the rate of geological phenomena; for instance, duration magma magmatic processes, tectonic processes, duration of mass extinctions and recoveries. All this relies on the accurate and precise knowledge of the (numerical) age of rocks, paleontological markers, and stage boundaries. The latter two can only be resolved by using dating horizons that are close to boundaries using geochronological methods that are accurate and precise, which is methodology we have used in this manuscript.

COMMENT: 1.13) "JKB" is not standard terminology. It appears hundreds of times in the text. "J/K boundary" is the norm. Alternatives for use are: the base of the Alpina Subzone, base of Berriasian Stage, Tithonian/Berriasian boundary, or, less precisely, the J/K interval, the boundary interval. . Care is required is using the phrase J/K boundary.

REPLY: We feel that abbreviations can take any form, as long as it is clearly stated in the text and consistently used throughout. The use of the "J/K boundary" is just a personal preference of the reviewer as is our choice to use "JKB" just because something might be considered the norm hardly qualifies it to be mandatory. We see no problem with this abbreviation. However, if the reviewer or the Handling Editor feel adamant about this, we can certainly accommodate it since it is a frivolous matter and simple to adjust. We have deliberately chosen not to vary the term JKB with its many analogs to avoid confusion, especially to that reader that is not familiar with the various synonyms that the term JKB takes. Since we aim to draw attention from a broader audience, we feel that the term JKB should stay fixed for clarity, even though it might come across as repetitive.

COMMENT: 1.14) Anything that is not exactly correlated with the base of the Alpina Subzone can be said to be in the J/K interval, but not at the boundary. The reader is sometimes not sure what interval is referred to, or what horizon. Many times a fossil or date is somewhere in the J/K interval, but, to be accurate, nowhere near the actual boundary.

REPLY: We do understand that we are introducing a new concept (the JKB interval) to a field that already has a plethora of analog terms. However, we want to make it clear that the JKB interval is NOT a substitute for the JKB and the JKB is not the JKB interval. The idea for the JKB interval mainly stems from the fact that the age of the JKB in both sections do not overlap within our analytical uncertainty, and are offset by ~670 ka (± 335 ka). Furthermore, as pointed out by the reviewer #2 in comment 2.1, the markers are offset in an age which, in our opinion, only builds a stronger case to leave the age of the JKB confined to an interval, the JKB interval. Nevertheless, we will try to make a great effort to make this distinction very clear in the revised version of the manuscript. To supplement this reply, we refer the reviewer to comment 2.16 (i.e., in reply to reviewer #2, J. Pálfy).

Closing remarks by the authors

In summary, we thank the review from W. Wimbledon. We believe that many of the reviewer's comments, although very interesting, do not pertain or are beyond the limits of interpretation of our data. Some request we deem unnecessary because they would force the manuscript to steer away from the primary focus of the manuscript for no plausible reason. It was not clear to us the reasoning behind these comments. For instance, the reviewer's comments on magnetostratigraphy are incongruous with the focus of the manuscript. Dating magnetozones, laying down the assumptions about using the magnetostratigraphic scale as a timescale, It seems like the reviewer did not grasp the aims of the manuscript. Additionally, the comments about the revision of the biostratigraphy we consider to be unnecessary and would make the manuscript too long. Therefore, we disagree with the reviewer that substantial improvement is

required. Some comments on spelling, English misusage, and development of the prose are on point. And we thank W. Wimbledon for his thorough correction of these mistakes which will undoubtedly improve the readability of the revised version and make it much more precise. Nevertheless, these can be considered minor adjustments to the revised version. We welcome these suggestions and will take them into account for the revised manuscript. Finally, we confident that we have dealt with all of the reviewer's comments and suggestions appropriately. Hopefully, the manuscript can be approved for a revised version.

Reply to W. Wimbledon - Supplementary Comments

Page 1, line 28 This is geochronological jargon: "final" usually means the reported age. We will replace the word "final" with "reported age".

Page 2, line 1 to 5 The reviewer pointed to a problem with the construction of the sentence. What we want to imply is that reported ages from a previous publication are imprecise and they do not overlap, which means that they do not match, agree, or have the same age. That is what is implied by no overlap. The main difficulty in finding a (numerical) age for the JKB has been the choice for the base of the Berriasian. Of course, this has been solved, but back in the day when the first attempts to date the boundary (1985, 1995) this was still an issue, and this had significant implications towards the numerical age of the JKB. Maybe this has been clarified for the reviewer. "... and this level has been the most popular boundary marker for around 30 years". The Killian group in their 2014 report (Reboulet et al., 2014) still recommend the base of the Barriasella as a marker for the JKB. This might not be the case for the Berriasian Working Group, where 76% have chosen the base of the Calpionella alpine Subzone, but this is apparently not an overwhelming consensus within the entire community.

Page 2, line 13 What we are implying here is that the base of the JKB is assumed to the be the base of the Calpionella alpine subzone, not the ash bed. We will rephrase for clear meaning.

Page 2, line 16 "Recent years" will be deleted.

Page 2, line 19 We will rephrase it to "We also report new nannofossil results from Mazatepec section."

Page 2, line 22 See reply above on the usage of the word "age" in reply to comment 1.12.

Page 2, line 24,25 We will rephrase it to "which in turn also validates our age for the early Berriasian and the JKB."

Page 2, line 27 We will replace "JKB" for "boundary" to avoid repetition. We want to avoid the use of the other many synonyms for JKB to prevent any confusion.

Page 3, line 3 "a" replaced by "the"

Page 3, line 5 .. of the Eastern. . . will be added

Page 3, line 6 Replaced outcrops by exposed, since fossils do not crop out.

Page 3, line 8 Gray is the American spelling. See comments on spelling in reply to comment 1.11

Page 3, line 15-16 We disagree with the reviewer. The sentence is well constructed. Zircon does not need to be pluralized since it related to the behavior of the mineral zircon in general.

Page 3, 17 Will deleting the word dated as suggested, because of the precision of language.

Page 3, line 21 Mazatepec will be inserted instead of the vague term "the section in Mexico."

Page 3, line 30 The comma will be added.

Page 3, line 31 The definite article "the" will be before the noun R (as in the statistical package)

Page 4, line 3 "The section" will be replaced by "The Las Loicas section" as suggested

Page 4, line 4-5 "Found in the Las Loicas section" will be deleted for it was redundant, as pointed out by the reviewer.

Page 4, line 14 "(ca. 15m stratigraphic height)" is there to facilitate and aid the reader to locate the position in Figure 4.

Page 4, line 19 Bralower's thirty year old results must be seen as totally overtaken by more recent results, and to a lesser extent it is true of Casellato 2010. You quote Wimbledon 2017 which shows a more recent situation Please see reply to comment 1.4

Page 4, line 19 T, remanei and C. massutiniana are decidedly not typically Berriasian Please see reply to comment 1.3

Page 4, line 29 Magnetozones will replace Magnetochrons.

Page 4, line 30 No. This is very very vague. In numerous sections the base of the Alpina Subzone is proved in the middle of M19n.2n We will change from coincident to the middle of M19.2n be more precise.

Page 5, line 5 Rather unsafe. Authors present no evidence on Arroyo Loncoche. They cannot interpret what is or is not M19n.2n at LL, as they say. How can the authors' results be close to those of Inglesia Llanos when they have no magnetostratigraphy to present at Las Loicas and do not work on AL? Please see reply to comment 1.2 and also 2.9 (in reply to J. Pálfy)

Page 5, line 25 Again, surely this is obsolete work to cite? More up to date references required. The Italian data has been superceded. By the way, Ogg et al. 2016 is not original resesrch but a compilation The authors do not agree with Wimbledon (2017) where he said that N. kamptneri kamptneri and N. steinmannii steinmannii bioevents, previously used as infallible biozonal indicators in M17r, have been found widely in lower M18r and the upper half of M19n (Figs. 1, 2). Based on Wimbledon (2017) figure 2 only in Puerto Escano these bioevents are correlated with the upper part of C. alpina. (M19n). Besides, according to Svobodova and Kostak 2016 (cited by Wimbledon (2017) only in "one" sample they recognized this bioevent in M19n1r and other is correlated with M18r. The record of N. steinmanii steinmannii and the biozone NK1 are correlated with the Calpionella Zone without specifying the subzone and it is recorded nearly one meter above the acme of C. alpina.

Page 5, line 30 This does not match evidence from lots of sites N. steinmannii steinmannii is not a marker for the Elliptica Subzone, especially when it occurs as low as the Alpina Subzone. N. steinmannii steinmannii defines the base of the NK1 zone and nowhere in the text have the authors considered this marker as a bioevent of the Elliptica Subzone. The authors explain that it is found associated with this calpionellid in the Mazatepec section of Mexico, in the same way of other sections cited and that the NK1 Zone has been correlated in different section with the Elliptica biozone. The authors does not state that this bioevent is a marker of this calpionellid biozone. You quote Wimbledon 2017? Citation will be deleted

Page 6, line 9-11 We will try to rewrite to make it clearer.

Page 6, line 18 "loc" was supposed to be located

Page 6, line 19 Tethys regions will be replaced by Tethys Ocean

Page 6, line 23 We agree the sentence does not read well. What weant to imply is that the age of ash bed LY5 is an age in the Tithonian. We will rephrase to make it more clear.

Page 7, line 1 Perhaps the sentence would read better if stated: "Therefore, our new ages for the base of the Berriasian and the early Tithonian yield an excepted duration for Tithonian."

Page 7, line 1 How is it "recommended"???? Ogg is just another publication. And not an ICS publication. We will try to make a clearer distinction between Ogg et al. 2016

and the ICS.

Page 7, line 8 subsection title We feel that this is a great subsection title, it instigates the reader to pose the question: Do the ages presented here present the age of the boundary globally? Meaning, if we could measure the age of the JKB in every section, would we find the same age everywhere? Although this is impractical because not every section has table horizons close to the boundary, we argue for the fact that the Las Loicas and the Mazatepec agree favorably our ages can be considered as the age of the JKB globally. As a hypothetical, suppose that the age of the Las Loicas was 140 Ma and that the age of the Mazatepec was 143 Ma, then it would be hard to argue that their age agrees. However, they are off by 600 ka, which is a short interval.

Page 7, line 29 The reviewer says the FAD of R. asper is much older. How older is the FAD R. asper? Can he precise how much older?

Page 8, 1st paragraph "And yet for 200 years geologists have divided up the geological column quite successfully, with no magnetic markers and with no geochemistry, nd the bulk of agreed GSSPs do not rely on these. Replace this sentence?" High-precision geochronology has enabled the understanding of Earth processes in great detail. The time scales at which we deal in the manuscript are in the order of 50 ka, in which preservation of the paleontological markers becomes of extreme importance. We never suggested that GSSPs do not rely on secondary markers, but rather a valuable tool. Additionally, we draw W. Wimbledon's attention to the comments of J. Pálfy (reviewer #2), where he suggests that we should use our high-precision ages in both sections to show how problematic it can be to assume time-equivalency of biozones. We also share Pálfy view. Our data clearly shows a slight mismatch at the sub 100 ka level. In this scenario, the diachroneity of FAD and LAD's becomes evident and thus the dating of the stratigraphic record using high-precision U-Pb geochronology becomes a powerful tool in unraveling such nuances. It is undebatable that paleontology has been successful in dividing the geological timescale in the past. However, integrating geochronology, stratigraphy, paleontology, geochemistry, and magnetostratigraphy can

push the limits of correlations and calibrations of the geological timescale and is the best way forward. Perhaps, in the suggested sentence, we could state that in the context of calibrating the age of stage boundaries at the sub 100ka level, preservation of paleontological markers is an issue. Maybe this way it would be made clearer.

Page 8, line 5 meaning? one level but rest of sentence is about a set of biological events that took place across the Upper Tith-lower Berriasian interval We did not understand the reviewer's comment.

Page 8, line 7 what 'explosions'? bloom of small C alpina? It comed after diversification of nannoconids We meant the bloom of small Calpionell alpine

Page 8, line 19-20 Vague, no justification shown Through out the paragraph we cite publications to support this last sentence.

Page 8, line 24 Its proper name is the "International Chronostratigraphuc Chart"? Will make the modification to "International Chronostratigraphic Chart".

Page 8, line 24 meaning? It is beyond the scope of the manuscript to go into detail on the issue of offset between Ar-Ar and U-Pb ages. However, it is an important statement to be made from a geochronological perspective.

Page 8, line 31 it is a hole in the sea bed, there is no section We will refer to it as core, as was done previously in the paragraph, instead of section.

Page 8, line 31 vague We will incorporate examples of the JKB markers in the sentence, even though at this point in the manuscript we are deep into the discussion and have stated and cited what the markers are and expect the reader to be following along.

Page 9, line 1 As a concluding sentence it is not effective. It says, more or less, our age agrees with othe ages. Not a very weighty ending We disagree, this last sentence sums up that our age agrees with more recent ages for the JKB, and can be considered the age of the JKB globally. The sentence, in our opinion, is actually quite important sentence and carries a lot of weight. No other study dealing with the age of the JKB

could make such a big claim.

Page 9, line 3 - comment of the title of section 5 – Conclusions and Summary Cretaceous rock/time is base Berriasian stage and start Berriasian age. What you discuss is geochronology and radiometic dates We are not sure what the reviewer meant by this comment. Not very clear.

Page 9, line 8 what interval, you just presented numbers The reviewer missed the point of the the meaning of the JKB interval. We talk about the JKB interval previously in the manuscript. For clarification with regards to the JKB interval, we refer to comment 2.16.

Page 9, line 10-11 This ammonite biozone is enormously long, what can it bracket or corroborate? Precision? We did not imply that we could bracket anything using an ammonite zones. Our bracketed interval is the JKB interval, which is bracketed with U-Pb ages which is staed in the manuscript.

Comments on Figures

Page 17, Figure 2 The main aim of figure 2 is to display our U-Pb data. The biozones are displayed merely conjecturally, since the exact age and duration of the biozones are not known. Therefore, adding boundaries to the biozones would be unrealistic and wrong. Spelling will be rectified.

Page 19, Figure 4 Spelling of species names will be rectified. Boundary abbreviations have been commented on previously in this reply. Additionally, abbreviations were adopted to make the figures more clear, less clustered, and easier to read.

We have applied U-Pb zircon CA-ID-TIMS dating techniques to single zircon grains, which yields $^{206}$Pb/$^{238}$U dates at

15  0.1-0.05% precision. The depositional age of ash beds has been calculated from the weighted means of the three to six youngest overlapping $^{206}$Pb/$^{238}$U dates (Fig. 2), This assumes that.... [Ok] record prolonged residence of zircon zircons magmatic systems as well as intramagmatic recycling. In the text, all quoted ages for the dated ash beds language precision - you dont record ages for the und... [Ok] $^{206}$Pb/$^{238}$U ages corrected for initial $^{230}$Th disequilibrium. A detailed description of the techniques for sample preparation, laboratory procedures, data acquisition, as well as data treatment are provided in the Supplementary Materials. The full U-Pb

20  data set is reported in Table S1.

The nannofossil biostratigraphy for the Mexican section for Mazatapec [Ok] ...ples from the Pimienta and Tamaulipas formations. For detailed calcareous nannofossil examination, simple smear slides were prepared using standard procedures (Edwards, 1963). Observations and photographs were taken using a polarizing microscope Leica DMLP with increased 1000X and accessories such as λ one sheet of plaster and blue filter. The slides are deposited in the Repository of

25  Paleontology, Department of Geological Sciences, University of Buenos Aires, under the catalog catalogue BAFC-NP: Nº 4190-4206. Optical images of selected species are shown in Fig. 4; the distribution chart for the calcareous nannofossil species is presented in supplementary Fig. 3. Please se reply on American versus British spelling

The age of the various paleontological palaeontological as the age of JKB in the Las Loicas, have been modeled modelled [Ok] using the Bayesian age-depth model Bchron of Haslett and Parnell (2008) and Parnell et al. (2008). The age-depth model This model

30  resulting uncertainty envelope is presented in Fig. 4A. The age-depth results are reported in TS.2 comma [Ok] ...ned to every meter metre ...igraphic height. The Bchron code used in in the R cal package environment (R Core Team 2013) is included in the Supplementary Materials. [Ok]

**4. Results and discussion**

[revised manuscript text omitted]

[Annotation: Rather unsafe. Authors present no evidence on Arroyo Loncoche. They cannot interpret what ... at LL, as they say. How can the authors' results ... of Inglesia Llanos when they have no magnetostratigraphy to present at Las Loicas and do not work on AL?]

**4.2 The age of the Jurassic/Cretaceous Bou...**

The Mexican [delete] section has a dense and well-established calpionellid zonation with close ties [=like that of] classical western Tethys zonation (López-Martínez et al., 2013b) (Fig. 4B). The nannofossil assemblages recognized in the Mazatepec section exhibit low diversity compared to contemporary associations of the Tethyan realm and a relatively poor [=compared to Tethys] degree of preservation of the nannofossils, which are charaterised... to heavy dissolution etching (Fig. 3). At stratigraphic height ~16 m (bed MTZ-65; López-Martínez et al., 2013b), 18 nannofossil species have been recognized (Fig. 3): the heterococcoliths are mostly represented by Watznaueriaceae including *Watznaueria barnesae, W. britannica, W. manivitae, Cyclagelosphaera marrgerelii, and C. deflandrei; Zeugrhabdotus embergeri* is another frequent constituent. The nannoliths are represented by *Conusphaera mexicana, Polycostella senaria, Hexalithus noeliae, Nannoconus globulus* and *N. kamptneri minor*. These nannofossils are indicative of a late Tithonian-early Berriasian age in the [=indicate a late T to early B age for the] part of the Tampaulipas Formation. The assemblage composed by *Conusphaera mexicana, Polycostella senaria* and *Hexalithus noeliae*, indicates a late Tithonian age. The only useful biological event recognized is the FAD of *N. kamptneri minor* [This is rather late/high, compared to Tethys?] 5 m above the base of the Alpina Subzone in the Berriasian. [delete] [with]

At stratigraphic height ca. 25m an increase in the diversity of nannofossils is identified, reaching 13 sp... (bed [spelling] MZT-87 sample). Among the nannofossils, the presence of *N. steinmanni steinmanni* stands out, a marker also used to define the base of the first biozone of the Berriasian (NK1 ... DSDP 534, Colme di Vignola Bosso and Foza with magne... [More up to date references required. The Italian data has been superceded. By the way, Ogg et al. 2016 is not original research but a compilation] ...Channell et al., 2010) as well as the Elliptica Subzone (Schnabl et al., ... nofossil datums with magnetostratigraphy has been a very useful development (e.g., Channell et al., 2010), although the integration of nannofossils with calpionellids ranges has been less exploited. Noteworthy is the correlation between NK1 and the Ellipitica [spelling] Subzone recognized here in Mazatepec which also coincides with the previously established relationship between these biozones in the ... 2010). Unfortunately, the presence of *N. steinmanni minor* or *N. wintereri* (Wim... [This does not match evidence from lots of sites / N. steinmannii steinmannii is not a marker for the Elliptica Subzone, especially when it occurs as low as the Alpi... / You quote Wimbledon 20...] azatepec section. However, it is reasonable to assume that both

of these markers would be close to the base of the Alpina Zone since the FAD *N. steinmanni* is only 5 m above the base of the Alpina Zone. Therefore, the relative age of the palaeontological markers in the Mazatepec section is in full agreement with the working model of Wimbledon (2017) for the JKB.

To constrain the age of the JKB in the Mazatepec section, we have dated the ash bed in bed 81 which is located within the Elliptica Subzone and stratigraphically 10.1m above the base of the Alpina Subzone (Bed MTZ-45 Fig. SC), i.e., JKB (López-Martínez et al., 2013b) (Fig. 4B). The age of ash bed MZT-81 is 140.512 ± 0.036Ma (Fig.2). Unfortunately, in the Mazatepec section ash beds are scarce. Therefore, it was not possible to bracket the age of the JKB, as was the case in the Las Loicas section. Consequently, to estimate the age of the boundary, we have to resort to assumed sedimentation rates to back-calculate the age of the JKB. Since the sedimentation rate in the Pimienta and Tampaulipas formations is unknown, we use both high and low sedimentation rate because this takes into account our conjectural knowledge of the sedimentation rate in the Pimienta and Tampaulipas formations. Here we assume a low sedimentation rate to be 2.5 cm/ka and a high sedimentation rate to be 4.5 cm/ka. Therefore, the age of the JKB is estimated to be 140.7 Ma and 140.9 Ma, respectively.

**4.3 The early Tithonian and the base of the Vaca Muerta Formation**

The base of the Vaca Muerta Formation contains a well-established early Tithonian ammonite assemblage of the *Virgatosphinctes andesensis* Zone (Riccardi, 2008, 2015; Vennari, 2016). Fortunately, the gradational contact between the Vaca Muerta and the Tordillo formations is very well exposed in the La Yasera section and contains ash beds very close to the contact (Fig. SB). We have dated an ash bed (LY-5) located at the contact and it yielded an age of 147.112 ± 0.078 Ma (Fig. 4C). The ash bed is located in the Tordillo Fm, 1.5m below the contact with the Vaca Muerta Formation, thus very close to the *Virgatosphinctes andesensis* Zone depending on the nature of the contact the Darwini Zone Tethys was an ocean not a 
[revised manuscript text omitted]
 [...] occurrences. Additionally, the [...] nannofossils considered to be markers (Wimbledon, 2017) and lack[...] markers. These facts collectively render [...] section biostratigraphically [...] ds to the JKB markers. In closing, we feel that the results

presented in this study are in good agreement with several other studies of the age of the JKB and thus it allows our bracketed interval to be considered as the age of the JKB globally.

> *As a concluding sentence it is not effective. It says, more or less, our age agrees with other ages. Not a very weighty ending*

> *Cretaceous rock/time is base Berriasian stage and start Berriasian age. What you discuss is geochronology and radiometric dates*

The age of the JKB has been contentious for the past decades with a spread of ages of ~10 Ma with varying approaches and geochronological methods being employed. Recent developments in high-precision U-Pb geochronology have proven to be a powerful tool in dating the stratigraphic record, allowing and allowing the accurate ...cation of stage boundaries. We have constrained the age of the JKB to an interval 40.9-140.7 Ma by dating two independent sections that span the JKB using high-precision U-Pb geochronology. This interval is supported by ammonite zonation, calcareous nannofossil, and calpionellid as well as in both sections. We consider the magnetochron M19n.2n at Arroyo Lonconche 7) as the most important secondary marker for the JKB, which has been shown to be within the late Tithonian *Substeueroceras koeneni* in the Neuquén Basin, close en... especially when the relative age between the various markers for the boundary is still not fully resolved. The agreement between high-precision U-Pb ages and the various markers for the boundary in both sections allows us to contest the current age for the JKB in the TSISC 2016 of 145.5 ± 0.8 Ma. Additionally, our radiometric age *tosphinctes andesensis* Zone, close to the Kimmeridgian-Tithonian Boundary, is in agreement with recent estimates for the age of the CM22An polarity interval ...interval. This preserves ,,, ...ation of ~7 Ma for the Tithonian and thus corroborate our ages for the JKB. In conclusion, we consider our results for the JKB to carry a global significance and should be viewed as a positive step forward in resolving the age of the JKB.

> *before the numbers not at the end of sentence*
> *what interval, you just prese...*
> *This ammonite ...usly long, what can it bra... ? Precision?  Please see reply*
> *If this OGG et al it is not the official ICS timescale*

[revised manuscript text omitted]

**Late Jurassic - Early Cretaceous disposition of continents**

[Figure]

[Figure]

Berriasian - spelling. There are no limits for any of the biozones. How can they be related to the dates?

Please see reply

**Figure 3**

[Figure]

*C. mexicana*

*C. mexicana*

*H. noeliae*

*H. geometricus*

*N. kamptneri minor*

*N. globulus*

*N. st. steinmannii*

*N. st. steinmannii*

*P. senaria*

*P. senaria*

*U. granulosa*

*E. primus*

*R. asper*

*R. asper*

*N. kamptneri minor*

*N. wintereri*

**Figure 4**

[Figure]

JKB as in the text. J/K boundary or Tithonian/ Berriasian boundary
Species names should not have a calital letter

OK

---

## Author Response (AR1)

**Report on the revised version of the manuscript: "Cross-continental age calibration of the Jurassic/Cretaceous boundary"**

Dear Handling Editor Silvia Gardin,

**Structure of the Report on the revised manuscript version**

Firstly, we present a bullet point list of all the relevant modifications made to the manuscript we felt would improve the manuscript. Other comments by the reviewers that we disagree or felt were inadequate where left out and we await the decision of the Editor and Handling Editor. Secondly, we present a point-by-point modifications relative to the reviewers comments structured as follows: (1)reviewer comment; (2) author reply; (3) author modification, starting with comments from reviewer #1 W. Wimbledon and subsequently reviewer #2 J. Pálfy. Reviewer's comments are in *italic blue* font, our reply in **black** regular font, and modification as in green regular font respectively. A marked up version of the manuscript to better aid the Editors and reviewers in proof the revised version. The revised Supplementary Material is also in this report. All modifications to the manuscript and Supplementary Material are high-lighted in green.

**Main modifications**

1. We have addressed the comment 2.13 which suggested that we compare out estimates for the duration for the Tithonian with that of the independent duration estimates of the Pacific M sequence of magnetic anomalies of Malinverno et al. (2012). This was incorporated into section 4.3.

2. The affirmation of that in p.6. 25 that there was a formal definition of the Kimmeridgian and
Tithonian boundary was removed as suggested in comment 2.11

3.  We have incorporated the suggestion by reviewer J. Pálfy in his comments 2.1 and 2.14 (see author's reply). As the reviewer suggested, we have refocused part of our discussion (section 4.5) to the embrace the mismatch between the ages of the JKB in the dated sections and discuss the pitfalls of regional and global biostratigraphical correlation. To accommodate the reviewers suggestion in comment 2.1 and 2.14 we have completely rewritten section 4.5.

4.  In the interest of addressing the comments 2.9 and 1.2 which questioned the magnetostratigraphic correlation between Las Loicas and Arroyo Loconche, we have completely removed the correlation between both sections, since both reviewers took issues and we felt better to remove it from the manuscript.

5.  Comment 2.15 was also addressed and incorporated into the *new* section 4.5

6.  Comments 2.16 and 1.14. The concept of the JK interval was rephrased and explained the reasoning behind and can be found in section 4.5.

7.  We have added a 4.6 section entitled: "A case for a younger J/K boundary age" where we specifically address our data and data from other publications that show that the age of the JKB is much younger than the one found in the ICS and the fragility of the biostratigraphy under which the ICS JKB age is grounded on.

8.  We have written the section 5. Summary and conclusions to accommodate the new refocusing of sections 4.5 and 4.6

9.  Figure 1: Title was modified; "distribution of continents" was replaced by "global paleogeography". We did not use a different base map, since no justification from the reviewer mas made as to why the maps map should be changed. Nor a suggestion as of which base map should be used. Migratory routes were left in the map because we do suggest that the rate migratory routes could be a possible explanation to the difference in age between Las Loicas and Mazatepec. Section that were not in the study were left in to convey the idea that the these section are contemporaneous.

10. Figure 2: Berriasian spelling was corrected. Fonts were made smaller, as requested; number of grain in LL10 was changed to 4 instead of 6.

11. Figure 3: We have not removed taxon names since we see this is a common practice in many publications. Additionally, the caption figure was rewritten to accommodate reviewer #2 requets to have the taxon names in the figures caption.

12. Figure 4: Names of species were corrected as pointed out. Some species names were uncapitalized and steinmannii spelling was corrected.  A dashed green line was put in to connect and high-light the mismatch between Las Loicas and Mazapetec. Stage names were added to every section, i.e., Tithonian, Berriasian. References were put at the top of each panel to make clear what is being cited and what is not, although also in the caption figure, to address comment 2.3. Arroyo Loconche section from Iglesias Llanos et al., (2017) was removed from the figure as well as the manuscript and the templates were renamed accordingly throughout the manuscript.

13. Renaming of the Figure 4 was were carried out throughout the manuscript to coincide with the new figure 4

14. Supplementary Material: We have added the calcareous nannofossil chart to the Supplementary Material, now named F.S1, as pointed out in comment 2.5.

**Modifications to the manuscript with respect to the comments by reviewer W. Wimbledon (reviewer #1) on the manuscript "Cross-continental age calibration of the Jurassic/Cretaceous boundary"**

**General Stratigraphic remarks – Magnetostratigraphy**

*1.1)   The text should perhaps say that there is no possibility of magnetic calibration of Las Loicas with the many Tethyan sites where it has been documented.*

A subset of the authors are pursuing the magnetic calibration in Las Loicas. The preliminary sampling has already been done, and some results are available yet not published. The main obstacle is that the basinal facies in Las Loicas which makes it difficult to have a dense suitable sampling for magnetostratigraphy. Therefore, efforts are being made to overcome this issue. Please see reply to comment 1.2 and 2.9 (i.e., in reply to reviewer #2 J. Pálfy) for further clarification on our attempt to correlate the magnetostratigraphic data from Arroyo Loconche with Las Loicas.

MODIFICATION: No modification was made since we feel we have answered this comment.

   *1.2)    the ammonite zonations applied at the LL and AL do not agree – a big problem.*

         It needs to be made clear that we do not present any new magnetostratigraphic data, but instead use the magnetostratigraphy of Iglesias 2017 to aid marginally and back-up our age of the JKB in Las

Loicas. Magnetostratigraphy is not the focus of the paper nor did we state in the manuscript that we aimed to do that, but rather an aside. Meaning, we use it as a reflection on how other substantial evidence for the JKB from the Neuquen Basin might agree with our data. We recognize that magnetostratigraphy is a significant component in calibrating and locating the JKB in sections that span the JKB. However, we are fully aware of the seemingly conflicting evidence from the ammonite zonation from Arroyo Loncoche and Las Loicas. We clearly stated that the ammonite zonation in Arroyo Loncoche is preliminary and also cited a discussion around the matter in López-Martínez et al. (2018). The main point discussed in López-Martínez et al. (2018) is that both discussions contain the different resolution of data. The ammonite biostratigraphy of Las Loicas is based on the bed by bed collection from 54 fossiliferous levels with 450 ammonite specimens. López-Martínez et al. (2017 Fig.

1) and Vennari et al. (2014) recorded 35 fossiliferous levels and studied 228 ammonite specimens. Therefore, we feel that the ammonite zonations in Las Loicas is well-defined and described.

         On the other hand, in the Arroyo Loncoche region there is not a single published section with the ammonite levels, or the number of specimens collected, which renders the definition of the biozones unreliable. It is also evident that the boundaries of the biozones in Arroyo Loncoche have been changing along the years, as well as unit thickness, the presence of sills, etc. We invite the reviewer to take a closer look at the discussion in Lopez-Martinez et al. 2018 but include here an extraction from the paper to illustrate the issue. Iglesia Llanos et al. (2017), p. 194 state that "*The boundary between ammonite zones in Arroyo Loncoche was placed according to the first occurrence of the index species*." However, the range chart with vertical distribution of the taxa (their Fig. 2) and the ammonite biozones do not follow this criterion. For instance, the base of the Corongoceras alternans zone is placed at the first occurrence of *Corongoceras* sp. and the index species is not even recorded in this section. Furthermore, the base of the Substeueroceras koeneni zone is placed on the first occurrence of *Substeueroceras* sp. (at 150m of the base of the section) while the index species appears higher (above 180 m). This more than 30m discrepancy explains the different biozonation of the same section published by Kietzmann et al. (2011 Fig. 3) where they placed the base of the Substeueroceras koeneni zone at 190 m of the base of the Arroyo Loncoche section. **Lastly, it is important to point out that in the absence of a reliable biostratigraphic framework, such as the case of Arroyo Loconche, magnetostratigraphy is just a floating scale.**

In conclusion, the paper aims to calibrate the numerical age of the JKB using high-precision geochronology in Las Loicas and Mazatepec using the base on the base of the Calpionella alpine zone as the primary marker for the JKB. In Figure 4, we correlate the JKB Arroyo Loconche and Las Loicas based on more compelling evidence for the JKB which is the M19.2n (Arroyo Loconche) and the base of the Calpionella alpine Subzone (Las Loicas). Therefore, we avoided normalizing the two sections based on ammonite. Incidentally, in his 2017 review of the JKB, W. Wimbledon (Wimbledon, 2017) has also normalized Las Loicas and Arroyo Loncoche disregarding the apparent mismatch of the ammonite zones in the working model for correlating the regions for the JKB. Furthermore, in our Figure 4, the correlation between the M19.2n in Arroyo Loncoche and the base of the Calpionella alpine subzone in Las Loicas is a dashed red line, which suggests that the correlation is merely conjectural. As W. Wimbledon pointed out, Phanerozoic stage boundaries are not dependent on geochemistry, magnetostratigraphy or geochronology. These are just tools used to aid the calibration of stage boundaries, and mismatches are commonplace. Therefore, we do not see this first point as a big problem. The reviewer #2, J. Pálfy, also took an issue with this matter. We kindly ask the reviewer to also read the reply on comment 2.9 (i.e., in reply reviewer #2). Hopefully, it will supplement this reply
and vice-versa.

MODIFICATION: We have removed the correlation of the magnetostratigraphy within the Neuquen
Basin between Las Loicas and Arroyo Loconche since both reviewers took issue with our approach, and
we acknowledge that there are many issues with the ammonite zonation in Iglesias Llanos et al., (2017)

     *1.3)    The calpionellid assemblage noted at Las Loicas is anomalous: such a mixed
         assemblage (with apparently derived Tithonian calpionellids) does not define or mark
         the base of the Berriasian. It should be made clear what is definitively lower Berriasian*
  *and what is not.*

    First, it is essential to recognize that sections containing datable horizons close to boundaries,
such as the Las Loicas and Mazatepec, are extremely rare which is a significant hindrance in calibrating
the age of stage boundaries in general. Tethyan and Mediterranean sections do not contain datable
horizons, because these sections are deposited in passive margins far from plate tectonic boundaries
where a considerable amount of acidic-aerial volcanism output is produced allowing for the deposition
of ash fall deposits (ash beds). Therefore, even though the issues surrounding the JKB have been
concentrated in the Tethys region, its age, on the other hand, will not. Although the reviewer claims that
the Las Loicas contains "anomalous" calpionellid assemblages, if we are ever to advance in the
knowledge of the numerical age of the JKB, we have to use everything at our disposal.

Replying to the reviewer's comments, the only reported "anomalies" in Las Loicas are a) he
presence of *Tintinnopsella remanei* in the upper part of the Crassicollaria Zone. This is a none typical
appearance in the Mediterranean Tethys, but usual in western Tethys as discussed in López-Martínez et
al., (2017), and b) the record of *Crassicollaria massutiniana* in the lowermost part of the Alpina
Subzone. Even when it can be unusual the presence of this species in the Lowermost Berriasian, this
does not affect the biozonation scheme as the Alpina Subzone is defined by the acme of *Calpionella
alpina* small and globular form and not the Last Occurrence of any species. Then, the Alpina Subzone is defined in the same way as in the Mediterranean Tethys and can be used as a marker of the JKB in Las Loicas.

MODIFICATION: No modification was made since we feel we have answered this comment.

*1.4)     The nannofossil literature cited as the justification for some of the text's discussion and conclusions is rather old - Bralower and Casellato references are now 10 -30 years old. Many Tethyan sites have since been documented, and that make some of the species FADS and the zones discussed obsolete. Some Italian localities cited in the text are seen as anomalous in the positions of their nannofossil FADs. Thus it is not clear why these*

*localities are selected by the authors for comparison with the LL and M sites, especially when they are not the best/most representative.*

We, unfortunately, have to disagree with this comment. There are two standard calcareous nannofossils zonations for the studied interval. (Bralower et al., 1989) proposed a calcareous nannofossil zonation for the Jurassic and Cretaceous based on southern European land sections and the western North Atlantic, DSDP Sites 391C and 534A. (Casellato, 2010) proposed a new calcareous nannofossil biostratigraphic scheme for the Tithonian–Early Berriasian established for the Southern Alps in Northern Italy. Even though many recent papers deal with nannofossils of this time interval, there are no new zonations for this interval. Therefore, these two papers form the basis for newer studies, with many of the recent publications still citing the zonation in the classic papers of Bralower et al. (1989) and Casellato (2010). We agree with the reviewer that these publications might be considered old, but in no way, shape, or form, can they be considered outdated or overtaken since the zonation presented in them form the basis of the more recent works on calcareous nannofossils zonation of this period.

To illustrate, we take the liberty of copying below excerpts from the newer publications on calcareous nannofossil of this period that promptly cite the work of Bralower et al. (1989) and Casellato (2010), as we have.

a) Grabowski et al., (2017). Sedimentary Geology 360, p. 57, state:

"*For biostratigraphic purposes, the available biostratigraphic schemes of Bralower et al. (1989), Bown and Cooper (1998) and Casellato (2010) were considered. The latter was selected to apply for the Lókút section, as the most appropriate for nannofossil record in this Tethys location*".

b) Hoedemaeker et al. 2016. Revue de Paleobiologie 35, p. 190, state:

"*CALCAREOUS NANNOFOSSILS (C. E. Casellato and S. Gardin)…. Calcareous nannofossils are rare to common and poorly to well preserved, with overgrowth more pervasive than etching. Assemblages are of Tethyan affinity ……(and) the biostratigraphic schemes adopted in this study are those of Bralower et al. (1989) and Casellato (2010)….*"

c) Ogg et al., (2012). A Concise Time Scale, p. 170, state:

Use for defining the JK boundary in the Mediterranean Tethys similar FOs of calcareous nannofossils as those used in our paper (fig. 13.2.).

d) Schnabl et al., (2015). Geologica Carpathica 66, p. 491, state:

"*For several generations, apart from occasional aberrations, definitions of a J/K boundary have*

*focused on one interval, between the base and top of one ammonite subzone (that of Berriasella jacobi), and, in the last thirty years, more and more, on the widespread and more consistently recognized turnover from Crassicollaria assemblages to small Calpionella…. Latterly this has been widely reinforced by the use of calcareous nannofossil FADs (references in Casellato 2010).*"

e) Sbodova and Kotsak 2016. Geologica Carpathica 67, p. 225 state:

"*Biostratigraphic data were interpreted with reference to the nannofossil zonation of Casellato (2010), commonly used for the Upper Jurassic and the Lower Cretaceous in the Tethyan/Mediterranean area*". Regarding some anomalous positions of the nannofossils.  p. 231. "*It should be noted, that the LO of N. kamptneri minor usually appears a little above the LO of N. steinmannii minor, but in this paper it occurs together with the LOs of N. steinmannii steinmannii and N. kamptneri kampteri in bed 35. This*

*anomaly can be explained by the very poor preservation and extreme etching of calcareous*

*nannofossils between beds 32 and 34. Moreover, the appearance of these four species together suggests the presence of a hiatus."*

f) Bakhmutov et al., (2018), Geological Quarterly, 62 p. 232, state:

*"The first appearances of species of significant calcareous nannofossils at Theodosia are shown in*
*Figure 23. The appearances are not consistently equivalent to all records in western Tethys (Casellato, 2010; Schnabl et al., 2015), one reason being that in this preliminary study we did not sample beds below a level we believe to be assignable to the lower to middle part of M19n.2n. ……. However, the FADs in M19n of H. strictus, C. cuvillieri, N. wintereri, N. steinmannii minor and N.kamptneri minor appear to be consistent with other regions."*

MODIFICATION: No modification was made since we feel we have answered this comment.

*1.5)    The dating of the magnetozones needs to highlight and discussed at more length in the Discussion.*

Very confusing comment. Also on magnetostratigraphy, the reviewer comments on the dating of magnetozones. Nowhere in the manuscript, it is stated that we have dated magnetozones, or implied doing so. This is clearly beyond the scope of the manuscript, and additionally, not even possible since we do not present any magnetostratigraphic new data in any of the studied sections. To date magnetozones, one would have to present magnetostratigraphic and geochronological data on the same
section, which we have not done nor said we had. It would not be scientifically sound to do otherwise. Therefore, we feel the request of the reviewer is rather odd and unjustified.

MODIFICATION: No modification was made since we feel we have answered this comment. This study is a detailed study, at the 100 ka level. Gross stratigraphical correlations are not possible in this level of detail. This is the importance of high-precision geochronology. Correlations that were once
possible in the past are no longer the case, because out understand if the geological time and stratigraphic record has and will improve dramatically due to this technique. Time equivalency of boundaries and markers need to be proven and not taken as a foregone conclusion. Our manuscript depicts this issue very clearly.

*1.6)    Also, the assumptions (as seen in most publications) about using the magnetostratigraphic scale as a time scale could be laid out fully in the Introduction*

The reviewer also asks us to discuss the assumptions of using magnetostratigraphy, which again, is not the aim of the paper since we do not present any magnetostratigraphic data whatsoever. Therefore, to discourse about the use of magnetostratigraphy is beyond the scope of the dataset we 10   present not to mention beyond the point of the problem we are trying to solve and the methods we use. Again, an odd comment from the reviewer.

MODIFICATION: No modification was made since we feel we have answered this comment.

*1.7)    Notably, Ogg et al 2016 is not at all 'official' and is not attributable to ICS, but this is* 15   *not clear from the text.*

The reviewer does have a point and will make the distinction clearer between the ICS and Ogg 2016.

MODIFICATION: We feel this has been made clearer in the revised version.

**Structure**

*1.8)    The chronostratigraphic and biostratigraphic background should be made clear before consideration of any new data on radiometric dates.*

The issues and intricacies with fixating the JKB are well-documented in several publications which W. Wimbledon himself has authored and co-authored. The biostratigraphy in both sections we use has also been well-documented in other publications by some of the authors in this manuscript. Therefore, we feel that this request is somewhat unnecessary since the issue has been dealt with quite thoroughly in other publications and have been cited throughout the manuscript when necessary. Furthermore, we have taken Solid Earth's recommendation on the manuscript type, where manuscripts should be short, concise, and to the point which is a trend among high-impact journals. We feel that reviewing the biostratigraphic framework would make the manuscript unnecessarily long. In closing, the manuscript is not dedicated to reviewing any previous data, but rather presenting new data and building on pre-existing biostratigraphy.

MODIFICATION: No modification was made since we feel we have answered this comment.

*1.9)*    *The extrapolation of the GMPTS to onshore localities is central. The paper is concerned with attaching radiometric dates to a biostratigraphic framework. But it says very little about how radiometric dates match the timescale used by, for instance, Gradstein et al 2012: a time framework linked to the oceanic magnetostratigraphic record, the GMPTS.*

*The last is hardly mentioned.*

Extrapolating the GMPTS would be over-interpreting our data and certainly beyond the scope of the manuscript. Nevertheless, we feel this is the main problem that the International Chronostratigraphic Chart of International Commission on Stratigraphy (2005 to 2018 versions) is facing. The 145 Ma age for the JKB boundary is based on the Shatsky Rise magnetozones. The main drawback for this assertion is the accepted age for the base of the Berriasian, supported by a poorly dated age of the Shatsky Rise in the Pacific Ocean. The radiometric Ar-Ar dating, even if only the best two samples were considered, have reduced plateaux that could indicate some [39]Ar recoil (Mahoney et al., 2005). The ages of these samples are 144.8 ± 1.2, 143.7 ± 3.0, and 142.2 ± 5.3 Ma, but 145 Ma was the preferred age as it coincides with the spreading rate assumed for this part of the Pacific Ocean floor (Ogg et al., 2012). Besides, when the biostratigraphic controls of the sediments of the Shatsky Rise intruded by the dated sills are taken in consideration, the results are not very well constrained (Mahoney et al. 2005 cited (Bown, 2005)):

Quoting Bown (2005) :

**"***Zone NK1; Berriasian (Site 1213): The Jurassic/Cretaceous boundary interval zonation of Bralower et*
*al. (1989) is based on a distinctive succession of nannolith appearances, notably Conusphaera and Nannoconus; however, these taxa were absent in this part of the section and the former was absent throughout. In addition, a number of important marker species of the family Cretarhabdaceae (C. cuvillieri, R. angustiforata, and Retecapsa octofenestrata) and genus Eiffellithus (E. primus, E. windii, and E. striatus), although present, are rare and restricted to a*
*small number of samples, and their first and last occurrences may not be biostratigraphically reliable. The lowest Cretaceous zones are thus identified using marker species, where present, together with alternative datum events and aspects of the entire assemblages.*

*The lowermost productive samples (Core 198-1213B-27R) yielded H. chiastia, L. carniolensis, Tubodiscus bellii, and R. laffittei, indicating Subzone NJKc or younger. The nannofossils*
*do not unambiguously indicate a Cretaceous age, but correlation with Zone NK1 is inferred based on the presence of the genus Tubodiscus and absence of R. angustiforata, P. fenestrata, and R. wisei (Bralower et al., 1989). Support for this interpretation also comes from radiolarian fauna that also indicate a Berriasian age for the lowermost cores (H. Kano, pers. comm., 2003)"*.

In conclusion, Bown (2005) makes it clear that the nannofossils **do not unambiguously**
**indicate a Cretaceous** age but also the **inferred** correlation with zone NK1 is based in the **presence of ONE GENUS and the ABSENCE of three species** (negative evidence!). There are not markers, no first appearances, etc. Regarding the radiolarians, the data are based only in personal communication. The facts stated above discredit the JKB extrapolation of the Global Magnetic Polarity Time Scale (GMTPS) based on the Shatsky Rise to onshore localities currently in use. In any case, it is not the
focus of this paper to criticize the age of the magnetic polarity time scale used to define the boundary, which needs a deep revision in our opinion.

MODIFICATION: No modification was made since we feel we have answered this comment. Gross stratigraphical correlations are not possible in this level of detail. Please see modification comment to 1.5.

*1.10)   The core of the paper could usefully be a careful examination of the calibration of the dated ash horizons and the levels with the key biostratigraphic markers – listing them in sequence, level by level.*

        The detailed biostratigraphy of Las Loicas with an indication of the fossiliferous levels and relevant markers is indicated in Vennari et al., (2014) and López-Martínez et al. (2017).  The objective
of this paper is to date this biostratigraphy with accurate and precise ages in the relevant interbedded tuffs coupled with age-depth modeling. In the Mazatepec section, biostratigraphy based in calpionellids in found López-Martínez et al. (2013) is complemented by new calcareous nannofossil occurrences which are presented here and presented in figures and pictures.

        MODIFICATION: No modification was made since we feel we have answered this comment.

**Precision, accuracy, English Language**

        *1.11)   There are numerous examples of rather problematic phrases and sentences which are not written in good English. But more critical is the lack of precision or looseness in language and terminology. This lets down the submission very badly. It is the thing that*
*needs the most attention in a revision by the authors.*

        We do admit that many of W. Wimbledon's suggestions on our English usage and grammar (or lack thereof) are correct and we welcome them. We incorporate all the suggested words, variations and rewrite all sentences pointed out that remain unclear and confusing. Names of species will be thoroughly revised. Specific replies to comments on the supplementary section are found below.

MODIFICATION: Reviewer's suggestion on English usage was made and can be found in the high-lighted revised version at the end of this report.

Corrections on some spelling mistakes are just differences between American and British English. Spellings such as gray, meter, catalog, paleontological, memorize, analog, analyze, defense, color, aging, inquiry, license among many other words are a correct and legitimate form of spelling in American English. Therefore, since Solid Earth does not dictate which kind of English is to be used in their publications, we chose to use American spelling. Furthermore, we feel we were consistent with our choice of spelling thought out the manuscript. Therefore, we have decided to disregard the reviewer's comments on these spelling mistakes.

MODIFICATION: No modification was made since we feel we have answered this comment.

*1.12)* *The loose wording of the Abstract's and Introduction's first sentences. No, the age of the J/K boundary is very clear. Lena et al. talk only about radiometric dating. They should say that the start of Berriasian age/base of the Berriasian stage has been more or less fixed for some years [the authors actually quote several relevant papers that show this]*

This comment is quite confusing, and we are not sure what the reviewer meant by this. We hypothesize it might have to do with how different fields in the Earth Sciences use the word "age" with subtle nuances, which is understandable. For instance, in the field of paleontology, an age of a fossil can sometimes be ascribed as an age of a stage. For instance, saying "fossil XY has a Tithonian age" or "is Tithonian" is perfectly acceptable when used in this context. However, in the matter of calibrating the numerical age of stage boundaries such usage of the word age is too loose because a stage boundary can last for millions of years; therefore, it lacks accuracy and precision. In the context of calibrating the age of a stage boundary, the word "age" needs necessarily to be taken as a numerical age (or radiometric age), usually arising from a physical measurement which carries a mean value and an error. Since this manuscript deals with the age calibration of stage boundary from a geochronological perspective, no other meaning of the word "age" is possible other than a numerical age. Therefore, every time the word age appears in the manuscript, it should necessarily be interpreted and understood as a numerical age.

Singling out what type of age we are talking about as radiometric age is redundant and unnecessary since no other meaning is possible. The main aim of the manuscript is to dispute the (numerical) age of the JKB, therefore "Berriasian age" is meaningless and confusing with the aim of calibrating the age of a boundary. Certainly, the sentence suggested by the reviewer "The base of the Berriasian stage has been fixed for some years" is 100% correct, which one would correctly interpret as the base of the Berriasian has been fixed at the base of the Calpionella alpine Subzone and has been for many years. However, the sentence does not bear any relation to the numerical age of the base of the Berriasian, aka the JKB, which is the foremost purpose of this manuscript. As described in the introduction, there have been many (numerical) ages for the base of the Berriasian over the years, 135 Ma, 140 Ma, 144 Ma, 145

Ma. This represents a span of 10 Ma, which begs the question: What is the age of the JKB after all? Having an age of a boundary that is floating around a span of 10 Ma is less the ideal. Therefore, by any standards, the age of the JKB has been contentious over the past years. Sure, one could suggest it to be Berriasian, but this is too loose of a definition for the sake of numerical calibrating the geological timescale. Since 2005, the ICS has the JKB at ~145 Ma, which means approximately 145 Ma. From a geochronological perspective this far from ideal for ascribing a numerical age to a boundary. Admittedly, for many outside the field of geochronology such nuance bears no meaning, but for an accurate division of the geological timescale is it is imperative to find a more realistic age for the JKB, where geochronological data from many sections seem to converge to a similar age. We are confident we have demonstrated this in the manuscript.

In short, geochronology is of the utmost importance to understand the rate of geological phenomena; for instance, duration magma magmatic processes, tectonic processes, duration of mass extinctions and recoveries. All this relies on the accurate and precise knowledge of the (numerical) age of rocks, paleontological markers, and stage boundaries. The latter two can only be resolved by using dating horizons that are close to boundaries using geochronological methods that are accurate and precise, which is methodology we have used in this manuscript.

MODIFICATION: No modification was made since we feel we have answered this comment.

*1.13)    "JKB" is not standard terminology. It appears hundreds of times in the text. "J/K boundary" is the norm. Alternatives for use are: the base of the Alpina Subzone, base of Berriasian Stage, Tithonian/Berriasian boundary, or, less precisely, the J/K interval, the boundary interval. . Care is required is using the phrase J/K boundary.*

We feel that abbreviations can take any form, as long as it is clearly stated in the text and consistently used throughout. The use of the "J/K boundary" is just a personal preference of the reviewer as is our choice to use "JKB" just because something might be considered the norm hardly qualifies it to be mandatory. We see no problem with this abbreviation. However, if the reviewer or the

Handling Editor feel adamant about this, we can certainly accommodate it since it is a frivolous matter and simple to adjust.

We have deliberately chosen not to vary the term JKB with its many analogs to avoid confusion, especially to that reader that is not familiar with the various synonyms that the term JKB takes. Since we aim to draw attention from a broader audience, we feel that the term JKB should stay fixed for clarity, even though it might come across as repetitive.

MODIFICATION: No modification was made since we feel we have answered this comment.

*1.14)   Anything that is not exactly correlated with the base of the Alpina Subzone can be said to be in the J/K interval, but not at the boundary. The reader is sometimes not sure what*

       *interval is referred to, or what horizon. Many times a fossil or date is somewhere in the J/K interval, but, to be accurate, nowhere near the actual boundary.*

We do understand that we are introducing a new concept (the JKB interval) to a field that already has a plethora of analog terms. However, we want to make it clear that the JKB interval is NOT a substitute for the JKB and the JKB is not the JKB interval. The idea for the JKB interval mainly stems from the fact that the age of the JKB in both sections do not overlap within our analytical uncertainty, and are offset by ~670 ka (± 335 ka). Furthermore, as pointed out by the reviewer #2 in comment 2.1, the markers are offset in an age which, in our opinion, only builds a stronger case to leave the age of the JKB confined to an interval, the JKB interval. Nevertheless, we will try to make a great effort to make this distinction very clear in the revised version of the manuscript. To supplement this reply, we refer the reviewer to comment 2.16 (i.e., in reply to reviewer #2, J. Pálfy).

MODIFICATION: We hope to have addressed this comment in section 4.5, 4[th] paragraph.

**Reply to W. Wimbledon - Supplementary Comments**

**Page 1, line 28**

MODIFICATION: This is geochronological jargon: "final" usually means the reported age. We will replace the word "final" with "reported age".

**Page 2, line 1 to 5**

The reviewer pointed to a problem with the construction of the sentence. What we want to imply is that reported ages from a previous publication are imprecise and they do not overlap, which means that they do not match, agree, or have the same age. That is what is implied by no overlap.

The main difficulty in finding a (numerical) age for the JKB has been the choice for the base of the Berriasian. Of course, this has been solved, but back in the day when the first attempts to date the boundary (1985, 1995) this was still an issue, and this had significant implications towards the numerical age of the JKB. Maybe this has been clarified for the reviewer.

MODIFICATION: No modification was made since we feel we have answered this comment.

"*... and this level has been the most popular boundary marker for around 30 years*".

The Killian group in their 2014 report (Reboulet et al., 2014) still recommend the base of the Barriasella as a marker for the JKB. This might not be the case for the Berriasian Working Group, where 76% have chosen the base of the Calpionella alpine Subzone, but this is apparently not an overwhelming consensus within the entire community.

MODIFICATION: No modification was made since we feel we have answered this comment.

**Page 2, line 13**

What we are implying here is that the base of the JKB is assumed to the be the base of the Calpionella alpine subzone, not the ash bed. We will rephrase for clear meaning.

**Page 2, line 16**

MODIFICATION: "Recent years" was deleted.

**Page 2, line 19**

MODIFICATION: We will rephrase it to "We also report new nannofossil results from Mazatepec section."

**Page 2, line 22**

See reply above on the usage of the word "age" in reply to comment 1.12.

MODIFICATION: No modification was made since we feel we have answered this comment.

**Page 2, line 24,25**

MODIFICATION: We have rephrased it to "which in turn also validates our age for the early Berriasian and the JKB."

**Page 2, line 27**

MODIFICATION: We will replace "JKB" for "boundary" to avoid repetition. We want to avoid the use of the other many synonyms for JKB to prevent any confusion.

**Page 3, line 3**

MODIFICATION: "a" replaced by "the"

**Page 3, line 5**

MODIFICATION:.. of the Eastern… will be added

**Page 3, line 6**

MODIFICATION: Replaced outcrops by exposed, since fossils do not crop out.

**Page 3, line 8**

Gray is the American spelling. See comments on spelling in reply to comment 1.11

MODIFICATION: No modification was made since we feel we have answered this comment.

**Page 3, line 15-16**

We disagree with the reviewer. The sentence is well constructed. Zircon does not need to be pluralized since it related to the behavior of the mineral zircon in general.

MODIFICATION: No modification was made since we feel we have answered this comment.

**Page 3, 17**

MODIFICATION: Deleted the word dated as suggested, because of the precision of language.

**Page 3, line 21**

MODIFICATION: Mazatepec will be inserted instead of the vague term "the section in Mexico."

**Page 3, line 30**

MODIFICATION: comma added.

**Page 3, line 31**

MODIFICATION: The definite article "the" has been added before the noun R (as in the statistical package)

**Page 4, line 3**

MODIFICATION: "The section" replaced by "The Las Loicas section" as suggested

**Page 4, line 4-5**

MODIFICATION: "Found in the Las Loicas section" deleted for it was redundant, as pointed out by the reviewer.

**Page 4, line 14**

"(ca. 15m stratigraphic height)" is there to facilitate and aid the reader to locate the position in Figure 4.
MODIFICATION: No modification was made since we feel we have answered this comment.

**Page 4, line 19**

*Bralower's thirty year old results must be seen as totally overtaken by more recent results, and to a*
*lesser extent it is true of Casellato 2010. You quote Wimbledon 2017 which shows a more recent situation*

Please see reply to comment 1.4

**Page 4, line 19**

*T, remanei and C. massutiniana are decidedly not typically Berriasian*

Please see reply to comment 1.3

**Page 4, line 29**

MODIFICATION: Magnetozones will replace Magnetochrons.

**Page 4, line 30**

*No. This is very very vague. In numerous sections the base of the Alpina Subzone is proved in the middle of M19n.2n*

MODIFICATION: We have removed any correlation with any magnetostratigraphic data

**Page 5, line 5**

*Rather unsafe. Authors present no evidence on Arroyo Loncoche. They cannot interpret what is or is not M19n.2n at LL, as they say. How can the authors' results be close to those of Inglesia Llanos when they have no magnetostratigraphy to present at Las Loicas and do not work on AL?*

Please see reply to comment 1.2 and also 2.9 (in reply to J. Pálfy)

MODIFICATION: We have removed any correlation with any magnetostratigraphic data

**Page 5, line 25**

*Again, surely this is obsolete work to cite? More up to date references required. The Italian data has*
*been superceded. By the way, Ogg et al. 2016 is not original resesrch but a compilation*

The authors do not agree with Wimbledon (2017) where he said that N. kamptneri kamptneri and N. steinmannii steinmannii bioevents, previously used as infallible biozonal indicators in M17r, have been found widely in lower M18r and the upper half of M19n (Figs. 1, 2). Based on Wimbledon (2017) figure 2 only in Puerto Escano these bioevents are correlated with the upper part of C. alpina. (M19n).
Besides, according to Svobodova and Kostak 2016 (cited by Wimbledon (2017) only in "one" sample they recognized this bioevent in M19n1r and other is correlated with M18r. The record of N. steinmanii steinmannii and the biozone NK1 are correlated with the Calpionella Zone without specifying the
subzone and it is recorded nearly one meter above the acme of C. alpina.

MODIFICATION: No modification was made since we feel we have answered this comment.

**Page 5, line 30**

*This does not match evidence from lots of sites N. steinmannii steinmannii is not a marker for the
Elliptica Subzone, especially when it occurs as low as the Alpina Subzone.*

  N. steinmannii steinmannii defines the base of the NK1 zone and nowhere in the text have the
authors considered this marker as a bioevent of the Elliptica Subzone. The authors explain that it is
found associated with this calpionellid in the Mazatepec section of Mexico, in the same way of other
sections cited and that the NK1 Zone has been correlated in different section with the Elliptica biozone.
The authors does not state that this bioevent is a marker of this calpionellid biozone.

*You quote Wimbledon 2017?*

MODIFICATION: Citation was deleted

Comments

**Page 6, line 9-11**

No modification was made since we feel the sentence is fine.

**Page 6, line 18**

"loc" was supposed to be located

MODIFICATION: Located was corredted

**Page 6, line 19**

MODIFICATION: Tethys regions was replaced by Tethys Ocean

**Page 6, line 23**

We agree the sentence does not read well. What weant to imply is that the age of ash bed LY5 is an age
in the Tithonian. We will rephrase to make it more clear.

MODIFICATION:  can be regarded as an age in the early Tithonian

**Page 7, line 1**

MODIFICATION: Perhaps the sentence would read better if stated: "Therefore, our new ages for the base of the Berriasian and the early Tithonian yield an excepted duration for Tithonian."

**Page 7, line 1**

*How is it "recommended"???? Ogg is just another publication.  And not an ICS publication.*

MODIFICATION: We have tried to make a clearer distinction between Ogg et al. 2016 and the ICS. Comments from page 7-9 have been disregarded since section 4.5, 4.6 and 5 have been completely rewritten.

**Page 7, line 8 subsection title**

We feel that this is a great subsection title, it instigates the reader to pose the question: Do the ages presented here present the age of the boundary globally? Meaning, if we could measure the age of the JKB in every section, would we find the same age everywhere? Although this is impractical because not every section has table horizons close to the boundary, we argue for the fact that the Las Loicas and the Mazatepec agree favorably our ages can be considered as the age of the JKB globally. As a hypothetical, suppose that the age of the Las Loicas was 140 Ma and that the age of the Mazatepec was

143 Ma, then it would be hard to argue that their age agrees. However, they are off by 600 ka, which is a short interval.

Comments from page 7-9 have been disregarded since section 4.5, 4.6 and 5 have been completely rewritten.

**Page 7, line 29**

The reviewer says the FAD of R. asper is much older. How older is the FAD R. asper? Can he precise how much older?

Comments from page 7-9 have been disregarded since section 4.5, 4.6 and 5 have been completely rewritten.

**Page 8, 1[st] paragraph**

*"And yet for 200 years geologists have divided up the geological column quite successfully, with no magnetic markers and with no geochemistry, nd the bulk of agreed GSSPs do not rely on these. Replace this sentence?"*

High-precision geochronology has enabled the understanding of Earth processes in great detail.

The time scales at which we deal in the manuscript are in the order of 50 ka, in which preservation of the paleontological markers becomes of extreme importance. We never suggested that GSSPs do not rely on secondary markers, but rather a valuable tool. Additionally, we draw W. Wimbledon's attention to the comments of J. Pálfy (reviewer #2), where he suggests that we should use our high-precision ages in both sections to show how problematic it can be to assume time-equivalency of biozones. We also share Pálfy view. Our data clearly shows a slight mismatch at the sub 100 ka level. In this scenario, the diachroneity of FAD and LAD's becomes evident and thus the dating of the stratigraphic record using high-precision U-Pb geochronology becomes a powerful tool in unraveling such nuances. It is undebatable that paleontology has been successful in dividing the geological timescale in the past. However, integrating geochronology, stratigraphy, paleontology, geochemistry, and magnetostratigraphy can push the limits of correlations and calibrations of the geological timescale and is the best way forward. Perhaps, in the suggested sentence, we could state that in the context of calibrating the age of stage boundaries at the sub 100ka level, preservation of paleontological markers is an issue. Maybe this way it would be made clearer.

Comments from page 7-9 have been disregarded since section 4.5, 4.6 and 5 have been completely rewritten.

**Page 8, line 5**

*meaning? one level but rest of sentence is about a set of biological events that took place across the Upper Tith-lower Berriasian interval*

We did not understand the reviewer's comment.

Comments from page 7-9 have been disregarded since section 4.5, 4.6 and 5 have been completely rewritten.

**Page 8, line 7**

*what 'explosions'? bloom of small C alpina? It comed after diversification of nannoconids*

We meant the bloom of small Calpionell alpina

Comments from page 7-9 have been disregarded since section 4.5, 4.6 and 5 have been completely rewritten.

**Page 8, line 19-20**

*Vague, no justification shown*

Through out the paragraph we cite publications to support this last sentence.

Comments from page 7-9 have been disregarded since section 4.5, 4.6 and 5 have been completely rewritten.

**Page 8, line 24**

*Its proper name is the "International Chronostratigraphuc Chart"?*

Will make the modification to "International Chronostratigraphic Chart".

Comments from page 7-9 have been disregarded since section 4.5, 4.6 and 5 have been completely rewritten.

**Page 8, line 24**

*meaning?*

It is beyond the scope of the manuscript to go into detail on the issue of offset between Ar-Ar and U-Pb ages. However, it is an important statement to be made from a geochronological perspective.

Comments from page 7-9 have been disregarded since section 4.5, 4.6 and 5 have been completely rewritten.

**Page 8, line 31**

*it is a hole in the sea bed, there is no section*

We will refer to it as core, as was done previously in the paragraph, instead of section.

Comments from page 7-9 have been disregarded since section 4.5, 4.6 and 5 have been completely rewritten.

**Page 8, line 31**

*vague*

We will incorporate examples of the JKB markers in the sentence, even though at this point in the manuscript we are deep into the discussion and have stated and cited what the markers are and expect the reader to be following along.

Comments from page 7-9 have been disregarded since section 4.5, 4.6 and 5 have been completely rewritten.

**Page 9, line 1**

*As a concluding sentence it is not effective. It says, more or less, our age agrees with othe ages. Not a very weighty ending*

We disagree, this last sentence sums up that our age agrees with more recent ages for the JKB, 15    and can be considered the age of the JKB globally. The sentence, in our opinion, is actually quite important sentence and carries a lot of weight. No other study dealing with the age of the JKB could make such a big claim.

Comments from page 7-9 have been disregarded since section 4.5, 4.6 and 5 have been completely rewritten.

**Page 9, line 3 - comment of the title of section 5 – Conclusions and Summary**

*Cretaceous rock/time is base Berriasian stage and start Berriasian age.   What you discuss is geochronology and radiometic dates*

We are not sure what the reviewer meant by this comment. Not very clear.

Comments from page 7-9 have been disregarded since section 4.5, 4.6 and 5 have been completely 25    rewritten.

**Page 9, line 8**

*what interval, you just presented numbers*

The reviewer missed the point of the the meaning of the JKB interval. We talk about the JKB interval previously in the manuscript. For clarification with regards to the JKB interval, we refer to comment 2.16.

Comments from page 7-9 have been disregarded since section 4.5, 4.6 and 5 have been completely rewritten.

**Page 9, line 10-11**

*This ammonite biozone is enormously long, what can it bracket or corroborate? Precision?*

We did not imply that we could bracket anything using an ammonite zones. Our bracketed interval is the JKB interval, which is bracketed with U-Pb ages which is staed in the manuscript.

Comments from page 7-9 have been disregarded since section 4.5, 4.6 and 5 have been completely rewritten.

**Comments on Figures**

Page 17, Figure 2

The main aim of figure 2 is to display our U-Pb data. The biozones are displayed merely conjecturally, since the exact age and duration of the biozones are not known. Therefore, adding boundaries to the biozones would be unrealistic and wrong. Spelling will be rectified.

MODIFICATION: Spelling rectified. The rest of the comment was disregarded since it not possible nor scientifically sound.

**Page 19, Figure 4**

MODIFICATION: Spelling of species names will be rectified. Boundary abbreviations have been commented on previously in this reply. Additionally, abbreviations were adopted to make the figures more clear, less clustered, and easier to read.

**Modifications to the manuscript with respect to the comments by reviewer J. Pálfy (reviewer #2) on the manuscript "Cross-continental age calibration of the Jurassic/Cretaceous boundary"**

**General comments**

*2.1)*     *"… I take several issues with the interpretation, and may suggest guidance for a revised version which could better avoid the pitfalls of confusing regional and global biostratigraphic correlation issues. Instead, a refocused discussion should emphasize the obvious significance of*
*the radioisotopic dates in highlighting problems and contradictions in biostratigraphy."*

REPLY: This is a significant point, and it highlights the importance of dating the stratigraphic record using high-precision geochronology to unravel its subtle nuances. If we have interpreted the reviewer's advice correctly, our ages clearly show that assuming time-equivalency of biostratigraphic zones can lead to erroneous correlations regarding the numerical ages of FAD and LOD. Possibly, this
difference can arise from the migratory rates of these species resulting in the diachroneity of FDA and LOD. This is an interesting point to explore and discuss in the revised manuscript and will be incorporated into the revised version. Nevertheless, we feel that the essential aspect of our data is how younger the age of the JKB is with regards to the long-lasting age of 145 Ma. This is the most crucial contribution of the manuscript, and the discussion around how the age of the JKB in both sections
favorably agree is still central to the manuscript.

MODIFICATION: We have incorporated the suggestion by reviewer J. Pálfy in his comments 2.1 and 2.14 (see author's reply). As the reviewer suggested, we have refocused part of our discussion to the embrace the mismatch between the ages of the JKB of the dated sections and discuss the pitfalls of regional and global biostratigraphical correlation. To accommodate the reviewers suggestion in comment 2.1 and 2.14 we have completely rewritten section 4.5.

**Specific comments**

2.2)    *The paper needs a proper "Geological and stratigraphic setting" chapter to augment and*
*replace the "Studied areas" in the current version. Formation names, i.e. the bare bone*
        *lithostratigraphy should be complemented with brief characterization of basin evolution and*
        *depositional environments, to provide context for assessment of stratigraphic completeness and*
        *sedimentation rates in the section, the latter being crucial in the authors' arguments in*
        *comparing the JKB age of different sections.*

REPLY: In the "Studies areas" chapter, we chose simply to give a brief description of where the studies sections are located and cite important publications relevant to where the sections are exposed. There are numerous publications on the tectonic architecture and basinal evolution where the sections are that are cited in the manuscript. As it stands, the manuscript is 4626 words long, which we feel is an adequate length for a publication. If we were to expand the "Studies areas" chapter with a detailed
"Geological and stratigraphic setting" chapter, it would increase the manuscript to another 800-1000 words. Even then, it would not do justice to fully review the geological setting of both geological settings within 1000 words (e.g., 500 words each basin). The reviewer claims that such an expansion of the regional geology would be useful to understand better the sedimentation rate in Mazatepec, which is an integral part of our discussion. However, we make it pretty clear in the manuscript that the
sedimentation rate in the Mazatepec section is **unknown**, and we further use both a low and high sedimentation rate to back-calculate the age of the JKB in the section. Even with a thorough knowledge of the sedimentological and stratigraphical background, there is no hard evidence for the rate of sedimentation rate in the Pimienta and Tamaulipas formations. Ultimately, this would inevitably leave us with a subjective choice of sedimentation rate based on the depositional environment and sedimentological structures present. Moreover, we also make the case that the choice of sedimentation rate is not that important. Nevertheless, we would not oppose slightly expanding the "Studies areas" chapter, or giving it a new title if the reviewer feels adamant about the subject. We leave this option to the discretion of the Handling Editor, because it influences the format with which publications in Solid
Earth are communicated.

MODIFICATION: No modification was made, and we await the Editors decision.

2.3)   *Care should be taken to ensure consistency in terminology and usage of biozones. Much biostratigraphic information is presented both in the text and in Fig. 4. However, it is not clear to the reader what, if any of these is new here, what is taken unchanged from the references*
*cited, and what is revised from published sources*

REPLY: In the caption for figure 4, there is ample information on the information that is new and what is cited from other publications. We will try to make the figure 4 clearer at the request of the reviewer well as its caption.

MODIFICATION: Names of species were corrected as pointed out. Some species names were uncapitalized and steinmannii spelling was corrected.  A dashed green line was put in to connect and high-light the mismatch between Las Loicas and Mazapetec. References were put at the top of each panel to make clear what is being cited and what is not, although also in the caption figure there is ample information on what was cited and what is new.

2.4)   *Cases where there is controversy in either the zonal subdivision of sections or their correlation, based on ammonoids, calpionellids and nannofossils (e.g., between Riccardi 2015 and Vennari et al. 2014) and the stance of the authors should be more clearly stated.*

REPLY:

Ammonoids: There is no discrepancy among the biozonation of Riccardi, (2015), the Vennari et al., (2014), and the present manuscript regarding the sequence and names of index species of each biozone. It is worth to mention here that Riccardi explicitly states: "*There is no attempt to deal here with the*

*precise definition of the Jurassic-Cretaceous limit, and therefore the use of terms such as "Tithonian," "Berriasian," "Upper/Late Jurassic" and "Lower/*

*Early Cretaceous have been kept to a minimum and is usually adopted when quoting other sources. It is considered that once biostratigraphic correlations are well-established definition of Stage and System boundaries will follow by convention*" (Riccardi 2015, p. 24).

Calpionellids: The data from this manuscript has been published by López-Martínez et al., (2013) for the Mexican section and López-Martínez et al., (2017) for the Argentine section.

Nannofossils: The data from this manuscript has been published by Vennari et al. (2014) for the

Argentine section. The data presented here for the Mexican section is new, and a systematic paper is in preparation (Lescano et al. in prep.).

MODIFICATION: No modification was made since we feel we have answered the comment.

*2.5)*     *The reader might suspect that calcareous nannofossil occurrences are newly obtained as Supplementary Fig. 3 is promised to present them (p. 3, l. 26), but this figure is missing.*

REPLY: Yes. Unfortunately, we have not placed the Supplementary Figure 3 (distribution chart for the calcareous nannofossil species) in the Supplementary Materials as stated in p.3, l. 26. We apologize and promise to rectify.

MODIFICATION: We have placed the calcareous nannofossil chat in the Supplementary Material as Fig. S1.

2.6)    *Details of reporting of the error and age interpretation would be better placed in the main text's Methods chapter rather than in the Supplementary Material.*

REPLY: The detailed account of the geochronological data is intended for full disclosure of its meaning and interpretation; however, this would only be appealing to a specific subset of the geochronology community. The average reader, drawn by the interest of knowing the age of the JKB, in our opinion, would be distracted by an excessively detailed description of the geochronological U-Pb data in the main text. Moreover, this information is not further referred nor directly used in the discussion and conclusion chapters, i.e., the meaning of a depositional age for the ash beds, number of grains selected for weighted means, etc. These are not information that is central to the discussion of the data and conclusions. This is why we decided to keep it in the Supplementary Materials. Nevertheless, we leave it at the discretion of the Handling Editor do choose what best fits the format of the journal because it would be an easy adjustment to make to the revised manuscript.

MODIFICATION: No modification was made since we feel we have answered the comment.

2.7)    *For the aimed global relevance in time scale studies, the most conservative error (i.e., that including the tracer calibration and decay constant errors) needs to be quoted and used for each U-Pb dates throughout the paper. This is typically still within 0.2 Ma, a commendable high-precision.*

REPLY: The reason high-precision ages are reported with three errors (as explained in the Supplementary Materials) is to allow for an appropriate propagation of errors when comparing different geochronological datasets that been acquired through different geochronological methods (e.g., $^{39}$Ar/$^{40}$Ar, U-Pb (SHRIMP, LA-ICP-MS)). In this manuscript, we do not directly compare datasets from other studies. We do, indeed, aim to challenge the JKB recommend age in the ICS is ~145 Ma, which mainly highlights the lack of precision and accuracy towards the JKB age. In any case, the JKB age in the ICS is based on the $^{39}$Ar/$^{40}$Ar age of Mahoney et al. (2005). Nevertheless, our ages are so much younger than that of Mahoney et al. (2005), making precision, not such a big deal for the sake of challenging the ICS age. Hopefully, other sections that span the JKB will be dated in the future and most likely use U-Pb CA-ID-TIMS since it has become a gold-standard in dating the stratigraphic record. Therefore, how we quote precisely in the manuscript is not that big of a deal.

MODIFICATION: No modification was made since we feel we have answered the comment.

2.8)    *The chapter "Results and discussion" needs to be split into two, allowing results to be clearly separated from the interpretation.*

REPLY: In the same vein as the reply to comment 2.6, we wanted to make a concise manuscript.

In this sense, we feel that the nitty-gritty dissection of the geochronological data should not be moved to a separate "Results" chapter in the main text. Instead, we describe the data along with the discussion, which in our opinion reads better and is not unusual in scientific communications. As far as the Solid Earth's author guideline goes, it does not mandate that results be separated from the discussion. Moreover, we think that the lack of a specific "Results" chapter does not compromise any of the discussion or conclusions in the manuscript. Therefore, we thought it might be better to leave the results and discussion together. We believe that this comment is more of a personal preference of the reviewer than a weakness of the manuscript. Nevertheless, we leave it at the discretion of the Handling Editor do choose what best fits the format of the journal because it would be an easy adjustment to make to the revised manuscript.

MODIFICATION: No modification was made, and we await the Editors decision.

2.9)    *Even though it is widely accepted that magnetostratigraphy is very useful for global correlation in the JKB interval, projecting the magnetozones identified in the Arroyo Loncoche section in the Neuquén Basin (Iglesia Llanos et al. 2017) introduces additional confusion (p. 5, l. 1-9, Fig.*

*4) to the already complex web of stratigraphic correlation of the three studied sections. The new results from Las Loicas do not appear to be closely correlatable with Arroyo Loncoche, Fig. 4 reveals that the placement of the JKB is offset by nearly one ammonoid zone, being near the base or at the top of the Substeueroceras koeneni zone, respectively. It would suffice to say that*

*magnetostratigraphy of the Las Loicas section will be desirable to enhance the utility of the newly obtained U-Pb ages and clarify contentious biostratigraphic correlation issues.*

REPLY: In the manuscript, we do not project the magnetozones of the Arroyo Loncoche section to the Las Loicas or any other section. We merely attempt to correlate the JKB in the Arroyo Loncoche to the Las Loicas section using the Alpina Subzone and the M19.2n, which are the most compelling evidence for the JKB in either section. We admit that there is a mismatch between the ammonite zonations, which is clearly stated in the manuscript (p. 5, l. 8-9). Additionally, we also cited a discussion on the matter in López-Martínez et al., (2018). Nevertheless, the thickness of biozones changes as a function of facies, randomness of finding markers in the field, the latter hugely influenced by preservation, and paleogeographical position within a sedimentary basin. Therefore, although we do see that better understanding the mismatch between both sections as an incentive for future research, we do not, however, see this as a significant issue to be explained.

One needs to keep in mind that another principal aim of this manuscript is to try to show that the age of the JKB in ICS is too old. In an idealized case, one would find the age of the M19.2n and the base of the Calpionella alpina Subzone to be the same age (assuming these markers are exposed in different sections as is the case in this manuscript). This would require that both of these markers have a datable horizon very close by. However, in the real world, this scenario is quite hard to come by, and we need to try and reconcile the available data despite its shortcomings. In the context of trying to show that the ICS age of the JBK (145 Ma) is too old, the data from Las Loicas and Arroyo Loconche seems to be in reasonable agreement, in our opinion. That is, if we consider that the most trustworthy markers for the JKB are the M19.2n and the base of the Calpionella alpine Subzone, even with the mismatch of the ammonite zones between Las Loicas and Arroyo Loconche (which would be a couple 100 ka), the age markers for the JKB of these two sections would not be off by 5 Ma. **Furthermore, it is important to point out that in the absence of a reliable biostratigraphic framework, such as the case of Arroyo Loconche, magnetostratigraphy is just a floating scale (very important to bear in mind).**

Therefore, from this perspective, even with the ambiguity in the correlation between these two sections, the age of the JKB at ~145 Ma is hard to reconcile. We do understand that the M19.2n in Arroyo Loconche might seem older than the base of the Calpionela alpine Subzone in Las Loicas when compared against the Substeueroceras koeneni biozone as a relative timescale. Nevertheless, this discrepancy would not allow, for instance, the interpretation that the age of the M19.2n in Arroyo

Loconche to be as old as 145 Ma and the age of the Calpionella alpina Subzone in Las Loicas to be at ~140 Ma, which would be the alternative to invalidating our conclusion. Furthermore, our age in the *Virgatosphinctes andesensis* biozone (Early Tithonian) would certainly not allow this interpretation. In closing, the explanation above only exposes the how poorly constrained the current age of the JKB is, that even with a crude correlation (which is what is available at our disposable at this conjecture) the age of the JKB at 145 Ma seems implausible.

Additionally, in the manuscript, we cite many references that have also dated the JKB and found ages similar to ours. Furthermore, our goes for the base of the Berriasian are much easier to reconcile with the ages for the Early Cretaceous ages (see page 8, lines 10-10 in the manuscript). In closing, there is substantial evidence from different fields that point to an age of the JKB that is much younger than in the ICS (We would also like to refer the reviewer to the reply on comment 1.2, i.e., in reply to reviewer #1).

Having said this, we realize that both reviewers took issue with our attempt to correlate the M19.2n in Arroyo Loconche and the base of the Alpina Subzone in Las Loicas in an attempt to build a more solid case for our age of the JKB. If our arguments remain unconvincing, we will not oppose removing entirely this from the discussion and figure 4. Hopefully, our explanation was satisfactory.

We would, in this case, value comments and advice from the handling Editor on the matter for the revised manuscript.

MODIFICATION: Even though it we argued in favor of correlating our geochronological and  data with the magnetostratigraphic data of Iglesia Llanos, et al., (2017), we have we have decided to remove this from the manuscript and figures since both reviewers took issues with this approach.

2.10)    *Discussion on the age of the JKB in the Mazatepec section includes an assumption on the FAD of a nannofossil taxon, Nannoconus steinmannii minor, not actually found in the section (p. 5, l. 31 – p. 6, l. 3). Such speculation is best avoided.*

REPLY: Our consideration of the *N. Steinmannii* is speculative, very short. We certainly do not substantiate any conclusion on this comment. Nevertheless, the N. steinmannii defines the base of the biozone and is the main bioevent, and the others are defined as close and secondary with regards to this bioevent. Therefore, this was just to give the reader food for thought, as so to speak.

MODIFICATION: No modification was made since we feel we have answered the comment.

2.11)    *Beware of the lack of formal definition of base Tithonian. There is no agreed-upon GSSP decision yet, contrary to what is implied here (p. 6, l. 25). The attendant uncertainties of stage boundary placement and its correlation with the Andean sections make the time scale calibration use of La Yasera U-Pb date more problematic than admitted here.*

REPLY: Indeed, there is no agreement on the GSSP for the Kimmeridgian-Tithonian boundary. We will remove the sentence in brackets that suggested otherwise (p.6, l 25), and will make it clear that the KmTB is not formally defined.

MODIFICATION: The affirmation of that in p.6. l 25 that there was a formal definition of the Kimmeridgian and Tithonian boundary was removed as suggested.

*2.12)   The discussion on the duration of the Tithonian is interesting but contains a factual error and misses some further opportunities. The Geological Time Scale 2016 (Ogg et al. 2016) is misquoted, it assigns 150.8 Ma to the base of Tithonian Stage and 145.5 Ma to the JKB.*

REPLY: In the ICS chart 2018, the age of the KmTB is $152.1 \pm 0.9$ Ma. Please see
http://www.stratigraphy.org/index.php/ics-chart-timescale. Additionally, in the compilation of Ogg et al., (2016), Chapter 12 – Jurassic, Figure 12.1 page 152, and Figure 12.4 page 157, the age quoted is 152.1 Ma for the base of the Tithonian.

MODIFICATION: No modification was made since we feel we have answered the comment.

*2.13)   It would be useful to compare two other, independent duration estimates. The Pacific M*
*sequence of magnetic anomalies has long featured in time scale calibration. The recent work of Malinverno et al. (2012) (the MHTC12 scale) suggests 6 m.y. for the Tithonian, i.e., between magnetochrons M22An and M19n2n.*

REPLY:  We thank the reviewer for pointing this out to us, and we will undoubtedly discuss and compare Malinverno et al., (2012) timescale for the Tithonian in the discussion, especially since it is
very close to our estimate for the duration of the Tithonian.

MODIFICATION: We have addressed the comment 2.13 which suggested that we compare out estimates for the duration for the Tithonian with that of the independent duration estimates of the Pacific M sequence of magnetic anomalies of Malinverno et al. (2012). This was incorporated into section 4.3.

*2.14)   The cyclostratigraphic analysis of Kietzmann et al. (2015; not cited by Lena et al.) identifies 10 long eccentricity cycles for almost the entire Tithonian, starting with the Virgatosphinctes mendozanus zone dated here at La Yesera, hence a duration of c. 4 m.y. The discussion should emphasize that the duration favored here is longer these previous estimates using other methods*

*and offer possible reasons to explain the difference, perhaps considering biostratigraphic correlation issues.*

REPLY: There are two issues here: First, in Kietzmann et al., (2011) the Tithonian was more than 210 m thick in Arroyo Loncoche; then in Kietzmann et al., (2015) the Tithonian is reported as 195 m thick; and finally in Iglesia Llanos et al., (2017) the Tithonian was reported with less than 160 m. This makes more inadequate the ten long eccentricity cycles for almost the entire Tithonian. The second issue is that following Vennari et al. (2014) and Riccardi et al. (2015), and in the present manuscript, the andesensis (former mendozanus) zone is correlated with the Tethyan ammonite zones, which are above the base of the Tithonian, i.e. the hybonotum zone is not represented in Vaca Muerta Formation.

MODIFICATION: No modification was made since we feel we have answered the comment.

*2.15) Perhaps my most important criticism and suggestion pertains to the projection of a sedimentation rate-based JKB from the Mexican Mazatepec section into Las Loicas in Argentina. The authors can make a much stronger case and build a more logical argument by projecting the actual U-Pb date, expressing the stratigraphic height from the age-model calculation as ~28.5 m and note the mismatch in biostratigraphies. Reading from Fig. 4, beds of the same numeric age thus appear assigned to nannofossil zone NJK-B vs. high in NJK-D, to calpionellid Crassicollaria zone vs. Calpionella zone (and its third subzone, the Elliptica subzone, and ultimately to lower Berriasian vs. upper Tithonian at Las Loicas and at Mazatepec, respectively. The discussion could thus be refocused to use the newly obtained high-precision and high-resolution U-Pb age framework to highlight biostratigraphic correlation issues, most likely due to diachronous FAD-LADs of certain key taxa.*

REPLY: We have partially addressed this inquiry in question 2.1. Nevertheless, we are happy with this comment because it further substantiates our arguments, especially for a JKB interval. We agree with J. Pàlfy that the mismatch in the age of the FAD-LAD in Las Loicas and Mazapetec is clear evidence that assuming age-equivalency of markers and stage boundaries is problematic when working at the sub-100 ka level and highlights the importance of high-precision geochronology to the stratigraphic record. Furthermore, in the context of the JKB, it stresses the importance of leaving **the**

**age** of the JKB confined to an interval (we further explore this in reply to question 2.16). We welcome this comment and will surely incorporate this into the revised manuscript because we see this as an essential implication from our data.

MODIFICATION: Comment 2.15 was also addressed and incorporated into section 4.5. Also see modification to comment 2.1.

*2.16)   To strengthen the argument for potential problems in biostratigraphic correlation, the authors might comment on the discrepancy of ammonoid-based correlation, and striking differences of thickness of zones in different sections even within the Vaca Muerta Fm. (e.g. Argenticeras noduliferum zone: ~27 m in Las Loicas vs. 5 m in La Yesera section).*

There are important facies and thickness changes between Las Loicas and La Yesera sections due to their different paleogeographic positions within the Neuquén Basin. La Yesera section is further east (see paleogeographic sections for example in Kietzmann et al. (2015).

MODIFICATION: Comment 2.15 was also addressed and incorporated into section 4.5. Also see modification to comment 2.1.

*2.17)   It the "Global correlation" chapter, the suggestion of understanding the JKB as an interval (p.*

*8, l. 1-10) is conceptually flawed and needs to be rephrased. By definition, the JKB boundary (as any other chronostratigraphic boundary) is a time line. It does indeed carry an uncertainty of our numeric calibration but it cannot be equated with an actual time interval in which different "boundary events" took place.*

The reviewer may not have understood what we meant by the term Jurassic/Cretaceous interval.

We want to make it clear the JKB is not tantamount to JKB interval; in other words, they are not the same thing. We did not suggest that the JKB be understood as an interval (at least that was not our intention), but rather the age of the JKB be left within a bracketed interval, thus the idea of the JKB interval. This mainly stems from the fact that the age of the JKB in both sections do not overlap within our analytical uncertainty, and are offset by ~670 ka (± 335 ka). Furthermore, as pointed out by the reviewer in comment 2.1, the markers are offset in an age which, in our opinion, only builds a stronger case to leave the age of the JKB confined to an interval, the JKB interval. In other words, what we propose here is that the interval constrained by our geochronology is short enough that the JKB can be placed somewhere in that interval because a single age is yet out of our reach. We feel confident that this interval can get tighter as newer sections are dated in the future. Even though they do not overlap, the ages presented here highlight a discrepancy between the age of the JKB in the ICS and the ages that we have measured.

MODIFICATION: Comments 2.17 and 1.14 regarding the concept of the JK interval was rewritten in section 4.5. Hopefully, the concept of the JKB interval is clearer.

*2.18)*    *Also in this final chapter, consider the significance of your argument for a significantly younger*

*JKB together with Martinez et al. (2015) suggested age for the base Valanginian at 137 Ma.*
          *This would make for a shorter than previously understood Berriasian Stage of a -3 m.y.*
          *duration. This in turn contradicts with the astrochronology of Kietzmann et al. (2015), who*
          *identify more than 10 long eccentricity cycles in the Berriasian part of the Vaca Muerta Fm.*

The issue with Valanginian boundary is presently in the discussion as well as the Hauterivian and Barremian ages by new high precision U-Pb CA-ID-TIMS dating together with cyclostratigraphy and the ammonoid and nannofossil biostratigraphy in the Neuquén Basin by Beatriz Aguirre Urreta and Mathieu Martinez (in prep.). Some results already published also show several million-year discrepancies with the ICS Time Table.

MODIFICATION: No modification was made since we feel we have answered the comment.

*2.19)    The statement in chapter "6. Data availability" suggests that some of the raw data will be withheld until completion of the thesis of the first author. Instead, all data should be made available at the publication of this paper. Understandable practice is not to release data in a thesis prior to publication, but there should be no reason to justify an embargo the other way*
*around.*

We will remove this section since all the data is reported in the data table in the supplementary materials. The reported U-Pb table data can easily be copied and pasted on the excel sheet, where it can easily be manipulated in Isoplot in Excel and or R Studio, for instance. Or instead, we can state the latter in chapter 6.

MODIFICATION: We have removed this section since we initially misinterpreted the data availability requirement. As stated in the reply, we do not withhold any data, and every data we use is presented in the manuscript and in the supplementary materials.

*2.20)    Table S1 contains the essential data for the U-Pb geochronology, it should be placed in the main part of the paper.*

We disagree with the reviewer to place the U-Pb data Table, T.S1 to the main text. With the aim of keeping the manuscript more appealing, we feel that by putting raw data tables cuts the flow of the written text and distracts the reader. Therefore, we think that the data table T.S1 is better viewed separately from the main text, especially when reading in a digital format (which we encourage). It allows going back and forth from the text to the data table more readily if the reader deems necessary.
Nevertheless, we leave it at the discretion of the Handling Editor do choose what best fits the format of the journal, and also because it would be an easy adjustment to make in the revised manuscript.

MODIFICATION: No modification was made and we await the decision of the Handling Editor

2.21)  *Fig. S is also worth transferring from the Supplementary Material to the main part. (However, its labeling needs re-coloring so it be legible in black and white print, panel C might be more informative to show the dated ash bed, D needs labels, and the figure needs a caption.)*

In trying to keep the manuscript short, concise and to the point, we have opted to leave field figures (Fig. S) in the Supplementary Materials. We feel that figures that do not directly support any of the discussion or conclusion and are best kept in the supplementary material. Nevertheless, we leave it up to the Handling Editor to advise us on what better suits the format of the journal. We thank the reviewer for pointing out that the figure was, unfortunately, left out the caption and we will incorporate his advice on how to better the figures such as recoloring for printing and better labeling.

MODIFICATION: Fig. S is now Fig. S2 (because the nannofossil chart is now Fig. S1). No modification was made as where the Fig. S2 where should be placed Supplementary Materials or the main manuscript. Ee await the decision of the Handling Editor.

Recoloring of Fig. S2 was not made since printing in black-white inevitably leads to loss in quality. Furthermore, the pictures will be in color in the online version and can be better viewed on a digital platform (computer, laptop, tablet, phone etc), which also avoid printing.

**Closing remarks from the authors**

In closing, we would like once more to show our appreciation to J.Páfly for reviewing our manuscript and accepting it for publication after the revision. Many of the reviewer's suggestions we agree and will fully accept, with only a very few where we disagree or would not favor the change. For instance, there where two comments that, in our opinion, that standout and substantially add to the manuscript. First, the refocusing the discussion around the apparent mismatch between the ages of the biozones in Las Loicas and Mazatepec, which we address in question 2.1 and 2.14. This is the most critical comment from the reviewer, and we welcome it and assure we will incorporate this will be added to the discussion in the revised version. Second, the renaming of the "Studies areas" section for a "Geological and Stratigraphical Setting" and an expansion of both sections. On this comment, we argue as to why we felt it was essential to leave the Studies areas section short, but did not oppose to reviewer's suggestion. In any case, leave it to the decision of the Handling Editor for the revised version.

All other comments from J. Pálfy, albeit pertinent, we feel that they are minor and straightforward to adjust. For instance, the reviewer suggests a "Results" section separate from the Discussion section, which would imply moving the description of the results found in the Supplementary Material to a new chapter entitled Results in the main text. Another similar request is to place the raw data tables in the Supplementary Material in the main text. Even though we oppose such changes in the structure, we do not see it as a significant modification to the manuscript, and we leave it to the Handling Editor to decide what would best fit the journal's format. Other requests pertain to improving the readability and clarity of the figures, adding a caption to one of the supplementary figures. Modification in the grammar usage, word choice, style, and spelling will promptly modify since they will improve the manuscript. In short, we feel that we have dealt with all of the reviewer's comments adequately and hopefully, the answers fulfill the requirements for publications by both the reviewer and the Handling Editor.

**Technical corrections**

*The comments below also include several suggestions for better English language,*

*style and word choice.*

*p. 1, l. 12 (and elsewhere): age ! numeric age*

We discuss this in reply to comment 1.10 to reviewer #1 W. Wimbledon. Since this is a paper that discusses the age of a boundary from a geochronological perspective age is necessarily a numerical age. In our view, it would be some tedious to specify age every time. In the introduction, however, we use the "absolute age" nomenclature to distinguish it from the more older ages derived from statistical interpolation. Therefore, in that context, we felt it was necessary to make the distinction. However, throughout the text when we mention "ages", it can only be numerical ages or numeric ages because what we present are U-Pb ages, which are numerical by definition. Therefore, we feel a distinction is not necessary.

MODIFICATION: We have deleted the adjective "absolute" to qualify the noun "age" as requested reviewer #2  and replaced it with "numerical".  Especially in the abstract, introduction wherever the distinction felt necessary as was pointed out by reviewer #2 in his technical comments. However, whenever the word age referred to our results or any of the data presented we did not qualify it as numerical since this would be redundant, because our results are necessarily numerical ages.

*elusive ! difficult to determine*

OK. Agreed

MODIFICATION made and can be found high-lighted version of the revised manuscript.

*l. 16: display ! contain*

OK. Agreed.

MODIFICATION made and can be found high-lighted version of the revised manuscript.

*l. 21: one of the last major Phanerozoic stage boundaries ! last Phanerozoic system*
*boundary*

OK. Agreed.

MODIFICATION made and can be found high-lighted version of the revised manuscript.

*l. 23: absolute ! [delete, avoid "absolute age" altogether]*

OK. Agreed.

MODIFICATION made and can be found high-lighted version of the revised manuscript.

*p. 2, l. 3: Calpionella alpina subzone (cf. l. 16) [ensure consistency in zonal names*
*and terminology]*

OK. Agreed.

MODIFICATION made and can be found high-lighted version of the revised manuscript.

*l. 17: selected ! suggested*

OK. Agreed.

MODIFICATION: fell it is not suggested because the base of the Aplina Subzone as the base of the Berriasian was voted on, and consequently selected not suggested.

*l. 21: Kamptneri ! kamptneri*

OK. Agreed.

MODIFICATION made and can be found high-lighted version of the revised manuscript.

*p. 3, l. 6: spans ! exposes*

OK. Agreed.

MODIFICATION made and can be found high-lighted version of the revised manuscript.

*l. 12: out of sequence numbering of figures (not as they appear in text)*

OK. Agreed. We will modify.

MODIFCATION: Figures are no in order of appearance, Fig. 1 page 2 line 10, Fig. 2 page 2 line 15, Fig. 3 and 4 page 2 line 23.

*l. 26: optical images ! photomicrographs*

OK. Agreed.

MODIFICATION made and can be found high-lighted version of the revised manuscript.

*p. 4, l. 3: The section ! The Las Loicas section*

OK. Agreed.

MODIFICATION made and can be found high-lighted version of the revised manuscript.

*l. 29: impose ! may provide*

OK. Agreed.

MODIFICATION made and can be found high-lighted version of the revised manuscript.

*p. 5, l. 2, 6: fossil density ! abundance of fossils*

OK. Agreed.

MODIFICATION made and can be found high-lighted version of the revised manuscript.

*p. 6, l 24: Tithonian*

OK. Agreed.

Comment no longer relevant since this part of the manuscript has been removed from the manuscript been rewritten.

*p. 9, l. 22: thank*

OK. Agreed.

Comment no longer relevant since this part of the manuscript has been removed from the manuscript been rewritten.

*p. 10, l. 3: Neuquén*

OK. Agreed.

MODIFICATION made and can be found high-lighted version of the revised manuscript.

*p. 11, l. 11: [delete] February*

OK. Agreed.

MODIFICATION made

*p. 12, l. 6: Potosí [+spell out journal name]*

OK. Agreed.

MODIFICATION made and can be found high-lighted version of the revised manuscript.

*l. 13: & [delete]*

OK. Agreed.

MODIFICATION made.

*p. 14, l. 4: Episodes [delete the rest of name]*

OK. Agreed.

MODIFICATION made .

*l. 14: Aguirre-Urreta*

OK. Agreed.

MODIFICATION made and can be found high-lighted version of the revised manuscript.

*l. 22: [provide doi instead of URL]*

*References cited in text but not listed in reference list: Edwards, 1963 R Core Team, 2013*

OK. Agreed.

MODIFICATION made and can be found high-lighted version of the revised manuscript.

*p. 15, l. 3: Distribution of continents ! Global paleogeography*

OK. Agreed.

MODIFICATION made

*l. 9-15 (Fig. 3): Give stratigraphic horizon of occurrence (e.g. m from base) fo each specimen photographed*

OK. Agreed.

MODIFICATION: Fig. S1 (nannofossil chart) should supplement this request.

*p. 16, Fig. 1: delete title, consider using different base map, do not show migrazion*

*routes and sections not discussed in text.*

OK. Agreed. We will consider just leaving only the two sections studied. However, it is quite common to add sections that are of the same age to a paleogeographical maps to give a to give the sense of the time equivalent between sections even though they are not discussed in the text.

MODIFICATION: Title was modified, "distribution of continents" was replaced by "global paleogeography". We did not use a different base map, since no justification from the reviewer mas made as to why the maps map should be changed. Nor a suggestion as of which base map should be used. Migratory routes were left in the map because we do suggest that the rate migratory routes could be a possible explanation to the difference in age between Las Loicas and Mazatepec. Section that were not in the study were left in to convey the idea that the these section are contemporaneous.

*p. 17, Fig. 2: Barriasian ! Berriasian*

*[J/K boundary interval – see comments about conceptual flaw here]*

*[fonts too small in the upper part, too large in the lower part]*

OK. Agreed.

MODIFICATION made

*p. 18, Fig. 3: [it is redundant to show taxon names here, it is customary to give them*

*in the caption only]*

OK. Agreed.

MOFIFICATION: We have not removed taxon names since we do not see it as redundant, nor is this relevant.

*p. 19, Fig. 4: [this is the key figure of the paper, already need to refer to in the*

*Geological setting, so make it Fig. 2; A: show meters; put Las Loicas section to a*

*separate panel B, making the others C and D; La Yesera: indicate placement of JKB;*

*some lettering uses illegibly small font]*

Here we disagree with the reviewer. There is no sedimentological or geological consideration is figure 4, but rather a comparison between the ages of markers from each section. Furthermore, Figure 4 should be within the discussion chapter as the bulk of the discussion pertains to this figure. Therefore, we do not see the purpose of it being placed at the beging on the manuscript.

MODIFICATION: Arroyo Loconche section from Iglesias Llanos et al., (2017) was put in its own panel A, and the other panels were renamed accordingly, B, C, D. Stratigraphic height was add to the Arroyo Loconche section, as requested.

*Supplementary Material*

*p. 1, l. 1: Ash beds were crushed ! Samples were crushed*

OK. Agreed.

MODIFICATION made and can be found high-lighted version of the revised Supplementary Materials.

*p. 3, part 5*

*Give weight of each sample so zircon yield can be assessed in this context.*

*Grains discarded as too old are erroneously quoted as >_150 Ma for each sample,*

*provide true cut-off age of grains not included in age calculation.*

Weight of the samples was not made because it is not customary to do so. Grains 150 Ma were discarded. The cut-off age for grains included in the weighted mean is sample dependent and are usually the youngest overlapping grains.

MODIFICATION : no modification was made since we feel the comment was answered.

*5.3 (p. 4): Ash bed LL10 has n=6 grains in Fig. 2, four in text*

OK. Agreed, will change it to 4, not 6.

MODIFICATION made and can be found high-lighted version of the revised Supplementary Materials.

*5.4. Ash bed LL13: include date of discarded grains in Table S1 (really older than 450*

*Ma?)*

We do not see the point of reporting the age of grains that are significantly older than the weighted mean of the ash bed. It serves no purpose. Ages much older than the weighted mean are hard to evaluate if they are detrital of inherited from older basement rocks volcanic source.

MODIFICATION : no modification was made since we feel the comment was answered.

*5.5. "Due to its proximity to the Tordillo Fm." [it is from the Tordillo Fm.]*

*inherited grains or detrital grains?*

OK. Agreed.

MODIFICATION made and can be found high-lighted version of the revised Supplementary Materials.

*5.6. MZT-81 (p. 5): check this descriptions, there are errors here. four discarded grains*

*(not five), the grain numbers are in error (belong to sample LL10)*

OK. Agreed. Thanks for pointing this out. Will be rectified.

MODIFICATION made and can be found high-lighted version of the revised Supplementary Materials.

*Fig. S needs a caption and should be transferred to the main part of the paper. The*

*labels of the figures need to be recolored so they are legible in black and white print as*

*well.*

OK. Agreed.

MODIFICATION: Caption has been given. No coloring was made. Transfer to the main text was not made since await the Editor's decision on the issue.

*Table TS.1 is essential to assess the U-Pb dates reported so it should be transferred*

*to the main part of the paper.*

Please see the discussion to comment 2.19.

MODIFICATION: No modification was made since we await the Editor's decision since we disagree with the reviewer.

*Sample LY5 in Table TS.1: why discard grain z67 and keep z10, when the first one is*

*not older and its error is not larger? This and similar issues of only marginally different aged grains*

*undermine the credibility of unbiased and rigorous selection of grains for the age interpretation.*

Weighted mean ages are nothing other than the average mean value of set of dates (youngest grains). In this case, grain LY z67 has a mean value of 147.740 Ma and the precision with what we know the true age of the grain is 93 ka. In figure 2, it is quite clear that LY z67 does not overlap with the weighted mean age of the youngest grains, which means it has little to no chance of statistically belonging to the subset of youngest grains of the population. On the other hand, LY z10 has a mean value of 147.8 Ma and the precision with which we know the age of the 1.1 Ma (much lower precision), and from Fig. 2 it clearly overlaps with the weighted mean age of the sample, which implies that it does have some probability of being a part of the subset of younger grains. In short, LY z10 statistically has a better chance of belonging to the subset of the youngest grains than LY z67, even though the mean value of LY z67 is slightly younger than LY z10. This is just a question of precision, or how well-known is the confidence interval for a particular physical measurement. We draw the attention of the reviewer to compare the Pb* concentration of these two grains. Here, precision is mainly limited by the amount of sample. If the sample size was any bigger, the precision would be higher. Thus the confidence interval reduced. And in that case, grain LY10 would have possibly been excluded from the weighted mean age of the ash bed.

MODIFICATION : no modification was made since we feel the comment was answered.

*8.2 (p. 11), Table TS.2: Why is the age value of 2 m any different from the age of LL13*

*taken from this level?*

This is because the stratigraphic height of LL13 is in fact at height three m and not two m. This will be rectified in the main text. Notice that the age of LL13 is 142.039 ± 0.058 Ma and the age of stratigraphic height 3 m is 142.04 ± 0.06 Ma, which is because we have rounded the numbered to two decimal places rather than three. Thank you for pointing that out.

MODIFICATION: made and can be found high-lighted version of the revised Supplementary Materials.

**REVISED MANUSCRIPT**

[revised manuscript text omitted]
. Grains that are much older than the youngest population are hard to interpret if they are inherited grains or detrital, therefore they were discarded. A total of 3 younger grains (z9, z33, z34) that overlapped were considered for the final weighted mean age of the ash bed of 142.039 ± 0.058/0.069/0.17 Ma, MSWD 3.5.

**5.5. Ash bed LY5**

Ash bed LY5 is located 1.5 m below the contact of the Vaca Muerta Fm. and the Tordillo Fm (FS. 1B). Zircon yield was high ca. > 150 grains. Zircons crystals ranged from 20-100 μm in size. Grains were mainly prismatic with aspect ratio of 1:8 and rounded grains with aspect ratio of 1:3 were also very common. The ash bed is located in the Tordillo Fm, a silisiclastic unit. This sample had a significant amount of inherited and or detrital grains in the sample distribution. In a first batch of dated grains by CA-ID-TIMS the age distribution had a very large interval 180-450 Ma. Therefore, to optimize time we scanned the distribution of ages of the sample via LA-ICP-MS U-Pb geochronology. Grains were imaged via Cathodoluminescence and 250 grains were analyzed. Subsequently, the twenty youngest grains (135-145 Ma), based on Concordia ages, were selected to be analyzed via CA-ID-TIMS to obtain a reliable depositional age for the ash bed. A total of seven grains were selected to represent the age distribution of the sample, with thirteen grains being discarded for being too old (>~150 Ma) and considered either inherited grains or prolonged magmatic residence or magmatic recycling. A total of four younger grains (z10, z20, z38, z44) that overlapped were considered for the final weighted mean age of the ash bed of 147.112 ± 0.078/0/088/0.18 Ma, MSWD = 0.81. Radiogenic Pb ranged from 1 to 6 pg (TS.1).

**5.6. Ash bed MZT-81**

Ash bed MZT-81 is located in stratigraphic height ca. 22.5 m in the Mazatepec section. Zircon yield high ca. > 100 grains. Zircons crystals ranged from 40-80 μm in size. Grains were mainly prismatic with aspect ratio of 1:8. Radiogenic Pb ranged from 1 to 6 pg (TS.1), notably lower than other the other samples. A total of eight grains were selected to represent the age distribution of the sample, with 4 grains being discarded for being too old (>~150 Ma) and considered either inherited grains or prolonged magmatic residence, or magmatic recycling. A total of four younger grains (z4, z6, z8, z10) that overlapped were considered for the final weighted mean age of the ash bed of 140.512 ± 0.031/0/048/0.16 Ma, MSWD = 0.56

**6. Age-depth modelling - Bchron Code**

```
library(Bchron)

mydata2 = read.table(file='\\Users\\fortesd0\\Documents\\R\\win-
library\\3.4\\Bchron\\OregonPliens.txt', header=TRUE)
GlenOut = Bchronology(ages=mydata2$ages,
            ageSds=mydata2$ageSds,
            calCurves=mydata2$calCurves,
            positions=mydata2$position,
            positionThicknesses=mydata2$thickness,
            ids=mydata2$id
            predictPositions=seq(0,8400,by=10),iterations = 10000)
plot(GlenOut,main="NeuquenBchron",xlab='Age (Ma)',ylab='Depth (cm)',las=1)
summary(GlenOut)
```

```
summary(GlenOut, type='convergence')
summary(GlenOut, type='outliers')
```

```
Output <- cbind(apply(GlenOut$thetaPredict, 2, quantile,  probs = c(.0025)),
    apply(GlenOut$thetaPredict, 2, quantile,  probs = c(.5)),
    apply(GlenOut$thetaPredict, 2, quantile,  probs = c(.975)))
write.csv(Output, file = 'whatever.csv', quote=FALSE, row.names = FALSE)
```

```
acc_rate = summary(GlenOut, type = 'acc_rate')
plot(acc_rate[,'age_grid'], acc_rate[,'50%'], type='l', ylab = 'cm per year', xlab = 'Age (k cal
years BP)', ylim = range(acc_rate[,-1]))
lines(acc_rate[,'age_grid'], acc_rate[,'2.5%'], lty='dotted')
lines(acc_rate[,'age_grid'], acc_rate[,'97.5%'], lty='dotted')
sed_rate = summary(GlenOut, type = 'sed_rate', useExisting = FALSE)
plot(sed_rate[,'position_grid'], sed_rate[,'50%'], type='l', ylab = 'Years per cm', xlab = 'Depth
(cm)', ylim = range(sed_rate[,-1]))
lines(sed_rate[,'position_grid'], sed_rate[,'2.5%'], lty='dotted')
lines(sed_rate[,'position_grid'], sed_rate[,'97.5%'], lty='dotted')

write.csv(sed_rate, file = 'NeuquenBchron_sed_rates.csv', quote=FALSE, row.names =
FALSE)
write.csv(GlenOut, file = 'NeuquenBchron_sed_rates.csv', quote=FALSE, row.names = FALSE)
```

**7. Supplementary Figure 1 – Fig. S1**

| | Conusphaera mexicana | Watznaueria barnesiae | Watznaueria fossacincta | Cyclagelosphaera margerelii | Watznaueria britannica | Cyclagelosphaera deflandrei | Watznaueria communis | Hexalithus noeliae | Zeugrhabdotus embergeri | Polycostella senaria | Cocosfera | Helenea chiastia | Watznaueria manivitiae | Nannoconus kamptneri minor | Nanoconus sp. | Retecapsa octofenestrata | Retecapsa surirella | Nannoconus steinmannii | Nannoconus globulus | Zeugrhabdotus erectus |
|---|---|---|---|---|---|---|---|---|---|---|---|---|---|---|---|---|---|---|---|---|
| MZT-92 | | X | | X | | | | | | | | | | | | | | X | | |
| MZT-87 | | X | X | X | X | | | | | | | | X | X | X | | X | **X** | X | X |
| MZT-84 | | X | | | | | | | | | | | | | | X | | | | |
| MZT-69 | | X | | X | | | | | | X | | | | | | | | | | |
| MZT-68 | | | X | | X | | | | X | X | | | | | X | | | | | |
| MZT-65 | | X | X | | | | | X | | X | | X | X | **X** | | | | | | |
| MZT-58 | | X | X | X | | | | | X | | X | | | | | | | | | |
| MZT-55 | | X | | X | | | | | | X | | | | | | | | | | |
| MZT-51 | | X | X | | | | | | | | | | | | | | | | | |
| MZT-47 | | X | | X | X | X | | | | | | | | | | | | | | |
| MZT-45 | **X** | X | X | | | | | | X | | | | | | | | | | | |
| MZT-30 | X | X | | X | X | X | X | X | | | | | | | | | | | | |
| MZT-25 | | X | | | | | | | | | | | | | | | | | | |
| MZT-16 | | X | X | | | | | | | | | | | | | | | | | |
| MZT-15 | X | X | X | X | | | | | | | | | | | | | | | | |
| MZT-12 | X | X | X | X | | | | | | | | | | | | | | | | |
| MZT-6 | **X** | X | X | X | X | | | | | | | | | | | | | | | |

**Supplementary Figure 1.** Stratigraphic distribution of calcareous nannofossils of Mazatepec section, Mexico.

**8. Supplementary Figure 2 – Fig. S2**

[Figure]

**Supplementary Figure 2** – Field photos. A) Field figure from the Las Loicas section. Location of ash bed LL10 and the location of the JKB in the section. B) Field figure of the lower part of the La Yesera section where the contact between the Vaca Muerta and the Tordillo formations, and the location of ash bed LY5. C) location of the JKB in the Mazatepec section in Mexico, see bed MTZ-46 and MTY-45 (see Lopey-Matinez et al. 2013) D) Outcrop view of the Mazatepec section.

**9. Data tables**

**9.1.          U-Pb geochronology data table TS.1**

| Fraction | | Dates (Ma) | | | | | Composition | | | | | | |
|---|---|---|---|---|---|---|---|---|---|---|---|---|---|
| | | $206Pb/$ $238U$ $<Th>$ **a** | $\pm2\sigma$ abs | $207Pb/$ $235U$ **b** | $\pm2\sigma$ abs | $Th/$ $U$ **c** | $Pb^*$ $(pg)$ **d** | $Pbc$ $(pg)$ **e** | $206Pb/$ $238U$ **f** | $\pm2\sigma$ % | $207Pb/$ $235U$ **f** | $\pm2\sigma$ % |
| **LL3** | | | | | | | | | | | | |
| | z1 | 139.43 | 0.11 | 138.0 | 1.1 | 0.97 | 10.4 | 0.75 | 0.021852 | 0.080 | 0.1456 | 0.88 |
| | z2 | 139.36 | 0.11 | 138.75 | 0.76 | 0.88 | 9.76 | 0.43 | 0.021842 | 0.079 | 0.14642 | 0.59 |
| | z11 | 139.47 | 0.12 | 138.7 | 1.3 | 0.80 | 3.37 | 0.27 | 0.021858 | 0.088 | 0.1464 | 0.98 |
| | z21 | 139.45 | 0.11 | 139.8 | 1.1 | 0.97 | 7.21 | 0.48 | 0.021856 | 0.082 | 0.1476 | 0.84 |
| | z22 | 139.44 | 0.12 | 139.2 | 1.2 | 0.79 | 4.10 | 0.22 | 0.021853 | 0.086 | 0.1469 | 0.93 |
| | **z23** | **139.16** | **0.14** | **136.4** | **1.5** | **0.90** | **4.68** | **0.42** | **0.021809** | **0.10** | **0.1437** | **1.2** |
| | **z25** | **139.244** | **0.067** | **136.71** | **0.70** | **0.75** | **5.08** | **0.21** | **0.021822** | **0.048** | **0.14413** | **0.55** |
| | **z26** | **139.27** | **0.10** | **139.45** | **0.77** | **0.79** | **7.78** | **0.37** | **0.021826** | **0.075** | **0.14721** | **0.59** |
| | **z32** | **139.23** | **0.13** | **139.2** | **1.1** | **0.90** | **6.41** | **0.37** | **0.021820** | **0.093** | **0.1469** | **0.88** |
| **LL9** | | | | | | | | | | | | |
| | **z2** | **139.98** | **0.11** | **144.6** | **5.2** | **0.73** | **8.68** | **1.02** | **0.021938** | **0.077** | **0.1531** | **3.9** |
| | z5 | 140.191 | 0.063 | 141.2 | 1.2 | 0.47 | 14.0 | 0.40 | 0.0219707 | 0.045 | 0.1492 | 0.90 |
| | z7 | 140.19 | 0.14 | 141.7 | 2.3 | 0.53 | 8.56 | 0.46 | 0.021971 | 0.097 | 0.1498 | 1.8 |
| | **z12** | **139.99** | **0.16** | **150** | **11** | **0.37** | **1.94** | **0.55** | **0.021938** | **0.11** | **0.159** | **8.2** |
| | z33 | 140.456 | 0.071 | 141.3 | 1.0 | 0.47 | 10.3 | 0.24 | 0.022013 | 0.051 | 0.1493 | 0.78 |
| | z34 | 140.814 | 0.078 | 141.5 | 1.7 | 0.78 | 5.62 | 0.21 | 0.022071 | 0.056 | 0.1495 | 1.3 |
| | **z51** | **139.87** | **0.22** | **153** | **15** | **0.37** | **2.54** | **0.96** | **0.021919** | **0.16** | **0.162** | **11** |
| | z52 | 140.67 | 0.11 | 143.6 | 5.0 | 0.81 | 3.13 | 0.34 | 0.022048 | 0.078 | 0.1519 | 3.7 |
| | z53 | 140.18 | 0.14 | 140.7 | 2.5 | 0.65 | 5.33 | 0.29 | 0.021970 | 0.10 | 0.1486 | 1.9 |
| | z54 | 140.35 | 0.11 | 142.4 | 3.3 | 0.71 | 3.72 | 0.27 | 0.021997 | 0.080 | 0.1506 | 2.5 |
| | **z55** | **139.945** | **0.097** | **141.5** | **2.5** | **0.50** | **6.57** | **0.39** | **0.021932** | **0.070** | **0.1495** | **1.9** |
| **LL10** | | | | | | | | | | | | |
| | z11 | 140.51 | 0.13 | 140.1 | 1.4 | 0.82 | 4.08 | 0.37 | 0.022023 | 0.096 | 0.1479 | 1.1 |
| | z12 | 141.20 | 0.35 | 140.2 | 3.8 | 0.93 | 1.01 | 0.24 | 0.022134 | 0.25 | 0.1481 | 2.9 |

|        | z    | 206/238 age | ± | 207/235 age | ± | 207/206 | ± | 206/238 ratio | ± | 207/235 ratio | ± |
|--------|------|--------|-------|--------|------|------|------|-----------|-------|---------|------|
|        | z13  | 140.61 | 0.26 | 139.6 | 3.0 | 0.93 | 2.57 | 0.49 | 0.022039 | 0.18 | 0.1474 | 2.3 |
|        | z14  | 140.68 | 0.23 | 139.5 | 2.6 | 1.10 | 2.78 | 0.45 | 0.022052 | 0.16 | 0.1473 | 2.0 |
|        | **z41** | **140.37** | **0.21** | **138.0** | **2.2** | **0.84** | **2.17** | **0.28** | **0.022001** | **0.15** | **0.1456** | **1.7** |
|        | **z42** | **140.29** | **0.12** | **140.3** | **1.2** | **1.14** | **6.31** | **0.45** | **0.021990** | **0.085** | **0.1481** | **0.94** |
|        | **z44** | **140.32** | **0.16** | **139.8** | **1.2** | **1.02** | **4.86** | **0.32** | **0.021994** | **0.12** | **0.1476** | **0.96** |
|        | z45  | 140.55 | 0.27 | 138.3 | 2.8 | 0.87 | 1.42 | 0.19 | 0.022029 | 0.19 | 0.1459 | 2.2 |
| **MZT-81** |  |  |  |  |  |  |  |  |  |  |  |  |
|        | **z4**  | **140.478** | **0.079** | **140.57** | **0.74** | **0.42** | **10.4** | **0.54** | **0.022016** | **0.057** | **0.14848** | **0.57** |
|        | **z6**  | **140.504** | **0.083** | **140.41** | **0.85** | **0.53** | **9.91** | **0.60** | **0.022021** | **0.060** | **0.14830** | **0.65** |
|        | z7   | 141.03 | 0.12 | 141.7 | 1.3 | 0.43 | 3.31 | 0.30 | 0.022103 | 0.088 | 0.1498 | 0.97 |
|        | **z8**  | **140.518** | **0.047** | **140.39** | **0.25** | **0.53** | **11.1** | **0.17** | **0.0220228** | **0.033** | **0.14828** | **0.19** |
|        | **z10** | **140.528** | **0.061** | **140.22** | **0.45** | **0.53** | **10.9** | **0.33** | **0.0220243** | **0.044** | **0.14809** | **0.34** |
|        | z11  | 140.85 | 0.10 | 140.4 | 1.1 | 0.48 | 6.49 | 0.51 | 0.022075 | 0.075 | 0.1483 | 0.83 |
|        | z12  | 141.22 | 0.29 | 141.4 | 3.4 | 0.47 | 2.47 | 0.60 | 0.022134 | 0.21 | 0.1494 | 2.6 |
|        | z15  | 141.02 | 0.14 | 140.1 | 1.4 | 0.53 | 5.98 | 0.58 | 0.022102 | 0.099 | 0.1479 | 1.1 |
| **LL13** |  |  |  |  |  |  |  |  |  |  |  |  |
|        | **z9**  | **142.106** | **0.085** | **142.49** | **0.76** | **0.64** | **8.39** | **0.41** | **0.022275** | **0.060** | **0.15066** | **0.57** |
|        | **z33** | **141.93** | **0.11** | **142.2** | **1.2** | **0.76** | **7.49** | **0.51** | **0.022247** | **0.076** | **0.1503** | **0.88** |
|        | **z34** | **142.05** | **0.12** | **142.6** | **1.1** | **0.93** | **7.09** | **0.39** | **0.022267** | **0.081** | **0.1508** | **0.80** |
| **LY5** |  |  |  |  |  |  |  |  |  |  |  |  |
|        | **z10** | **147.8** | **1.1** | **144** | **13** | **0.71** | **0.665** | **0.56** | **0.02319** | **0.77** | **0.152** | **9.7** |
|        | z19  | 148.07 | 0.42 | 146.8 | 4.9 | 0.51 | 1.49 | 0.51 | 0.023221 | 0.29 | 0.1556 | 3.6 |
|        | **z20** | **147.12** | **0.26** | **144.5** | **3.0** | **0.77** | **3.92** | **0.75** | **0.023071** | **0.18** | **0.1529** | **2.2** |
|        | z22  | 148.01 | 0.36 | 147.0 | 3.8 | 0.39 | 1.34 | 0.34 | 0.023211 | 0.25 | 0.1558 | 2.8 |
|        | **z38** | **146.99** | **0.29** | **145.3** | **3.3** | **0.74** | **2.07** | **0.45** | **0.023050** | **0.20** | **0.1539** | **2.5** |
|        | **z44** | **147.118** | **0.086** | **146.49** | **0.82** | **0.75** | **3.37** | **0.15** | **0.023071** | **0.058** | **0.15520** | **0.60** |
|        | z67  | 147.740 | 0.093 | 147.09 | 0.96 | 0.49 | 6.20 | 0.41 | 0.023168 | 0.064 | 0.1559 | 0.70 |

Legend T.S1
a)   Corrected for initial Th/U disequilibrium using the radiogenic 208Pb and Th/U (magma)= 3.5
b)   Isotopic dates calculated using 238= 1.55125 E-10 (Jaffey et al. 1971) and 235=9.8485E-10
  (Jaffey et al. 1971)
c)   Th contents calculated from radiogenic 208Pb and $239^{Th}$-corrected 206Pb/207Pb date of the
  sample, assuming concordance between U-Pb and Th-Pb systems
d)   Total mass of radiogenic Pb
e)   Total mass of common Pb
f)   Measured ratios correct for fractionation, tracer and blank.
## 9.2.    Age-depth model data table TS.2

| Stratigraphic Height (m) | Age (Ma) | Std Err (2 S.D) (Ma) | Stratigraphic Height (m) | Age (Ma) | Std Err (2 S.D) (Ma) |
|---|---|---|---|---|---|
| 54 | 139.24 | 0.05 | 15 | 141.31 | 0.56 |
| 53 | 139.30 | 0.15 | 14 | 141.37 | 0.56 |
| 52 | 139.35 | 0.20 | 13 | 141.43 | 0.52 |
| 51 | 139.41 | 0.22 | 12 | 141.50 | 0.51 |
| 50 | 139.46 | 0.23 | 11 | 141.56 | 0.50 |
| 49 | 139.51 | 0.23 | 10 | 141.62 | 0.49 |
| 48 | 139.57 | 0.24 | 9 | 141.68 | 0.47 |
| 47 | 139.62 | 0.23 | 8 | 141.74 | 0.44 |
| 46 | 139.68 | 0.24 | 7 | 141.80 | 0.40 |
| 45 | 139.73 | 0.23 | 6 | 141.86 | 0.36 |
| 44 | 139.79 | 0.22 | 5 | 141.92 | 0.33 |
| 43 | 139.84 | 0.21 | 4 | 141.97 | 0.23 |
| 42 | 139.90 | 0.17 | 3 | 142.04 | 0.06 |
| 41 | 139.96 | 0.07 | 2 | 142.10 | 0.32 |
| 40 | 140.00 | 0.10 | 1 | 142.15 | 0.42 |
| 39 | 140.03 | 0.12 | | | |
| 38 | 140.07 | 0.12 | | | |
| 37 | 140.11 | 0.13 | | | |
| 36 | 140.15 | 0.13 | | | |
| 35 | 140.18 | 0.13 | | | |
| 34 | 140.22 | 0.13 | | | |
| 33 | 140.27 | 0.12 | | | |
| 32 | 140.31 | 0.11 | | | |
| 31 | 140.35 | 0.08 | | | |
| 30 | 140.42 | 0.23 | | | |
| 29 | 140.48 | 0.30 | | | |
| 28 | 140.54 | 0.37 | | | |
| 27 | 140.60 | 0.41 | | | |

| 26 | 140.66 | 0.43 |
| 25 | 140.72 | 0.46 |
| 24 | 140.78 | 0.50 |
| 23 | 140.84 | 0.50 |
| 22 | 140.90 | 0.52 |
| 21 | 140.96 | 0.53 |
| 20 | 141.02 | 0.55 |
| 19 | 141.08 | 0.56 |
| 18 | 141.14 | 0.55 |
| 17 | 141.20 | 0.56 |
| 16 | 141.26 | 0.56 |

---

## Referee Report (RR1)

Comment on the manuscript « **Cross-continental age calibration of the Jurassic/Cretaceous boundary** » by Luis Lena et al.

Bruno Galbrun

Dear Editor

Thank you for giving me the opportunity to read this manuscript.
As this manuscript has already been the subject of two detailed and argued reviews, I will only make general comments and analyse whether the authors have taken these previous reviews into account.

I carefully read the manuscript, the reviewers' comments and the authors' responses to the comments.

General comment

This manuscript is quite interesting because it provides radiometric data over a poorly documented time interval. This is the very positive point of this manuscript. Unfortunately, this manuscript suffers from weaknesses: no magnetostratigraphic data (or no real discussion on previous data) while magnetostratigraphy is a key element to discuss the position of the Jurassic-Cretaceous boundary, too poor biostratigraphic data whose reliability is not sufficiently criticized, completeness of the sections not sufficiently discussed.
These shortcomings make the manuscript's conclusions a little too affirmative.

The authors have taken into account most of J. Palfy's comments. However, they have taken into account only very few of W. Wimbledon's comments. This is very surprising because W. Wimbledon as chairman of the ICS Berriasian working group is probably the most competent person to discuss the Jurassic-Cretaceous boundary, and his comments are pertinent.
The most common sentence in the authors' response is "*No modification was made since we feel we have answered this comment*". It seems that the authors spent more time denying the comments than trying to make the necessary changes.

Specific comments on the Introduction

Page 1, lines 23-24 : « *Approaches have varied from the coupling of magnetostratigraphy with biostratigraphy (Larson and Hilde, 1975)…* ».
This reference is inadequate: in their manuscript Larson and Hilde only consider oceanic magnetic anomalies (the Hawaiian lineation pattern), there were no magnetostratigraphic results on the Jurassic-Cretaceous boundary in the early 1970s… They just stuck to their magnetic polarity sequence the Geological Society of London (1964) time scale.

Page 1, Line 25 : « *…(Gradstein et al., 1995; Kent and Gradstein, 1985; Lowrie and Ogg, 1985; Ogg and Lowrie, 1986) ».*
Maybe add the reference: Channell et al., SEPM Sp Pub 54, 1995.

Page 1, Lines 26-28 : « *Due to the scarcity of numerical ages for the Late Jurassic and Early Cretaceous, a lot of the available JKB age data was derived from interpolating distant tie points for arguably large intervals of time (~25 Ma)* ».
It is not clear whether the authors refer here to sedimentary successions or to marine magnetic anomaly M-sequence. It's a little more complicated. The authors should provide some details on the general methodology previously used to propose an age of the Jurassic-

Cretaceous boundary : magnetostratigraphic results on sedimentary successions with very rare radiometric ages + correlations with the M-sequence of marine magnetic anomalies + very rare radiometric ages directly on the M-sequence (one or two on the Middle Jurassic ?) + Interpolation on the M-sequence between these tie-points (of various origins) considering a constant oceanic spreading rate + some cyclostratigraphic results (especially on Oxfordian and Kimmeridgian)... and so on. This methodology is widely developed in the GTS2012 Elsevier book (Geomagnetic Polarity Time Scale, Jurassic and Cretaceous chapters).

Page 2, Line 28 : « *More importantly, the data presented here permits to put o the test the currently ICS accepted age of the JKB* ».
As this goal seems to be an important objective of the authors they should better explain in this Introduction what is the criterion chosen to define the Jurassic-Cretaceous boundary in the most recent Geological Time Scale (the ICS and/or the GTS2012 - Gradstein et al -), as well as how a numerical age is proposed for this boundary (see my previous comment on the correlations between magnetostratigraphy and marine magnetic anomalies). This is necessary because it is not sufficiently included in the discussion. Perhaps the authors could at least indicate in this introduction the age of this boundary in the ICS scale, not only waiting the section 4.6.

Conclusion

This manuscript is likely to be published due to the provision of radiometric data. However, I wonder about the authors' conclusions. It seems to me that the main conclusion, the age of the Jurassic-Cretaceous boundary must be younger than currently accepted, seems premature. This manuscript provides numerical data but is not at all a "*Cross-continental age calibration...*", I think the title should be changed.
It seems to me that the authors should be a little more cooperative. I recommend the publication of this manuscript if only the authors make an effort to do so.

---

## Author Response (AR2)

Dear Editor S. Gardin,

First, we would like to thank you for your valid comments and for inviting B. Galbrun to review the manuscript. Comments from you both have made us re-think the aims of the manuscript and revisit previous comments from W. Wimbledon and J. Pálfy. We are confident that all comments have significantly improved this second revised version.

**Modifications in the main text**

- We have changed the title of the manuscript to be more in synchrony with limitations of our data and conclusions, as pointed out by B. Galbrun. The new title is: "High-precision U-Pb ages in the Early Tithonian to Early Berriasian and implications for the numerical age of the Jurassic/Cretaceous boundary"
- 2. The abbreviation of "Jurassic/Cretaceous boundary" was replaced by "J/K boundary" as suggested by the Editor and W. Wimbledon in comment 1.13 (See reply to W. Wimbledon) and also by the Editor.
- 3. The introduction was completely re-written to accommodate the insightful comment from B. Galbrun suggesting we provide a more detailed methodology on the assumptions the current numerical age of the J/K boundary is anchored on.
- 4. We have revisited J. Pálf's comment 2.2 (see reply to J. Pàlfy) on having a proper "Geological and stratigraphic setting" section. We have renamed section 2 to "Geological context and studied sections". Here we have expanded the section to a more thorough description of the geological context.
- 5. Figure 1 was replaced by a figure that could better serve the re-written section 2, "Geological context and studied sections". We have provided regional geological maps for each section.
- 6. We have revisited J. Pálfy's comment that suggested we should separate the Results from the Discussion section. In this revised version, Section 4 is now the Results sections, with three subsections.
- 7. In the Results section, we have added a subsection "*Numerical age of faunal assemblages in studied sections*" to accommodate W. Wimbledon's comment 1.10 where we evaluate the numerical age of the key biostratigraphical markers. In this section, to estimate the age of the base of the Alpina Subzone, we use the sedimentation rate of 2.5 cm/ka which is much more realistic due to the stratigraphical and sedimentological setting.

Letter to the Editor S. Gardin.

- 8. Due to B. Galbrun's comments on the biostratigraphy, we have decided to open the Discussion section (now section 5) with a "*The Chronostratigraphic and biostratigraphical framework of the studied sections*". Here we discussion the biostratigraphical background from both sections with regards to its limitations especially when compared to Mediterranean Tethys sections and the working model for the J/K boundary of Wimbledon 2017. Incidentally, this new section also addresses the previous comment 1.8 from W. Wimbledon, where he suggests the biostratigraphic background should be made clear before considering the radiometric ages.
- 9. As pointed out by the Editor and B. Galbrun the biostratigraphy is poor; therefore, we have removed our previous discussion on the global correlations since it was an over interpretation of our data. In the absence of magnetostratigraphy such a correlation is not sound because the precise location of the J/K boundary in the studied section is contentious. We can only correlated the age of the studied section based on biostratigraphy and geochronology. Nevertheless, this is not a sufficient condition to locate the J/K boundary, as the lack of magnetostratigraphy hinders such correlations. In the previous revised version, we justified the mismatch in age between both sections based on J. Pálfy's comments 2.1 and 2.14 where he suggests the mismatch is a result of the diachronous FAD-LAD of the key taxa. Nevertheless, as we discuss in section 5.1 *"The Chronostratigraphic and biostratigraphical framework of the studied sections"* the calcareous nannofossil biostratigraphy is still preliminary as well as the Calpionellids in Las Loicas, and does not yet fully agree with the current framework for the J/K boundary of Wimbledon 2017. Therefore, long-distance correlations at this juncture are not advisable or not possible as suggested by W. Wimbledon on comment 1.1.
- 10. In the new section 5.2 we have readily updated the references on the calcareous nannofossils as was well pointed out by the Editor and W. Wimbledon on comment 1.4.
- 11. In substitution of section 4.5 we have limited our discussion on the possible age of the J/K age using our geochronological data to the new subsection 5.4 "*Constraining the numeric age of the J/K boundary between the studied sections*". Here we attempt to constrain the age of the boundary based on the age of markers in both section, and refrain from any global meaning for the age of the J/K boundary from Las Loicas and Mazateptec, but rather suggest that these results indicate that the age of the boundary could be younger than accepted.

Letter to the Editor S. Gardin.

- 12. We have kept our discussion on the base of the Vaca Muerta and the Early Tithonian, since none of the reviewers have taken any issues with the discussion, and it seems reasonably sound.
- 13. The previous discussion on "A case for a younger age on the J/K boundary" has been replaced by the subsection 5.4 "Implications for the numerical age of the J/K boundary". This is a less conclusive discussion on the age of the J/K boundary as an accommodation to B. Galbrun's comments on how affirmative the revised version was. Here we refocus the discussion on the possibility that the age of the J/K boundary could be younger. We discuss this possibility along with newer ages in the Early Cretaceous and how these could affect the M-sequence model of Ogg (2012) and how our newer ages would fit in this scenario.

**Modifications to Figure**

- We have replaced Figure 1 from the previous revised version by a Regional Location map, with geological maps of the studied sections. The previous Figure 1 contained suggestions of migratory routes of key taxa, since we no longer touch on that issue we have replaced it. This also satisfies the comments from J. Palfy that we should replace the Figure we, and thus we did.
- Figure 4 in the previous revised version has now been renamed to Figure 2 as suggested by J.
   Pálfy in his comments technical corrections. The reviewer suggested moving the main figure, i.e. Figure 4, to the manuscript of the Geological settings section and is now figure 2.
- 3. The now Figure 2 has dotted lines in the calcareous nannofossil zonations in Las Loicas and Mazapetec as well as occurrences of nannofossils are now as FO an no longer as FAD as found in Vennari et al. (2014) or previous versions of this manuscript.
- 4. We have added a Figure 5 to the revised version. The figure is a modified after Figure 1 of Wimbledon 2017, which is used to indicate the correlation with the current numerical ages around the J/K boundary.

Lena, et al., Cross-continental age calibration of the Jurassic/Cretaceous boundary

Reply to Comments by B. GalbrunSolid Earth, EGU

**Modifications to the manuscript with respect to the comments by reviewer B. Galbrun (reviewer #3) on the manuscript "Cross-continental age calibration of the Jurassic/Cretaceous boundary"**

Comments by the reviewer have been copied and pasted in the *italic* blue font, the replies
are found immediately below in regular **black** font. The comments are in order of appearance in
the reviewer's comments. We have taken the liberty of numbering the comments from 3.1
through 3.8.

8 Preface

9 First we would like to thank the reviewer for his very comments insightful comments,
10 which have made us re-think the aims of the manuscript. Additionally, his comments have made
11 us re-think and re-visit the previous comments from W. Wimbldeon and J. Pálfy and re-write the
12 manuscript in a way the address all of these comments.

13

**14 General comment**

3.1 "This manuscript is quite interesting because it provides radiometric data over a poorly
documented time interval. This is the very positive point of this manuscript.
Unfortunately, this manuscript suffers from weaknesses: no magnetostratigraphic data
(or no real discussion on previous data) while magnetostratigraphy is a key element to
discuss the position of the Jurassic-Cretaceous boundary, too poor biostratigraphic data
whose reliability is not sufficiently criticized, completeness of the sections not sufficiently
discussed."

REPLY: Also agree with his point of view that the lack of magnetostratigraphy limits our data set. Therefore, we agree with the reviewer that the without the aid of magnetostratigraphy we can not be too affirmitive as to the location of the J/K boundary. We have included a section within the discussion that adresses the limitations of the biostratigraphy in the studied sections. We have also refrained from being to affirmative and conclusive, now that the reliability of the biostratigraphy has been pointed out to us.

1

**Lena, et al., Cross-continental age calibration of the Jurassic/Cretaceous boundary**

Reply to Comments by B. Galbrun

**Solid Earth, EGU**

3.2 The authors have taken into account most of J. Palfy's comments. However, they have taken into 28 29 account only very few of W. Wimbledon's comments. This is very surprising because W. 30 Wimbledon as chairman of the ICS Berriasian working group is probably the most competent person to discuss the Jurassic-Cretaceous boundary, and his comments are pertinent. The most 31 32 common sentence in the authors' response is "No modification was made since we feel we have 33 answered this comment". It seems that the authors spent more time denying the comments than 34 trying to make the necessary changes." 35 REPLY: We have indeed re-visited some of W. Wimbledon's comments. For instance, B. Galbruns's 36 comment that we should criticize our biostratigraphy is in line with W. Wimbledon's comment that we 37 should have describe the chrono and biostratigraphical framework of the studied sections, which is now in 38 the 5.1 of the discussion. Additionally, we address the seemingly odd calpionellids assemblage in Las Loicas, as pointed out by Wimbledon in his comments. The examination of the calibrated dated ash bed 39 with the paleontological markers foun din each section, now found in the Restuls section 4.3. We have 40 41 taken W. Wimbledon's suggestion to use the appropriate nomenclature for the Jurassic/Cretaceous 42 boundary as J/K boundary. 43 44 Specific comments on the Introduction 45 46 3.3 "Page 1, lines 23-24 : « Approaches have varied from the coupling of magnetostratigraphy with biostratigraphy (Larson and Hilde, 1975)... ». This reference is inadequate: in their 47 48 manuscript Larson and Hilde only consider oceanic magnetic anomalies (the Hawaiian lineation 49 pattern), there were no magnetostratigraphic results on the Jurassic-Cretaceous boundary in the 50 early 1970s... They just stuck to their magnetic polarity sequence the Geological Society of 51 London (1964) time scale. 52 REPLY: The reviewer is right, and we have rectified. 53 3.4 Page 1, Line 25 : « ... (Gradstein et al., 1995; Kent and Gradstein, 1985; Lowrie and Ogg, 1985; Ogg and Lowrie, 1986) ». Maybe add the reference: Channell et al., SEPM Sp Pub 54, 1995. 54 55 REPLY: We have taken the reviewer's advice and added the reference. 56 3.5 Page 1, Lines 26-28 : « Due to the scarcity of numerical ages for the Late Jurassic and Early Cretaceous, a lot of the available JKB age data was derived from interpolating distant tie 57 58 points for arguably large intervals of time (~25 Ma) ».

**Reply to Comments by B. Galbrun**

**Solid Earth, EGU**

59 It is not clear whether the authors refer here to sedimentary successions or to marine magnetic 60 nomaly M-sequence. It's a little more complicated. The authors should provide some details on 61 the general methodology previously used to propose an age of the JurassicCretaceous boundary : magnetostratigraphic results on sedimentary successions with very rare radiometric ages + 62 63 correlations with the M-sequence of marine magnetic anomalies + very rare radiometric ages 64 directly on the M-sequence (one or two on the Middle Jurassic ?) + Interpolation on the M-65 sequence between these tie-points (of various origins) considering a constant oceanic spreading rate + some cyclostratigraphic results (especially on Oxfordian and Kimmeridgian)... and so on. 66 67 This methodology is widely developed in the GTS2012 Elsevier book (Geomagnetic Polarity Time Scale, Jurassic and Cretaceous chapters). 68

REPLY: We have re-written the introduction of the of the manuscript to accommodate this important
comment. We have given a more thorough description on the methodology used to arrive at the age of the
J/K boundary. This also addresses the comment 1.9 by W. Wimbledon which requested that we discuss at
more lengths the time scales used by GTS of Gradstein 2012.

- 73 3.6 Page 2, Line 28 : « More importantly, the data presented here permits to put o the test the currently ICS accepted age of the JKB ». As this goal seems to be an important objective of the 74 75 authors they should better explain in this Introduction what is the criterion chosen to define the 76 Jurassic-Cretaceous boundary in the most recent Geological Time Scale (the ICS and/or the 77 GTS2012 - Gradstein et al -), as well as how a numerical age is proposed for this boundary (see my previous comment on the correlations between magnetostratigraphy and marine magnetic 78 79 anomalies). This is necessary because it is not sufficiently included in the discussion. Perhaps the 80 authors could at least indicate in this introduction the age of this boundary in the ICS scale, not
- 81 *only waiting the section 4.6.* "

REPLY: We have decided to answer these three comments as one, because they all seem somewhat
related. Ultimately, the all these three comments pertain to how badly we have address the assumptions
made to arrive the current age of the J/K boundary. In the previous version we very briefly described
which came across as confusing and unclear. We have completely re-written the Introduction

- 86
- 87 *Conclusion*

88 3.7 However, I wonder about the authors' conclusions. It seems to me that the main conclusion,
89 the age of the Jurassic-Cretaceous boundary must be younger than currently accepted, seems
90 premature.

**Lena, et al., Cross-continental age calibration of the Jurassic/Cretaceous boundary**

Reply to Comments by B. GalbrunSolid Earth, EGU

91 REPLY: The reviewer is completely right. As he pointed out, the biostratigraphy is poor and as pointed 92 out still needs further improvement. The lack of magnetostratigraphy is also a major problem to challenge 93 the age of the boundary with such certainty. Therefore, we have made our conclusion mush less 94 affirmative since our data does not unequivocally allow the challenging of the J/K boundary age. 95

- 3.8 This manuscript provides numerical data but is not at all a "Cross-continental age
  calibration...", I think the title should be changed.
- 98 REPLY: We agree with the reviewer rand we have changed the title of the manuscript to something more
- 99 within the confines of what our data allows.

Reply to Comments by S. Gardin

Solid Earth, EGU

**Reply to comments by editor S. Gardin on the manuscript "Cross continental age calibration of the Jurassic/Cretaceous boundary"**

Comments by the editor have been copied and pasted in the *italic* blue font, and the answers are found immediately below in regular black font. The comments are in order of appearance in the editor comments.

4.1 FAD's and LAD's and their use for dating and correlating. I personally recommend to use FAD and LAD when it is sure that we are closest to the very first, evolutive appearance /disappearance of a taxon or species (FAD's and LAD's age can be extrapolated). In a more specific contest such as in Mazatepec section it is preferable to use FO (first occurrence) or LO (last occurrence) because these bio-horizons are highly affected by taphonomic processes (preservation, facies change and unfavourable lithologies...), sample density, and also different analytical methods which can alter their « true » apparition/disparition level. You should consider a larger confidence interval for the bio-horizons, especially when they result from a poor data set.

The Editor's suggestion about this point is correct and we agree with her. In the new version of the manuscript, the FADs and LADs were replaced by FO and LO.

4.2 Calcareous nannofossil biostratigraphy. The calcareous nanofossil data set of Mazatepec is, unfortunately, very poor so I'would't « force » such poor data to fit in a stratigraphic framework.

We accepted Dr. Gardin's comments as valuable, for that reason we accepted using dot lines to define biozones in Figure 4.

4.3 The Umbria granulosa specimen reported in figure 3 (fig 3-K) which should illustrate calcareous nannofossils from the Mazatepec section in Mexico, is exactly the same

Reply to Comments by S. Gardin

Solid Earth, EGU

specimen reported by Vennari et al (2014) from Las Loicas section in Argentina ! Surprisingly, Umbria granulosa is not reported in the range chart of Mazatepec (supplementary material)... I hope that this unfortunate « copy and paste » is just accidental... Also, from the illustration the diagnosis of this species is wrong. Please, fix this issues. Once these modifications are made the paper will surely have a better impact and deserve publication.

The illustrated specimen of Umbria granulosa corresponds to the Las Loicas section, as it appears in the legend of Figure 3. This species has not been recognized in the section studied in Mexico as we reported in the Mazatepec range chart. In our opinion, the diagnosis of Umbria granulosa is correct and we have recorded it in several sections of the Nequen Basin (e.g. Arroyo Loncoche section, among others).

**High-precision U-Pb ages in the Early Tithonian to Early Berriasian and implications for the numerical age of the Jurassic/Cretaceous boundary**

Luis Lena1, Rafael López-Martínez2, Marina Lescano3, Beatriz Aguirre-Urrreta3, Andrea Concheyro3, 5 Verónica Vennari3, Maximiliano Naipauer3, Elias Samankassou1, Márcio Pimentel4, Victor A. Ramos3, Urs Schaltegger1

[revised manuscript text omitted]